# Self-lofting of wildfire smoke in the troposphere and stratosphere: simulations and space lidar observations

Kevin Ohneiser[1], Albert Ansmann[1], Jonas Witthuhn[1], Hartwig Deneke[1], Alexandra Chudnovsky[2], Gregor Walter[1], and Fabian Senf[1]

[1]Leibniz Institute for Tropospheric Research, Leipzig, Germany
[2]Tel Aviv University, Porter School of the Environment and Earth Sciences, Tel Aviv, Israel

**Correspondence:** K. Ohneiser
(ohneiser@tropos.de)

**Abstract.** Wildfire smoke is known as a highly absorptive aerosol type in the shortwave wavelength range. The absorption of Sun light by optically thick smoke layers results in heating of the ambient air. This heating is translated into self-lofting of the smoke up to more than 1 km in altitude per day. This study aims for a detailed analysis of tropospheric and stratospheric smoke lofting rates based on simulations and observations. The main goal is to demonstrate that radiative heating of intense smoke plumes is capable of lofting them from the lower and middle free troposphere (injection heights) up to the tropopause without the need of pyrocumulonimbus (pyroCb) convection. The further subsequent ascent within the lower stratosphere (caused by self lofting) is already well documented in the literature. Simulations of absorbed solar radiation by smoke particles and resulting heating rates, which are then converted into lofting rates, are conducted by using the ECRAD (European Centre for Medium-Range Weather Forecasts Radiation) scheme. As input parameters thermodynamic profiles from CAMS (Copernicus Atmosphere Monitoring Service) reanalysis data, aerosol profiles from ground-based lidar observations, radiosonde potential temperature profiles, CALIOP (Cloud Aerosol Lidar with Orthogonal Polarization) aerosol measurements, and MODIS (Moderate Resolution Imaging Spectroradiometer) aerosol optical depth retrievals were used. The sensitivity analysis revealed that the lofting rate strongly depends on aerosol optical thickness (AOT), layer depth, layer height, and black carbon (BC) fraction. We also looked at the influence of different meteorological parameters such as cloudiness, relative humidity, and potential temperature gradient. To demonstrate the applicability of our self-lofting model, we compared our simulations with the lofting processes in the stratosphere observed with CALIOP after major pyroCb events (Canadian fires, 2017, Australian fires 2019-2020). We analyzed long-term CALIOP observations of smoke layers and plumes evolving in the UTLS (upper troposphere and lower stratosphere) height region over Siberia and the adjacent Arctic Ocean during the summer season of 2019. Our results indicate that self-lofting contributed to the vertical transport of smoke. We hypothesize that the formation of a near-tropopause aerosol layer, observed with CALIOP, was the result of self-lofting processes because this is in line with the simulations. Furthermore, Raman-lidar-based aerosol typing (at Leipzig and the High Arctic) clearly indicated the dominance of smoke in the UTLS aerosol layer since August 2019, most probably also the result of smoke self-lofting.

# 1 Introduction

Uncontrolled intense fires on large areas of regional scale have become more frequently in recent years in many regions on Earth (Jolly et al., 2015; Peterson et al., 2021). Enormous amounts of biomass-burning smoke were emitted into the atmosphere by fire storms in Canada 2017 (Peterson et al., 2018) and Australia 2019-2020 (Peterson et al., 2021). When reaching the stratosphere, wildfire smoke can sensitively influence the stratospheric composition on a hemispheric scale (Bond et al., 2013; Baars et al., 2019; Kloss et al., 2019; Yu et al., 2019; Rieger et al., 2021; Ohneiser et al., 2022) and thus can affect the Earth's climate (Das et al., 2021; Yu et al., 2021; Hirsch and Koren, 2021; Stocker et al., 2021; Heinold et al., 2022; Rieger et al., 2021; Sellitto et al., 2022) and the ozone layer (Ohneiser et al., 2021, 2022; Voosen, 2021; Yu et al., 2021; Rieger et al., 2021; Stone et al., 2021; Solomon et al., 2022; Bernath et al., 2022; Ansmann et al., 2022). One typical way of biomass-burning smoke plumes to reach the stratosphere is via pyrocumulonimbus (pyroCb) convection (Fromm and Servranckx, 2003; Fromm et al., 2010; Peterson et al., 2018; Rodriguez et al., 2020). These fire-generated clouds can loft large smoke amounts to the tropopause level in less than an hour (Rosenfeld et al., 2007; Rodriguez et al., 2020). From here, smoke is able to ascend deeply into the lower stratosphere by so-called self-lofting processes. Khaykin et al. (2020), Ohneiser et al. (2020, 2022), Kablick et al. (2020), and Hirsch and Koren (2021) analyzed extended smoke layers, originating from the record-breaking Australian wildfires in December 2019 and January 2020, that ascended from 14 to more than 30 km height within 2 months between January 2020 and March 2020 as a result of self-lofting processes.

Figure 1 provides a first impression of the relevance of self-lofting on observable smoke layering features. MODIS satellite observations of smoke plumes that traveled from Australia to New Zealand in the beginning of January 2020 are shown. The brown colors, partly above white cloud layers (see yellow arrows in Fig. 1), indicate extended wildfire smoke fields in the UTLS (upper troposphere and lower stratosphere) height range. The inhomogeneous structures in the smoke layers reflect differences in the smoke aerosol optical thickness (AOT). Plume segments with high AOT above white, strongly sun-light-reflecting cloud fields absorb much more solar radiation because significant amounts of upwelling solar radiation is absorbed as well. Thus, they can ascend much faster in the stratosphere (for example on 5 January 2020 in Fig. 1) and reach greater heights than smoke plumes with similar AOT over cloud-free areas. As a consequence, originally well-defined smoke layers with clear base and top height and a vertical extent of, e.g., 1-2 km close to Australia may develop diffusive smoke structures caused by different ascent rates during long-range transport and may show up as inhomogeneous, 4-8 km deep layers at much higher altitudes after the journey of 10000 km. In addition, differences in wind speed at different traveling heights contribute to complex smoke profile structures and layering features, as observed over Punta Arenas in southern Chile in January 2020 (Ohneiser et al., 2020, 2022), far away from the smoke source regions. The dependence of smoke lofting on AOT, meteorological conditions, black carbon fraction, injection height and further relevant quantities will be discussed in Sect. 3.

Recently, smoke self-lofting was discussed as a potential option to loft smoke from the initial injection height of 2-6 km (Amiridis et al., 2010) to the tropopause (Ohneiser et al., 2021; Ansmann et al., 2021a). Large fires in Siberia in July and August 2019 caused a high smoke AOT of the order of 1-3 over an area north and northeast of Lake Baikal of more than 500-1000 km for several weeks. It was hypothesized that self-lofting of smoke, which absorbs solar radiation and heats the

air, was responsible for the ascent of large amounts of smoke towards the tropopause because pyroCb activity was low to that time. As indicated by preliminary simulations, ascent rates are low (of the order of days before smoke reaches the tropopause) compared to pyroCb lofting rates (of the order of one hour for the vertical transport from near surface heights to the tropopause).

During late summer 2019, smoke layers, to our opinion originating from these severe Siberian fires, were even observed in the stratosphere over Central Europe (Ansmann et al., 2021b). The smoke layers polluted also the UTLS region over the High Arctic until May 2020 and were thus observable during the first half of the MOSAiC (Multidisciplinary drifting Observatory for the Study of Arctic Climate) expedition, the largest Arctic research initiative in history (Ohneiser et al., 2021). The occurrence of smoke layers in the stratosphere at times without significant pyroCb activity motivated the study presented here.

Wildfire smoke is a highly absorptive aerosol type mostly consisting of organic carbon (OC) with a few percent of black carbon (BC). By absorbing solar radiation, optically thick smoke layers are able to considerably heat the ambient air. This heating creates buoyancy which may cause the warmed layers to ascend from tropospheric heights towards the stratosphere (Boers et al., 2010; de Laat et al., 2012) or from the tropopause or stratospheric heights (Yu et al., 2019; Torres et al., 2020) towards greater altitudes. The self-lofting process can prevail for several weeks to months (Kablick et al., 2020; Khaykin et al.,

2020; Allen et al., 2020; Lestrelin et al., 2021). The importance of self-lofting is that wildfire smoke can reach high altitudes in this way and can then be efficiently distributed over large parts of the hemisphere (Baars et al., 2019; Kloss et al., 2019; Rieger et al., 2021). This lofting process leads to a prolongation of the residence time of aerosols in the stratosphere and counteracts the sedimentation and removal of particles (Ohneiser et al., 2022).

Not only smoke layers are able to ascend. A similar lofting behavior was observed during the first few days after volcanic

eruptions (Muser et al., 2020; Stenchikov et al., 2021) caused by dense light-absorbing ash plumes. In the case of Kuwait oil fires in 1991, smoke layers were typically found at heights below 2-3 km within 50 km around the source region, and several plumes were detected later on at an altitude of 6-7 km after a travel of 2000 km (Limaye et al., 1991). In the spring and summer seasons during the Iraq War (2003-2010) extended plumes of mineral dust mixed with black smoke (due to military activities) were observed, covering large areas over Iraq and adjacent countries (Chudnovsky and Kostinski, 2020). The observed polluted

dust features may have been partly influenced by self-lofting effects prolonging residence times and dispersion of pollution and dust over larger areas especially during the summer half years.

Stratospheric smoke self-lofting processes have already been highlighted in a number of articles (Kablick et al., 2020; Khaykin et al., 2020; Allen et al., 2020; Torres et al., 2020; Das et al., 2021; Lestrelin et al., 2021; Heinold et al., 2022). de Laat et al. (2012) discussed tropospheric smoke self-lofting effects for the first time. However, a detailed tropospheric

analysis that includes an extended study of the impact of uncertainties in the numerous input parameters in simulations has not been presented. Such an in-depth analysis is presented in this article. Besides the error analysis, we will apply the developed self-lofting simulation tool to several cases of ascending stratospheric smoke layers as observed with the space lidar CALIOP (Cloud Aerosol Lidar with Orthogonal Polarization) of the CALIPSO (Cloud-Aerosol Lidar and Infrared Pathfinder Satellite Observations) mission (Winker et al., 2009). These comparisons of simulations with observations allow us to adjust important

input parameters such as the BC fraction and to obtain an improved insight into the chemical composition and microphysical properties of wildfire smoke particles.

The paper is organized as follows. In Sect. 2, the modeling tools applied to simulate the smoke self-lofting processes are introduced. Section 3 presents an uncertainty discussion with focus on the influence of the BC fraction, plume height and geometrical layer depth, and aerosol plume optical thickness on the smoke lofting rates. In Sect. 4.2 and 4.3, CALIOP observations of different ascending stratospheric smoke plumes originating from the record-breaking Canada wildfires (in August 2017) and Australia bushfires (in January-February 2020) are compared with respective model simulations. As an important part of the study, aerosol layers observed with CALIOP in the UTLS height range over Siberia and the Arctic Ocean in the summer of 2019 will be discussed in Sect. 5 with focus on potential smoke self-lofting aspects.

It is important to mention that the scientific discussion concerning the major cause of the aerosol in the ULTS height range over Siberia and the adjacent Arctic Ocean in the second half of 2019 shows no signs of dwindling. Opposing viewpoints have been expressed in recently published papers. One side argues that the major cause for the lower stratospheric aerosol is the Raikoke volcanic eruption (Boone et al., 2022), while the other side counters with the presence of an aerosol layer in the lower stratosphere, that consisted of a mixture of smoke (80-90% fraction) and sulfate aerosol (10-20%) (Ohneiser et al., 2021). The latter was largely observable with lidar in the UTLS height range over the High Artic until May 2020. In Sect. 5, we therefore present a complete order of events regarding the beginning of the long-lasting Siberian fire season (in June 2019), about the transport of smoke towards and across the Arctic Ocean (continuously from June to August 2019), and regarding the potential development of smoke layers near the tropopause and within the entire UTLS height range. Another case of potential smoke self-lofting observed in August 2021, also discussed in Sect. 5, completes our tropospheric self-lofting studies. A summary and concluding remarks are given in Sect. 6.

## 2 Radiative transfer and lofting rate calculations

Biomass-burning smoke particles are assumed in many atmospheric applications to consist of a BC-containing core that is coated with OC substances (Ansmann et al., 2021a). Dahlkötter et al. (2014) found a typical BC core diameter of 130 nm with a mean coating thickness of 105-136 nm in wildfire smoke layers at 10 km height after long-range transport from North America towards Central Europe. Yu et al. (2021) and Torres et al. (2020) assumed that the aged wildfire smoke plumes in the stratosphere contain a BC fraction of around 2.5%. Such a small fraction of BC is the main driver for a strong persistent radiative heating of the smoke layers. The resulting ascent rate, however, depends in a complex way on the vertical gradient of the potential temperature, BC/OC ratio, plume height and depth as well as plume aerosol optical thickness (Boers et al., 2010; de Laat et al., 2012). Significant differences in the tropospheric and stratospheric lofting characteristics exist (Ohneiser et al., 2020, 2021).

The calculation scheme for the self-lofting rate of a smoke layer is shown in Fig. 2 and consists of two independent steps. First, the radiative heating caused by smoke absorption of solar radiation is determined based on radiative transfer calculations. Second, the heating rate is converted to a lofting rate considering the atmospheric profile of potential temperature following Boers et al. (2010). We will explain the scheme in detail in Sect. 2.1-2.4. At TROPOS, several efforts have been conducted to quantify the radiative effects of clouds and aerosols (e.g. Hanschmann et al. (2012); Kanitz et al. (2013); Barlakas et al.

(2020); Witthuhn et al. (2021); Barrientos-Velasco et al. (2022)). This has lead to the development of a Python-based utility library called the TROPOS Cloud and Aerosol Radiative Effects simulator (TCARS). This library has been applied in our study together with the ECRAD radiative transfer scheme (ECWMF Radiation scheme, Hogan and Bozzo (2018); ECRAD (2022)) to determine the radiative heating rate caused by the absorbing smoke aerosol layers. ECRAD generally allows for 3D calculations of radiative transfer in cloudy and cloud-less aerosol-polluted atmospheres (Hogan and Bozzo, 2018). The 3D

calculations were, however, not used here. The key input parameter of interest for our purpose is the vertical profile of the aerosol mass mixing ratio or the particle extinction coefficient (see Fig. 2).

## 2.1 Aerosol profiles as input

Figure 3a shows an example of particle backscatter and extinction profiles at 532 nm as measured with ground-based lidar at Punta Arenas, Chile, in January 2020 (Ohneiser et al., 2022). Australian fire smoke reached heights of 19-23 km by self-

lofting during the long-range transport from Australia to South America. The extinction profile is obtained by multiplying the backscatter coefficient with an extinction-to-backscatter ratio (lidar ratio) of 91 sr. This high lidar ratio is indicative for strongly absorbing smoke particles (Ohneiser et al., 2020, 2022).

For the ECRAD model, Gaussian-shaped profiles in Fig. 3b were then used as input. The green curve is scaled to the same AOT of 0.18 as well as same layer height and layer thickness as observed. The blue curve is scaled to an AOT of 0.22, and the

olive profile to an AOT of 0.1. Layer center height and thickness were varied in the different simulation scenarios.

ECRAD requires mass mixing ratio profiles (see Fig. 2). Therefore, the extinction coefficient $\alpha$ was converted to the mass mixing ratio $m = \alpha c_{\mathrm{v}} \rho_{\mathrm{aer}}/\rho_{\mathrm{air}}$. The volume-to-extinction conversion factor $c_{\mathrm{v}} = 0.13 \cdot 10^{-12}$ Mm is taken from Ansmann et al. (2021a) for aged smoke far away from fire regions. For the smoke particle density $\rho_{\mathrm{aer}}$ we assumed a value of $1.15\,\mathrm{g\,cm^{-3}}$ (Ansmann et al., 2021a). The air density $\rho_{\mathrm{air}}$ is calculated from CAMS (Copernicus Atmosphere Monitoring Service) mete-

orological reanalysis data (CAMS, 2022). The aerosol was then handled as an external mixture of organic carbon particles and black carbon particles with adjustable concentrations. It is known that black carbon can exist in one of several possible mixing states (Jacobson, 2001). The assumption on the selected mixing state of the smoke particles in the simulations is further discussed in Sect. 2.4.

## 2.2 Heating rate calculation

With the aerosol profiles in Fig. 3, aerosol optical parameterization settings in Fig. 4, given Sun position, and the CAMS reanalysis meteorological data file, ECRAD calculates the upwelling '↑' and downwelling '↓' shortwave 'short' and longwave 'long' radiation $F_{\uparrow,\mathrm{short}}$, $F_{\downarrow,\mathrm{long}}$, $F_{\uparrow,\mathrm{long}}$, and $F_{\downarrow,\mathrm{short}}$, respectively, at each pre-defined height and pressure level (see Fig. 2).

Simulations were conducted for 991 pressure bins between the surface and 40 km height. The resulting radiation information is translated into radiative heating information. The radiative flux divergence, meaning the differential change of the radiative

flux between the top and the bottom of a layer defines the radiative heating of the layer. The change of temperature $\mathrm{d}T$ with time $\mathrm{d}t$ is defined by the ratio of the potential temperature $\Theta$ and the temperature $T$, the gravitational acceleration $g_{\mathrm{E}}$, the

specific heat of air $c_p$, and the net change of radiation $dF_{tot}$ between two pressure levels $dp$ with $dT/dt = \frac{g_E}{c_p}(dF_{tot}/dp)$ with $dF_{tot}=dF_{short}+dF_{long} = dF_{\uparrow,short}-dF_{\downarrow,short}+dF_{\uparrow,long}-dF_{\downarrow,long}$.

In order to get a daily-average heating rate in K day$^{-1}$, the radiative heating calculations take a Sun position parameterization into account in terms of time, geographical coordinates, and season (see Fig. 2). To avoid a very detailed consideration of the diurnal cycle of the Sun, we computed the heating and lofting rates every three hours between 0 and 21 UTC and used the mean value of the eight calculations as the representative heating and lofting rate for this specific day. The daily-average heating rate is calculated inside the predefined aerosol layer.

## 2.3 Lofting rate calculations

In the last step of the calculation of the lofting rate $dz$ (see Fig. 2, red part), the heating rate from Sect. 2.2 (now $d\Theta=\frac{\Theta}{T}dT/dt$) is divided by the potential temperature gradient $\Gamma$ as in Eq. 3 in Boers et al. (2010) rewritten as $dz=d\Theta/\Gamma$. Figure 5 shows height profiles of the potential temperature gradient $\Gamma$ at Punta Arenas (-53.17°N, -70.93°E, Chile) for January 2020, Olenek (68.50°N, 112.43°E, Russia) for July-August 2019, and Port Hardy (50.68°N, -127.36°E, USA) for August 2017. In this post-processing step, the CAMS data is not used. Instead, the radiosonde atmospheric data shown in Fig. 5 is applied (**?**). As the potential temperature gradient is strongly influencing the ascent rate, the local radiosonde data is chosen as the most precise data for this application. These profiles were individually applied to the final lofting rate calculations in the respective regions. The profiles look very similar to each other with potential temperature gradients around $5\,\mathrm{K\,km}^{-1}$ in the troposphere, a strong gradient of the potential temperature gradient at the tropopause and a steadily increasing gradient within the stratosphere up to $40\,\mathrm{K\,km}^{-1}$ at 30 km height. Only the height of the tropopause differs between the three locations with around 10 km at Punta Arenas and around 12 km at Port Hardy.

## 2.4 Optical properties of smoke particles

The variety of optical properties of different fire smoke mixtures, and as a consequence, the number of parameterizations used in simulations is large. In this study, an external mixture of black carbon and brown carbon with adjustable black carbon fraction is prioritized instead of an aggregate consisting of both aerosol types. Jacobson (2001) found that an external mixture of black carbon and other aerosol types potentially leads to an underestimation of radiative forcing compared to an internal mixture. Shiraiwa et al. (2008) estimated that internal mixing enhances the BC absorption by a factor of 1.5–1.6 compared to external mixing. For smaller BC cores (or fractal agglomerates) consideration of the BC and brown carbon as an external mixture leads to relatively small errors in the particle single scattering albedo of <0.03 (Lack and Cappa, 2010). Lesins et al. (2002) found, however, that the difference in extinction, single scattering albedo, and asymmetry parameter, between an internal mixture and external mixture of black carbon and ammonium sulfate can be >25% for the dry case and >50% for the wet case for typical mass mixing ratios. Liu and Mishchenko (2007), on the other hand, show that the optical cross-sections of externally mixed aggregates is up to 20% larger than of multi-component aggregates. Typically, the organic carbon fraction was set around 97.5% and the black carbon fraction around 2.5% (Yu et al., 2019, 2021; Torres et al., 2020). More work on the relationship between smoke chemical and microphyscial properties and resulting optical properties is required.

The optical properties in terms of single scattering albedo $SSA$ and asymmetry factor $g$ of the hydrophobic organic matter as well as three different black carbon parameterizations are summarized in Fig. 4. $SSA$ describes the ratio of scattering efficiency to total extinction efficiency and $g$ describes the mean cosine of the scattering angle when integrating over the complete scattering phase function. The organic carbon parameterization (OPAC: Optical Properties of Aerosols and Clouds) (Hess et al., 1998) shows a high single scattering albedo as well as high asymmetry factors at all wavelengths between 200 nm and 3400 nm compared to the three different shown black carbon parameterizations. Even though the aerosol mixture usually consists of only $\approx$3% BC aerosol, the low BC $SSA$ and $g$ values widely determine the total absorbing characteristics. Three BC parameterizations are shown for comparison in Fig. 4. Parameterization 1: OPAC (Hess et al., 1998), parameterization 2: Bond and Bergstrom (2006), parameterization 3: Stier et al. (2007). All the three parameterizations are quite similar regarding their optical properties, however, parameterization 1 shows a slightly enhanced asymmetry factor. In the following calculations all parameterizations are used, however, if not stated differently in the text, the OPAC parameterization is applied.

## 3 Sensitivity of self-lofting rate simulations

The following section focuses on the simulations of lofting rates for different smoke plume characteristics and aerosol scenarios in the troposphere and stratosphere. Different BC parameterizations and BC/OC fractions, AOT, and layer thicknesses are considered. Smoke lofting will be simulated, e.g., in the case of Canadian wildfires and Kuwait oil fires. Furthermore, the influence of different atmospheric background situations regarding cloudiness or relative humidity on the lofting rate is investigated.

For an overview, Fig. 6 shows a map with all discussed major wildfire events. The Canadian fires in 2017 and the Australian fires in 2019/2020 were accompanied by pyroCb convection. The stratospheric wildfire smoke observed over the Arctic (Ohneiser et al., 2021) probably originated from record-breaking fires over central-eastern Siberia, north and northeast of Lake Baikal in July and August 2019 (more details are given in Sect. 5). Surprisingly, pyroCb activity, usually responsible for smoke lofting up to the UTLS region, was absent over the main fire places during the strongest fires from mid-July to mid-August 2019. In the absence of pyroCb activity, the only left pathway is, to our best knowledge, smoke self-lofting. As will be shown below, self-lofting from typical injection heights (caused by the ascent of the rather hot air over the fires) of 2-6 km (Amiridis et al., 2010) to the tropopause takes several days, during which the aerosol particles can complete the aging process and as a result get compact and spherical in shape. As a consequence, the measurable particle depolarization ratios get very low. Perfect spheres produce no light depolarization.

The Kuwait oil fires in 1991, indicated in Fig. 6, were simulated (for comparison) as the smoke from burning oil fields has a much higher BC fraction (up to almost 50% shortly after emission) (Hobbs and Radke, 1992). Fresh and aged biomass burning smoke show BC fractions of 5-30% (Mereuță et al., 2022) and around 2-3% (Yu et al., 2019; Torres et al., 2020), respectively. The predominant burning material consists of fir, aspen, and cedar trees in the case of Canadian forest fires, and spruce, pine, and larch trees in the case of Siberian fires. In Australia, however, the oil containing eucalyptus trees might lead to more BC containing absorptive smoke aerosol compared to the Canadian and Siberian smoke layers (Ohneiser et al., 2022).

In Fig. 6, we distinguish pryo-Cb-related lofting (1 h stands for a short tropospheric residence time, too short for particle aging) and self-lofting smoke events (3-7 days stand for a time period long enough to complete particle aging). Smoke lofting into the UTLS height region via the pyroCb pathway is a well accepted and well documented lofting process. All observed pyroCb-related stratospheric smoke plumes, without any exception, show a high particle linear depolarization ratio up to 0.2 at 532 nm (Haarig et al., 2018; Ohneiser et al., 2020). This light-depolarization information is the basic criteria in the CALIOP aerosol typing scheme (Kim et al., 2018; Ansmann et al., 2021a; Knepp et al., 2022) to identify stratospheric wildfire smoke (in the absence of volcanic ash producing strong depolarization ratios as well). The high depolarization ratio is caused by irregularly shaped carbonaceous particles (fractal-like aggregates). The particle shape obviously remained widely unchanged after emission and during the short lofting process from the lower troposphere to the tropopause within pyroCbs. The tropospheric residence time is too short to initiate significant particle aging (condensation of gases on the emitted smoke particles, photo-reaction and chemical processes, coagulation of particles) so that a compact core-shell structure (morphology) of the particles can not develop.

An indication for the complete termination of the aging process is a very low particle depolarization ratio of <0.03 indicating spherical (liquid, semi-solid, or glassy) smoke particles. Particle aging in the troposphere and the development of spherical particle structures require at least two days (Fiebig et al., 2003; Ansmann et al., 2021a), provided the environmental conditions are favorable (high relative humidity, high amount of condensable gases). The aging process may take weeks in a dry tropospheric air mass or even months in the dry stratosphere (Baars et al., 2019; Ohneiser et al., 2022). As long as the particle depolarization ratio is significantly enhanced (>0.05) the aging process is not finalized and some deviations from the ideal spherical shape remained. After completion of the aging process, most of the smoke particles consist of a BC-containing core and a spherical OC-rich shell (coating). Since we observed rather low particle depolarization ratios in the smoke layer in the summer of 2019, we assume that well-aged smoke particles polluted the UTLS height range (Ansmann et al., 2021b). The spectrally resolved extinction-to-backscatter ratios (lidar ratios) help to distinguish volcanic sulfate aerosol and self-lofted smoke particles (Haarig et al., 2018; Ohneiser et al., 2020, 2022). A compact overview of the microphysical, chemical, optical and cloud-relevant properties of tropospheric and stratospheric smoke and changes of these properties during the aging process can be found in Ansmann et al. (2021a, 2022).

### 3.1 Impact of smoke layer AOT and layer height on heating and lofting rates

The AOT widely determines how much shortwave radiation can be absorbed and how strong an aerosol layer can heat up. In Fig. 7, a 2 km thick smoke layer is simulated. This smoke layer was parameterized in the way described in Fig. 3b, however, now for a 2 km instead of a 4 km thick layer. In the simulation, the center height of the smoke layer was stepwise increased by 1 km between 1 and 28 km height, i.e., calculations were performed for layers from 0-2 km to 27-29 km height. The bottom and top heights of the layer are defined by the extinction profile. The extinction coefficient starts to increase with height above the bottom height and is again height-independent (does not further decrease with height) above the top height. As in Fig. 3b the AOT was scaled as indicated in the legend. Four different AOTs are assumed. In this way the difference between heating and lofting conditions in the troposphere and stratosphere become visible. As can be seen, the heating rate in Fig. 7a increases

exponentially with height and approximately linearly with AOT. Every 5 km the heating rate is doubled for the same AOT. Less dense air can be heated up much more efficiently. A 2 km thick aerosol layer with an AOT of 1.5 at 1 km height is heated with 3.5 K day$^{-1}$ while the same layer at 25 km height would hypothetically heat up as much as 200 K day$^{-1}$.

The heating rate in Fig. 7a is transferred into a lofting rate in Fig. 7b by using the gradient of the potential temperature in Fig. 5 (using the Punta Arenas January 2020 data). Generally, the lofting rate increases with increasing height in the troposphere and in the stratosphere. However, there is a pronounced lofting inhibition (minimum) at the tropopause around 12 km height. Because this aspect is of key importance in the discussion of CALIOP observations and identification of smoke self-lofting in Sect. 5, it is worth to explain this specific feature in the lofting rate profile in more detail. The heating rate is relatively low in the troposphere and rather strong in the stratosphere as shown in Fig. 7a. Since the gradient of the potential temperature is also low in the troposphere (see Fig. 5) the ratio of the heating rate to the potential temperature gradient is still positive. In other words, smoke lofting (defined in Step 2 in Fig. 2) is possible even in the troposphere at weak heating rates. However, in the upper troposphere (from 3-4 km below the tropopause to the tropopause at 10.5 km in the case of our simulations, based on the Punta Arenas radiosonde profile in Fig. 5), the gradient of the potential temperature increases fastly with height, from about 3.5 K km$^{-1}$ at 7 km to 18 K km$^{-1}$ at 11 km height, so that the denominator grows faster than the numerator in the equation in Step 2 in Fig. 2. As a consequence, the lofting speed decreases and reaches a minimum close to tropopause. Above the minimum, in the stratosphere, the continuously increasing heating rate dominates finally the lofting rate at all heights, disregarding the increasing strength of the potential temperature gradient in the stratosphere (see Fig. 5). Thus, the tropopause is a clear barrier for self-lofting processes. Smoke ascending from the smoke injection height (in the lower to middle free troposphere) towards the tropopause will have to accumulate below the tropopause.

As can be seen in Fig. 7b, the strong increase of the gradient of the potential temperature at the tropopause leads to a reduction in lofting speed by more than a factor of 2 compared to the lofting velocity at 8 km height. Within the stratosphere ascent rates increase again. For an aerosol optical thickness of 1.5 the lofting rate would reach 3 km day$^{-1}$ at around 15 km height.

The minimum in the lofting rate profile at the tropopause is an important feature of the entire self-lofting process. As a consequence, lofted aerosol will accumulate below and around the tropopause during situations with a steady upward flow of smoke particles towards the tropopause. The formation of such a tropopause layer should be observable with the space-borne CALIOP lidar instrument when such an upward transport of smoke takes place over several days or even a week and more. This aspect is further discussed in Sect. 5 based on CALIOP observations. Such a layer, predicted by the simulations, was found around the tropopause.

## 3.2 Impact of smoke absorption characteristics on smoke lofting

Figure 8 focuses on the difference in the smoke absorption characteristics. The three different black carbon parameterizations discussed in Sect. 2.4 and shown in Fig. 4 are considered. The simulations were performed with Gaussian-shaped 2-km thick profiles (vertical profile shape as in Fig. 3b, but for a 2-km deep layer) in 1 km steps up to 20 km height. Figure 8a generally

shows an exponential increase of the heating rate with height. Differences in the heating rates by using the three different parameterizations become visible in the stratosphere.

Regarding the lofting rates, Redfern et al. (2021) showed that lofting also depends on the relative humidity as well as wind speed and wind shear. The atmospheric profiles of potential temperature gradient and the relative humidity at Punta Arenas measured on the 26 Jan 2020 are shown in Fig. 8b. The gradient of the potential temperature is around 5 K in the troposphere and around 20 K in the stratosphere up to 20 km height. In between, there is a strong change of the temperature gradient at the tropopause. The relative humidity shows an almost saturated moist layer at 1-4 km height, a dry layer between 5 and 7 km, and a slightly increased relative humidity of 25% between 7 and 9 km. In the stratosphere 0-10% relative humidity was found for that day.

The resulting lofting rates in Fig. 8c show an increase of $0.5\,\mathrm{km\,d^{-1}}$ at 1 km height to $\approx 4\,\mathrm{km\,d^{-1}}$ below the tropopause for all three BC parameterizations. At the tropopause lofting rates decrease to less than $2\,\mathrm{km\,d^{-1}}$ due to the strong increase of the potential temperature gradient. Again, higher up in the stratosphere lofting rates increase to $4\text{-}6\,\mathrm{km\,d^{-1}}$. Comparing the three different parameterizations in Fig. 8d yields differences in the lofting rates of less than $0.6\,\mathrm{km\,d^{-1}}$. Relative uncertainties in the lofting rate simulations are smaller than 20%.

The differences between 'BC3-BC2' as well as 'BC3-BC1' shows the obvious impact of the relatve humidity on the lofting rates. The impact is weak in the case of 'BC2-BC1'. The local maxima in the relative humidity at 2.5 km and 8 km height coincide with the local maxima in the differences of the parameterizations. This behavior reflects differences in the water up take efficiency (of the smoke particles) in the different parameterizations. Different hygroscopic properties lead to slight changes in the chemical composition of the coating of the smoke particles and thus of the light-absorption properties. Smoke particles with a liquid coating may focus solar radiation to the core of the particle and increase the absorption coefficient by up to a factor of 2 (Liu and Mishchenko, 2018) which would increase the heating rate and hence lofting rate of the smoke layers. Parameterization 'BC2' is systematically slightly increased compared to 'BC1' between 0.1 and $0.5\,\mathrm{km\,d^{-1}}$ with increasing humidity, potentially linked to different absorption characteristics. Nevertheless, all the differences are within a small range and it is reasonable to use BC parameterization number 1 (Hess et al., 1998).

### 3.3 Impact of low level clouds on smoke lofting

An additional parameter that influences the lofting rate of an aerosol layer is the fraction of clouds that are located below an aerosol layer. Solar radiation is efficiently reflected by clouds. Therefore, large fractions of the radiation are passing twice through the aerosol layer and increase the heating rate significantly. Typical albedo values of the Earth's surface are around 0.3 whereas low level clouds can have an albedo of up to more than 0.9.

Figure 9 provides insight into the impact of low level clouds on the lofting of lofted layers. A smoke layer, initially centered at 4 km height after injection, is simulated. Gaussian-shaped profiles (in terms of light extinction profile) were simulated (with 2 km or 4 km thickness) and scaled to an AOT of 2. Again, the daily average heating rates (considering the Sun position every three hours from 0 UTC to 21 UTC) were used to calculate the heating rates and lofting rates. The radiation that passes through an aerosol layer during an overcast cloud situation was set to 1.7 times the initial radiation for the overcast 'c' scenario. The

black and red curves represent the overcast situation and show a much larger lofting rate compared to the respective orange
curve (orange curve vs red curve) that represents clear sky conditions. In an overcast situation, it takes 184 hours (7.5 days)
for an aerosol layer with an AOT of constantly 2 and a layer thickness of 4 km to ascend from 4 km height to 16 km whereas
it would take 313 hours (13 days) in a clear sky situation. Thus, a cloud layer below an absorbing aerosol layer increases the
lofting rate by around 70%.

Analogously, also the layer thickness influences the lofting rate. The 4 km thick aerosol layer (red curve) shows a much
lower lofting rate compared to the case with a 2 km layer geometrical thickness (black curve) and same AOT (and thus a factor
of 2 higher particle extinction coefficients). During an overcast situation a 2 km thick layer would ascend from 4 to 16 km
within 89 hours while a 4 km thick layer would need 184 hours.

## 3.4 Impact of height-dependent heating and lofting on ascending layer structures

It was shown that smoke layers have a significant lofting potential in the troposphere and the stratosphere. In the troposphere,
the heating rates are comparably low but also the potential temperature gradient is low. In the stratosphere, the heating rates
are much larger for the same aerosol optical thickness, however, the limiting factor, the potential temperature gradient, is also
strongly enhanced. All layers have in common that they encounter meteorological stresses like wind shear and turbulence that
may destroy coherent structures of an aerosol layer and the ability of a layer to efficiently ascend. Also radiative effects cause
an additional stress on the aerosol layer structures.

Figure 10a shows a rectangular-shaped smoke particle extinction profile in the troposphere between 4-6 km with an average
extinction coefficient of $600 \, \mathrm{Mm}^{-1}$. In Fig. 10f, a stratospheric smoke profile between 24-26 km with an average extinction
coefficient of $80 \, \mathrm{Mm}^{-1}$ is simulated. The respective profiles of the heating rate in Fig. 10b and g show differential heating
of the aerosol layer. The layer top is much more heated than the layer bottom. The tropospheric heating rates show values
around $6 \, \mathrm{K \, day}^{-1}$ at the layer bottom and $9 \, \mathrm{K \, day}^{-1}$ at the layer top, whereas the stratospheric heating rates are $35 \, \mathrm{K \, day}^{-1}$
at layer base and $50 \, \mathrm{K \, day}^{-1}$ at layer top. In relative numbers, the heating of the layer top of the 2 km thick layer is 50%
higher than heating of the layer base. However, lofting strongly depends on the potential temperature gradient as well. In the
troposphere (Fig. 10c), the potential temperature gradient slightly decreased from $4.2 \, \mathrm{K \, km}^{-1}$ at 4 km height to $3.4 \, \mathrm{K \, km}^{-1}$
at 6 km height. In the stratosphere (Fig. 10h), the potential temperature gradient strongly increased from $27 \, \mathrm{K \, km}^{-1}$ at 24 km
height to $31 \, \mathrm{K \, km}^{-1}$ at 26 km height. These different meteorological conditions result in different shapes of the lofting rate
profiles in the troposphere and the stratosphere. In the troposphere (Fig. 10d), the lofting rate is almost twice as large at the
layer top compared to the one at layer base, whereas in the stratosphere (Fig. 10i) the lofting rate is quite constant throughout
the aerosol layer. This means that the layer top is lofted more efficiently compared to the layer base in both cases. However, this
effect is much more pronounced in the troposphere. The resulting new aerosol extinction profiles in Fig. 10e for the troposphere
and in Fig. 10j for the stratosphere show the structure of the aerosol layer after one day of ascent. The tropospheric aerosol
layer is now found between 5.0 km-7.8 km height. The layer depth increased by 50%. As a consequence, the layer mean particle
extinction coefficient is reduced by 30%. In contrast, the boundaries of the stratospheric layer are found between 25.0 km and
27.2 km. The layer thickness is only increased by 10% and the layer mean extinction coefficient decreased by only 10%.

The simulations illustrate why stratospheric layers are able to show coherent structures over long time periods compared to smoke structures in the troposphere. Tropospheric smoke layers are stretched more in vertical direction so that additional wind shear and turbulence can easily destroy coherent structures in the less stratified troposphere. Furthermore, the related stronger decrease of the layer mean extinction coefficient (of tropospheric layers) lead to smaller heating rates at the next time step and therefore smaller lofting rates on the next day. All these reasons and influences must be kept in consideration when comparing CALIOP observations of tropospheric and stratospheric ascending smoke layers (and structures) with respective simulations.

One should emphasize here that our goal is to study the principle capability of self-lofting to transport smoke up to the tropopause, and not to precisely simulate 3D air motions as a result of the absorptive heating and associated buoyancy production. To fully account for the coupling of smoke occurrence, aerosol-radiation interaction, and resulting dynamical processes, a 3D chemistry-climate model must be used (Das et al., 2021). By using our simulation scheme it is possible to realistically model the ascent behavior in the stratosphere, as will be shown in Sect. 4.1 . However, regarding the troposphere it is expected that convection of differently heated air parcels (for given realistic 3D fields of vertically and horizontally inhomogeneous aerosol scattering and absorption coefficients) leads in most situations to turbulent aerosol features and structures. As a consequence, it seems to be almost impossible to detect self-lofting of smoke layers in the (turbulent) troposphere, e.g., by means of daily CALIOP snapshot-like observations of smoke layering downwind of strong fires. Therefore, in the case of tropospheric CALIOP smoke observations we only discuss the observed smoke layering features, and use our simulation results as background information to identify potential self-lofting signatures. This discussion is presented in Sect. 5.

## 3.5 Impact of a steadily decreasing smoke layer AOT on the ascent behavior

Figure 11 shows the temporal evolution of the layer center of a 2.5 km thick aerosol layer (with Gaussian-shaped aerosol profile) in dependence on the AOT. The initial layer center was at 3 km height (at day 0). The profiles were scaled to an AOT of 1, 2, and 3, as shown in the legend, analogously as compared to Fig. 3b. Three different BC fractions are considered. Again, daily-average heating rates are used and respective daily-average lofting rates are calculated. In reality, the smoke layers diverge with time so that the AOT decreases. In the scenarios in Fig. 11, the AOT decreases by 15% from day to day. On each next day, the new smoke layer center in terms of extinction is calculated from the layer center height of the last day plus the 24 h mean lofting rate. The layer thickness is always 2.5 km and the AOT is 15% less compared to the day before. All curves in Fig. 11 indicate ascending layers that accelerate in the higher troposphere although AOT is decreasing by 15% per day. At the tropopause all aerosol layers ascend slowly. The higher the BC fraction, the higher is the finally reached altitude. After 14 days of lofting, a smoke layer with a 2.5% BC content would typically be found 1-3 km below the height of an aerosol layer with a 3.5% BC content. A higher BC fraction is directly related to a larger ascent rate, however, there are too many free parameters that influence the ascent rate which makes it hard to determine the BC fraction from model simulations when comparing to an observed lofting rate.

We added a simulation of an ascending aerosol layer with a high BC fraction of 15%. In 1991, extreme oil field fires in Kuwait released large amounts of black aerosol. Aircraft observations in May-June 1991 showed very little soot (4% by mass) in the white smoke, black smoke contained 20 to 25% soot (BC fraction), and the blackest smoke contained up to 48% soot by

mass (Hobbs and Radke, 1992). The optical depth of the plumes for visible radiation at about 100 km from the fires was 2 to 3. According to satellite observations in March 1991 most plumes were below 3 km height within 50 km around the sources (Limaye et al., 1991). In a distance of 2000 km, several smoke plumes reached, however, 6-7 km height, after about 18-24 h travel time with wind speeds of 25 m s$^{-1}$.

As can be seen, this aerosol layer with 15% BC content ascends fast, by about 3-5.5 km during the first 48 hours (blue dotted curve in Fig. 11). This fits very well to the ascent of the Kuwait oil smoke plumes of about 3-4 km within one day. The wildfire smoke AOT typically needs to be larger than 2 in cases with 2%-3.5% BC fraction to reach the tropopause level.

Our simulations show that it is in principle possible to loft an aerosol layer to the tropopause in the absence of any pyroCb convection. As the lofting process is quite efficient in the upper troposphere, even in the case of moderate pyroCb development with cloud tops reaching 8-10 km height only, smoke layers can easily be lofted higher up into the stratosphere.

In addition to optically thick smoke layers that ascend from the troposphere to the stratosphere, Figure 11 provides an estimate regarding the minimum AOT that is required to further loft the aerosol plume (and thus to dominate over downward motion by sedimentation). The red lines represent a smoke layer in the stratosphere at 13 km height (day 0) with an initial AOT of 0.5 and 0.1 at 532 nm. Every day, the AOT reduces by 15%. As can be seen, for an AOT of 0.5 the ascent is about 4 km within two weeks. For an initial AOT of 0.1 the height gain is about 500 m within 14 days or 36 m per day. Further simulations with an AOT of 0.05 and 0.01 yield ascent rates of 40 m and 8 m per day. However, it must be noted that these values are representative for a 2.5 km thick smoke layer at 13 km. Smoke layers with a layer thickness of 10 km or 15 km, as it was observed for the Siberian and Australian wildfire smoke would be lofted around 4 m and 1 m per day, respectively, for an AOT of 0.01. During the autumn and winter months this ascent rate is even lower. Therefore, for such a small AOT of 0.05 and lower, the sedimentation of the smoke particles in the stratosphere would dominate the self-lofting process. After the strong Pinatubo volcanic eruption in June 1991, we observed an average descent rate of the stratospheric Pinatubo layer center of about 5-6 m per day from 1992-1994 (4 km in 750 days) (Ansmann et al., 1997).

## 3.6   Summary of simulation uncertainties

The self-lofting efficiency for given smoke layers depends on many factors. The most important parameters are summarized in Table 1. The aerosol optical thickness plays a key role in lofting aerosol layers. Usually an AOT of >0.5 in the stratosphere or >2 in the troposphere is necessary in order to significantly loft smoke plumes by a few kilometers. A doubling in AOT means a doubling in heating rate and hence lofting rate (thus the impact is described as linear in Table 1). Also the layer thickness sensitively influences the lofting rate. If the same AOT is distributed over a larger vertical column, the layer mean particle extinction coefficient and the corresponding heating rate is decreased and therefore the lofting rate is decreased.

Another indirect effect (caused by vertical stretching) is the increasing impact of vertical wind shear with increasing vertical extent of the smoke plumes. Wind shear can effectively destroy the aerosol layer structures and can sigificantly reduce the lifetime of the layer. The height of a smoke plume itself is relevant for the lofting rate. The higher the smoke plume center height, the higher is the heating rate. As discussed in Sect. 3.1 the heating rate is approximately doubled at every 5 km height step. Thus this impact is described as nonlinear in Table 1.

The lowest troposphere and the tropopause are height regions which allow only for comparably small lofting rates. Further-more, it is important to note that the initial smoke layer height (meaning the injection height) needs to be precisely known for long-lived smoke plumes. Especially in the stratosphere the underestimation/overestimation of the injection height leads to an underestimated/overestimated lofting rate which defines the new height of the next iteration step. The wrong estimation will sum up in every step. As will be shown in Sect. 4.3 for the Australian wildfire smoke an injection at 1 km higher altitude would lead to an additional lofting of 5 km. As discussed, the lofting rate is strongly dependent on the layer height itself, so the error will not linearly sum up but exponentially to some extent.

The BC/OC ratio is usually around 2.5% which is the main driver for the self-lofting process. The higher the BC/OC ratio is, the larger is the lofting rate. Not only the BC/OC ratio but also the parameterized optical properties of the BC aerosol have an impact on the modeled lofting rate. The single scattering albedo as well as the asymmetry factors can vary slightly depending on the concrete case. Depending on the forest type, as well as the fire type (discussed in Ohneiser et al. (2020)) each fire aerosol event can evolve different wildfire aerosol particles in terms of absorbing characteristics of the BC particles (see Sect. 3). Slight changes in SSA and g can already create 10% of uncertainty in the lofting rate calculations.

Also the atmospheric situation itself can significantly contribute to the lofting of smoke layers. Clouds above the aerosol layer will decrease the amount of radiation and hence the lofting rate of smoke particles. Smoke above a reflecting cloud layer will lead to an almost doubled lofting rate as the shortwave radiation is reflected by the clouds and enters the smoke layer twice. This process is especially important in the stratosphere.

Increased atmospheric relative humidity can lead to a liquid coating of the BC/OC aerosol mixtures, especially in the tro-posphere. This has also slight influence on the absorption of shortwave radiation, depending on the black carbon type. Smoke particles in the troposphere with a liquid coating may focus solar radiation to the core of the particle.

All these uncertainties do not cancel out each other. Uncertainties can sum up with travel time (time for lofting). The calculated layer heights can signifIcantly deviate from observations after 10-20 days. Slight changes in the initial smoke plume characteristics can lead to significant differences in the heights at which a strong smoke plume can be found after a few weeks. 1D simulations of smoke ascent rates cannot be used to predict the ascent behavior of convective smoke-laden air parcels in the turbulent troposphere. However, the next section demonstrates that the presented ECRAD-based simulation tool is a powerful instrument to explain self-lofting in the stratosphere as observed with the space-borne CALIOP lidar.

## 4 Comparison of stratospheric CALIOP smoke observations with ECRAD simulations

### 4.1 Observational data sources

To check the usefulness and applicability of the developed ECRAD-based self-lofting simulation scheme we compared smoke self-lofting events as observed with the space-borne CALIOP lidar with respective simulation results. In Sect. 4.2 and 4.3, two cases of stratospheric smoke layers (Canadian smoke in 2017, Australian smoke in 2020) are discussed. The general strategy was to determine (or estimate) the geometrical properties of the detected smoke layers (layer depth, center, top and base heights) as well as the AOT values, from the CALIOP observations on a daily basis. These data were then used as input

in the simulations. The lofting rates as observed with CALIOP were finally compared with simulated lofting rates. As a free parameter, we adjusted the BC fraction in the simulations to optimize the match between simulated and observed ascending smoke features.

CALIOP quicklooks (colored height-time displays of the attenuated backscatter coefficient at 532 nm) were downloaded CALIPSO (2022) and displayed in time series over days to weeks (see for example Fig. 12a-g). Layer bottom and top heights were determined by visual inspection of the backscatter features. With the layer base and top height information, the height of the layer center is obtained. Furthermore, AOT was estimated from the total (Rayleigh plus particle) attenuated backscatter profiles. A small stratospheric Rayleigh AOT contribution of the order of 0.005 at 532 nm was ignored. The AOT is calculated from the layer mean attenuated backscatter coefficient multiplied by a typical smoke lidar ratio of 65 sr and 91 sr for the Canadian and Australian wildfire smoke (Baars et al., 2019; Ohneiser et al., 2021, 2022), respectively, and finally multiplied by the layer geometrical depth. This AOT value may underestimate the true AOT by 50% when using attenuated backscatter information instead of the true particle backscatter coefficient profile and by ignoring multiple scattering effects in the retrieval. Therefore, we considered also AOT values multiplied by a factor of 1.5 as input in subsequent simulations.

In addition, we used AOT observations with MODIS (Moderate Resolution Imaging Spectroradiometer) aboard the Terra and Aqua satellites (MODIS, 2022) in the case of Australian smoke scenarios as an independent approach to obtain smoke AOT information. MODIS AOT values are more reliable but contain information about the entire vertical column including contributions of aerosol particles in the lower troposphere of the order of 0.02-0.06 over the southern Pacific and Southern Ocean. We used a window of 9×9 pixels (around a central pixel) and removed all cloud contaminated pixels. Then all remaining valid AOT pixel were averaged.

## 4.2 Ascending Canadian wildfire smoke

Khaykin et al. (2018), Baars et al. (2019), Torres et al. (2020), Das et al. (2021), and Lestrelin et al. (2021) discussed cases with ascending stratospheric Canadian wildfire smoke detected in the summer of 2017. Figure 12a-g shows CALIOP measurements of an ascending Canadian smoke plume between 14 August 2017 and 4 September 2017. In Fig. 12 (panel h), the CALIOP-derived AOT observations are shown. In addition, the parameterization of the AOT as used as input in the ECRAD simulations is presented. The simulation results regarding self-lofting are shown in Fig. 12 (panel i) together with the smoke lofting behavior as observed with CALIOP. A BC fraction of 1.5% and 2.5% was assumed for comparison in the simulations. All simulations were performed for cloud-free conditions, i.e, in the absence of sun-light-reflecting clouds below the smoke layers.

As can be seen, within 21 days the smoke layer ascended by self-lofting processes by 6 km. In the beginning, the smoke plume height increased from around 12 km on 14 August 2017 to around 16 km on 19 August 2017. The lofting rate was thus almost 1 km per day. The particle attenuated backscatter coefficients in the layer slowly decreased with time from values $>5 \, \mathrm{Mm^{-1} \, sr^{-1}}$ to values around $1 \, \mathrm{Mm^{-1} \, sr^{-1}}$. With decreasing backscatter (and light-absorption) the height gain decreased as well. Within the following 16 days the height of the smoke plume increased from 16 km on 18 August 2017 to 19 km on 4 September 2017.

## 4.3 Ascending Australian wildfire smoke

Ohneiser et al. (2020, 2022), Kablick et al. (2020), Khaykin et al. (2020), and Allen et al. (2020) studied the ascent behavior of the Australian smoke layer in 2020. Here, we deepen this discussion. In Fig. 13a-j, CALIOP observations of Australian
fire smoke are shown. From 31 December 2019 to 5 January 2020, extremely intense fires over large areas in southeastern Australia in combination with the evolution of more than 40 individual pyrocumulonimbus storms (Peterson et al., 2021; Ohneiser et al., 2022) caused the injection of record-breaking amounts of fire smoke into the UTLS region. Extended smoke fields were detected at 14 km height on 31 December 2019 (Fig. 13a). During the first 12 days after injection, the smoke layer ascended and reached the 20 km level on 11 January 2020. Similarly to the Canadian stratospheric fire smoke, the aerosol
backscatter decreased from values $>5\,\mathrm{Mm}^{-1}\,\mathrm{sr}^{-1}$ in the beginning of January to values $<0.5\,\mathrm{Mm}^{-1}\,\mathrm{sr}^{-1}$ mid of February. In contrast to the Canadian fire smoke plume, the specific Australian smoke plume (considered in this study) remained compact and as a consequence, ascended more efficiently over weeks. During the following 30 days the smoke layer reached almost the 30 km height level.

The stratospheric AOT values in Fig. 13 (panel k) is calculated by using CALIOP and MODIS data. The CALIOP AOT data
were taken from Kablick et al. (2020). The parameterization is mostly based on the high AOT values (see the scattered AOT data before 10 January 2020) clearly indicating the presence of smoke. In the CALIOP extinction computation, a lidar ratio of 65-70 sr is usually applied to convert backscatter into extinction values. However, the lidar ratio of Australian smoke was much higher (Ohneiser et al., 2020, 2022). Therefore, we also used a parameterization with a factor of 1.5 higher AOT values.

As mentioned, we used MODIS observations as an independent approach to obtain an AOT time series (for 550 nm wave-
length) along the smoke travel pathway. We subtracted a minor AOT contribution of 0.03-0.05 for tropospheric aerosols. As can be seen in Fig. 13 (panel k) the MODIS-derived AOTs are around a factor of 3-4 higher than the CALIOP AOT values until 16 January 2020. Obviously, the tropospheric AOT contribution was much higher than assumed. Therefore, MODIS-AOT-based simulations started on 17 January 2020. The CALIOP and MODIS AOT parameterizations (dashed blue and solid blue and orange lines) were then used in the model calculations presented in Fig. 13 (panel l).

Again, all simulations were performed for cloud-free conditions. A good match between the simulated and observed ascent of the smoke layer is obtained for the CALIOP AOT parameterization considering more realistic lidar ratios in the backscatter to extinction conversion (1.5*CALIOP AOT). The used BC fraction of 2.5% is in agreement with studies of Yu et al. (2019, 2021) and Torres et al. (2020) who also concluded that the BC fraction must be around 2.5% to explain the observed smoke lofting of stratospheric wildfire smoke layers. By using the MODIS AOT simulations, starting on 17 January 2020 (orange dashed
curve), the self-lofting process is slightly overestimated by the model even in the case with a reduced BC fraction of 1.5%. The agreement between simulated and observed ascent rates is improved when the starting height of the smoke layer is lowered by 1 km. The sensitive impact of the smoke injection height was discussed by Heinold et al. (2022).

The decreasing lofting rate is well captured by the simulations (when using the 1.5*CALIOP AOT parameterization) and the final height at 30 km after 2 months is in good agreement with the observations. This shows that our ECRAD-based simulation

scheme can reproduce the observed lofting rates of the smoke plumes. However, there are a lot of parameters to be set that can sensitively influence the simulations as was discussed in Sect. 3.

## 5 Self-lofting signatures in the upper troposphere over Siberia and the Arctic Ocean in 2019 and 2021?

As pointed out in Sect. 3.4, coherent lofting of horizontally extended smoke fields over days and weeks, as observable in the stratosphere, may only be found in the troposphere under very certain conditions. Turbulent, convective, incoherent motions of
ascending smoke-filled air parcels prohibit a direct detection of self-lofting signatures that can be detected in CALIOP observations (by comparing smoke fields from day to day, over several days). Thus, the strategy of comparing CALIOP observation with our 1D simulations, as successfully applied to stratospheric smoke observation, is not used in the following discussion. We therefore applied an alternative concept to identify signatures of smoke self-lofting as will be explained below.

The focus in this section is on the record-breaking wildfire smoke outbreaks over central Siberia in the summer of 2019.
Ohneiser et al. (2021) already presented a detailed analysis of the smoke conditions over the Siberian burning areas, mainly north of Lake Baikal, and discussed the potential contribution of self lofting to the formation of a smoke dominated UTLS aerosol layer over Siberia in July 2019. The goal in this study is now to detect further hints on the impact of smoke self-lofting processes in the CALIOP observations over Siberia and the main outflow region, the Arctic Ocean, in July and August 2019 and later on. Only weak pyroCb activity was noticed over Siberia during the summer of 2019 (Knepp et al., 2022).

The 2019 fire season in central Siberia began most probably around 15 June 2019 (Sorenson et al., 2022; Xian et al., 2022). A first intense fire period was noticed on 4-5 July 2019 (Johnson et al., 2021). The most intense fire storm then started on 19 July and lasted almost one months, until 14 August 2019 (Johnson et al., 2021). During this time, rather intense wildfires occurred over central Siberia (56°-63°N, 100°-115°E, see Fig. 6 and Fig. 3 in Ohneiser et al. (2021)). At the end of July 2019, the 550 nm AOT reached record-breaking values of more than 2.5 over an area as large as 500×1000 km for several days
(Ohneiser et al., 2021). The August 2019 mean 550 nm AOT over the analyzed Siberian burning area, north of Lake Baikal, was the highest monthly mean value within the analyzed 20 year MODIS data set (2000-2019) (Ohneiser et al., 2021). Very low wind speeds and weak horizontal air mass transport (stagnant conditions) provided favorable conditions for the accumulation of smoke, the evolution of high AOTs on a regional scale, and thus of self-lofting of smoke-containing air masses.

From the articles of Johnson et al. (2021), Xian et al. (2022), and Sorenson et al. (2022) we conclude that the Siberian smoke
was to a large part transported to the Arctic. Shortly after the onset of the Siberian fire season, Xian et al. (2022) reported an almost monotonic increase in the 550 nm AOT over the Arctic. Xian et al. (2022) combined AERONET (Aerosol Robotic Network) AOT observations with NAAPS (The Navy Aerosol Analysis and Prediction System) AOT reanalysis products. The area mean 550 nm AOT over the Arctic region from 70°-90°N increased from daily mean values around 0.05 on 20-22 June 2019 over 0.2 on 5 August 2019 to more than 0.4 on 11 August 2019. The authors stated that such a High Arctic mean AOT
was never observed before (within the analyzed time period from 2003-2019). Extreme AOTs, defined as any AOT greater than the 95[th] percentile ($AOT_{95}$) of a given AOT distribution (in our case all AOTs measured from 2003-2019) (Xian et al.,

2022), were observed over the Arctic from 24 July to 22 August 2019 in phase with the most intensive Siberian fire period from 19 July to 14 August 2019.

With optically dense smoke layers, covering large areas for days to weeks, and this during the northern hemispheric summer, with long sunshine periods, the conditions for tropospheric self-lofting of smoke-containing air masses over Siberia and the adjacent Arctic Ocean were very favorable. We hypothesize that the AOT was non-uniformly distributed and partly exceeded 1.0 over large regions for days and that many smoke layers were located over extended cloud fields. It is also likely that the mean BC content in the smoke particles varied considerably from plume to plume and partly exceeded the 5-10% level. All this increased the probability for efficient self-lofting of the Siberian smoke according to our simulations.

To have an overlap with the lidar observation performed aboard the icebreaker Polarstern in the High Arctic, here we analyzed the full CALIOP data set collected at latitudes >60°N from mid of June until the end of October 2019. The MOSAiC expedition started at the end of September 2019 (Ohneiser et al., 2021). The backscatter in the UTLS layer was continuously detectable from July to October in the CALIOP data. The Polarstern lidar observed this layer and allowed a clear aerosol typing, indicating that wildfire smoke was the dominating aerosol component (Ohneiser et al., 2021). According to the analyzed CALIOP observations over central and eastern Siberia and the main outflow regime towards the Arctic, the lower stratosphere was rather clean until the end of June. A few spot-like pyroCb-lofted aerosol layers, probably generated over the North American continent, were detected by the space-borne lidar (indicated by enhanced particle depolarization ratios). From the end of June to mid-July, the number of spot-like layers with strong backscattering increased. Besides smoke layers, more and more volcanic sulfate plumes appeared at high northern latitudes.

Figure 14 shows four aerosol scenes observed with CALIOP over northern Siberia and the adjacent Arctic from mid-July to mid-August 2019. The scenes are selected because they show aerosol layering features that may be the result of smoke self-lofting. In the uppermost part of the troposphere, diffuse layers were continuously found from mid-July to mid-August 2019. Later on in September and October, diffuse aerosol structures prevailed in the lower stratosphere as well. To clarify whether the diffuse layers in Fig. 14a, b, and d were below or above the tropopause, we analyzed numerous Arctic radiosonde temperature and humidity profiles (Uni-Wyoming, 2022) on 15 and 25 July and 10 August 2019. It seems that the layers were at all below, but rather close to the tropopause and thus to the height with minimum lofting rate according to Fig. 7b. As a consequence of the decreasing lofting speed with height in Fig. 7b, ascending smoke accumulates and forms a layer with diffuse base and the tropopause as layer top height.

Such a coherent aerosol layer at the tropopause was found in the CALIOP data for the first time on 15 July 2019, and shown in Fig. 14a. The diffuse aerosol layer around 9 km height occurred over high northern latitudes west to northwest (downwind) of the main fire areas. The near-tropopause structure along the CALIOP flight track from 65°N over northern Canada to 82°N (maximum CALIOP measurement latitude) and then to 78°N over the Arctic Ocean north of Siberia, suggest that a large area with more than 3000 km in diameter was covered with smoke in the upper troposphere. The layer in Fig. 14a was well distinguishable from the plume-like pyroCb-related smoke layers and volcanic sulfate plumes (patchy features within the stratosphere) at 12-14 km height. The 532 nm AOT, estimated from the CALIOP backscatter coefficient profiles (and multiplied with a smoke lidar ratio of 65 sr) was about 0.05 for the 8-10 km near-tropopause aerosol layer. The plumes and

layers in the stratosphere at 12-14 km produced spot-like AOTs of about 0.2 on this day. Similar sharp structures (with 500-1500 m vertical extent) were observed over Europe at latitudes around 50°N in July 2019 (Vaughan et al., 2021; Ansmann et al., 2021b). Accumulation of smoke just below the tropopause was found in the CALIOP data on all following days in July 2019. Examples are shown in Fig. 14b and c. Besides the diffuse structures from 7-11 km height, again layers with strong backscattering from 12-15 km height occurred. The 532 nm AOT was about 0.1-0.15 in the diffuse layer below the tropopause.

The 26 July scenario in Fig. 14c was already discussed by Ohneiser et al. (2021). Stagnant air flow conditions favored the accumulation of smoke over central Siberia in the entire troposphere over days. Strong backscattering at heights below 4 km and strong attenuation of the lidar signals (dark areas below 3 km) in Fig. 14c indicate AOTs greater than 2.5 and, thus, good conditions for self-lofting processes. Smoke structures are visible everywhere in the troposphere up to the tropopause at about 10-11 km height. Diffuse layering is again visible from 9-11 km height.

Figure 14d shows the CALIOP observation on 10 August 2019. The 532 nm AOT of the 8-10 km layer was of the order of 0.05-0.1. A mixture of ascending smoke and sulfate particles, that formed in the lower stratosphere by conversion of Raikoke $SO_2$, was probably present above the tropopause, up to 13-14 km height. A pure sulfate layer is then visible around 16 km height. As mentioned, the occurrence of diffuse smoke layers in the uppermost troposphere was expected and predicted by the simulations in Fig. 7. We explain it as a consequence of the decrease of the lofting speed with height and the ascent rate minimum at the tropopause (shown in Fig. 7b). This aspect was explained in detail in Sect. 3.1. In the case of a continuous upward flow of particles from the middle to the upper troposphere on regional scales over days and weeks and a decreasing lofting speed with height, an accumulation of aerosols around the tropopause over extended areas of several 1000 kilometers in diameter is the logical consequence. These particles cause enhanced backscattering which is then detectable with CALIOP. Above the tropopause, on the other hand, the smoke became apparently quickly distributed over vertically and horizontally large air volumes, favored by the increasing lofting velocity with height within the stratosphere as shown in Fig. 7b, and was thus no longer clearly detectable with CALIOP.

We used the opportunity that an UTLS smoke layer crossed our lidar station at Leipzig (51.3°N) on 14 August 2019 and that CALIOP crossed Germany almost at the same time about 150 km west of Leipzig. This case is shown in Fig. 15 and was discussed in detail in Ansmann et al. (2021b). The aerosol layer extended from the tropopause up to 14 km height. Both lidars saw this UTLS aerosol layer. Backward trajectories presented in Ansmann et al. (2021b) indicated an air mass transport from Siberia over Alaska, Canada, North Atlantic towards Europe. The 532 nm AOT of the 10-14 km layer was about 0.08. The backscatter maximum was just above the tropopause, according to a nearby upwind radiosonde station, 100 km southwest of Leipzig. Other radiosonde stations to the northeast of Leipzig, however, showed tropopause heights around 10.2 km so that the backscatter maximum was just below the tropopause according to these radiosonde profiles. The ground-based Raman lidar observation shows the dominance of smoke in the UTLS aerosol layer up to 14 km height. The aerosol type was identified by the simultaneous measurement of the extinction-to-backscatter ratio (lidar ratio) at 355 and 532 nm. The layer mean lidar ratio was around 95 sr at 532 nm and thus 25 sr larger than the respective value around 70 sr for 355 nm (Ansmann et al., 2021b). Such a high 532 nm lidar ratio together with a much lower 355 nm lidar ratio is a unique fingerprint of aged smoke particles (Wandinger et al., 2002; Müller et al., 2005; Haarig et al., 2018; Ohneiser et al., 2020, 2021). The particle depolarization ratios

were <0.03 at both wavelengths, a clear signature of perfect spherical particles, and also a clear sign that pyroCb convection was not involved in the lofting processes. In cases of pyroCb-aided lofting, the particle depolarization ratio was always observed to be >0.1 during the first months after entering the lower stratosphere (e.g., Baars et al., 2019).

As suggested by the observations in Fig. 15 and confirmed by the lidar observations of dominating smoke in the UTLS height range over the High Arctic from October 2019 to May 2020 during the one-year MOSAiC expedition (Ohneiser et al., 2021), we hypothesize that the smoke continuously moved upward towards the tropopause and into the lower stratosphere in August and the first weeks in September (as long as sunlight conditions were sufficient). As a result, this continuous smoke lofting caused the smoke dominance in the UTLS aerosol layer up to about 13-14 km height. The Raikoke sulfate fractions

was estimated to be of the order of 10-15% from the MOSAiC Raman lidar observations aboard Polarstern since the late days of September 2019 (Ohneiser et al., 2021). The CALIOP observations showed a weakly backscattering diffuse aerosol layer in the lower stratosphere at high northern latitudes in September and October 2019 in agreement with the Polarstern lidar observations. However, CALIOP observations do not permit a trustworthy aerosol typing and estimation of the aerosol mixing state (sulfate vs smoke fraction) in contrast to the multiwavelength Raman lidar (Ohneiser et al., 2021).

It is noteworthy to mention, that Boone et al. (2022) in a recent paper discussed observations of stratospheric infrared absorption spectra (in the framework of the satellite-based Atmospheric Chemistry Experiment mission) and concluded that the stratospheric aerosol in the second half of 2019 over the Arctic consisted of Raikoke sulfate aerosol only. The authors found no indication for the presence of smoke. However, all our results inEngelmann et al. (2021), in Ohneiser et al. (2021), and in this lofting study unambiguously point to the dominance of smoke in the UTLS aerosol layer. We therefore have our doubts

that one can obtain a clear picture of the aerosol composition from infrared absorption spectra alone. A clear indication for the presence of smoke was the enhanced carbon monoxide (CO) concentration in the lower stratosphere observed in August 2019. We checked satellite-based observations of the CO concentration from July to October 2019 for the area from 67°-143°E and 70°-87°N, i.e., over the northern part of central Siberia and over the adjacent Arctic (AIRS, 2022), and found a clearly enhanced monthly mean CO concentration in the upper troposphere and lower stratosphere (100-150 hPa) in August 2019

compared to the respective August mean values of the background years 2013-2018, 2020, and 2022.

In summary, to our opinion, we found clear indications supporting our hypothesis that self-lofting processes played a key role in the evolution of a smoke-dominated UTLS aerosol layer over the Arctic Ocean. Typical optical fingerprints for aged smoke particles were found in this layer from the dual-wavelength Raman lidar observations over Leipzig in August 2019 and over the High Arctic during the winter half years 2019-2020 (Ohneiser et al., 2021). The observed low particle depolarization

ratios clearly showed that lofting by pyroCb convection played no or only a minor role. Note, however, that in this article, we leave out a discussion of the further ascent of smoke from the tropopause towards greater heights. Generally, self-lofting within the stratosphere is already a well-documented process in transport of light-absorbing particles higher up (Kablick et al., 2020; Khaykin et al., 2020; Allen et al., 2020; Lestrelin et al., 2021; Das et al., 2021).

## 5.1 Case study of self-lofting during the summer 2021 wildfires

A promising way to directly observe self-lofting processes in the troposphere may be the combination of backward and forward trajectory analysis and subsequent CALIOP observations (overflights over almost the same area) within a likewise short time period of 12-24 hours. Such an example (case study) is briefly discussed in this subsection. Two CALIOP overflights of the same, heavily polluted air mass within 15 hours during the record-breaking Siberian fires in 2021 provided an opportunity to directly detect self-lofting effects. We selected an extreme smoke situation on 5 August 2021 as shown in Fig. 4 in Xian et al.

(2022). Large areas from central Siberia up to the North Pole were covered with smoke. In extended areas the smoke AOT exceeded 2. Favorable conditions for self lofting were given.

Figure 16 shows the smoke situation as seen by CALIOP in the afternoon (local time) of 5 August at 65°N, 88°E and 15 hours later on 6 August, about 4 hours after midnight 68°N, 82°E. The air mass experienced 8 hours of sunlight during these 15 hour travel. The two selected CALIOP tracks were almost orthogonal to each other, the flight around 7:20 UTC was from

northwest to southeast and thus along the main wind direction according to HYSPLIT trajectory computations. In Fig. 16a, we arranged the data the other way around, from southeast to northwest. The flight around 22:20 UTC was then from northeast to southwest (Fig. 16b). Only the two scenes below the vertical arrows in (a) and (b) can be directly compared as the forward and backward HYSPLIT trajectories (ensembles) indicate. According to the forward trajectories the air mass traveled about 300 km within 15 hours or with 5 m s$^{-1}$ in northwesterly direction. We see roughly an ascent rate of 2.5 km within these

15 hours. This ascent rate is in good agreement with our simulation shown in Fig. 11 for a BC content of 3.5% and an initial AOT of 3.0 (black dashed line).

The stratospheric mean CO concentration over the northern part of central Siberia in August 2021 was much higher than in the foregoing record-breaking smoke year of 2019. It seems to be again probable that self-lofting may have contributed significantly to the smoke transport towards the lower stratosphere. CALIOP observations often showed smoke pollution up

to the tropopause level at latitudes of 70°-80°N, especially during the second half of August 2021. The depolarization was partly very low, but also sometimes enhanced. We conclude that self-lofting as well as pyroCb-related lofting occurred and contributed to the smoke in the UTLS height range in 2021.

## 6 Summary and outlook

A detailed simulation study regarding the potential of optically thick smoke layers to reach the UTLS region by self-lofting

processes was presented. Goal was to show that there is an alternative way to loft considerable amounts of smoke towards the tropopause in the absence of pyroCB convection.

An ECRAD-based simulation scheme was developed that allowed us to estimate the self-lofting rates of smoke layers caused by heating due to absorption of solar radiation by smoke particles leading to the subsequent ascent of the layer in the troposphere and stratosphere. We discussed the influence of the required input parameters such as AOT, layer thickness

and height, absorption properties of smoke particles, and of atmospheric parameters (cloudiness, relative humidity, potential temperature gradient) on the self-lofting results in the framework of a sensitivity and uncertainty study. CALIOP observations

of lofting processes in the stratosphere after major pryoCb events (Canadian fires in August 2017, Australian fires around New Year 2020) were compared with simulations and demonstrated the good performance of the self-lofting model.

An open issue is the proper consideration of smoke optical properties in the simulations. More laboratory efforts and airborne in situ observations of aged smoke are required to improve our knowledge about relationships between the chemical composition (including a better understanding regarding internal vs external mixing) and microphyscial properties (size distribution, shape features), as well as light-scattering and absorption properties of aged smoke particles after long-range transport over weeks to months.

As an important part of the study, we analyzed long-term observations of Siberian smoke layers and plumes evolving in the troposphere and the UTLS region over Siberia and the adjacent Arctic during the record-breaking wildfire season of 2019. Based on several independently measured aerosol properties and layering features, mentioned in Section 5, we have two main arguments that smoke self-lofting obviously played a significant role in the built up of a strong, long-lasting, smoke-dominated aerosol layer in the UTLS height range in July and August 2019. The observed high AOT and the inverse spectral slope of the measured lidar ratio point to a smoke-dominated aerosol layer in the UTLS height range. The low depolarization ratio indicate that pyroCb-related smoke lofting was of minor importance so that only smoke self-lofting remained as the main smoke lofting process. And the second argument is the occurrence of a near-tropopause layer, best visible from mid of July to mid of August 2019, which is in line with our self lofting simulations, predicting an accumulation of smoke at the tropopause in the case of a steady upward transport of smoke over several days or even one or two weeks.

Disregarding all these corroborating facts we collected and found in the observations indicating that self-lofting of smoke can be a major player in smoke lofting processes it remains an open question whether or not self-lofting was responsible for the development of the UTLS smoke layer over the High Arctic in the summer of 2019. Further work is needed with more sophisticated models coupling realistically radiative and 3D dynamical effects in the troposphere. Also further smoke observational studies with focus on self lofting processes are needed in combination with detailed trajectory analysis to further clarify the role of self-lofting processes as an alternative to pyroCb lofting.

Meanwhile, there is so much smoke around the world (in tropical and southern Africa, South America, western North America, Alaska, Canada, Siberia, in the Mediterranean, Middle East and Central Asia, Southeast Asia, Australia), so that we believe, self-lofting colud play a very important role on a global scale. However, it remains rather difficult to provide clear observations of slowly ascending, likewise thin aerosol layers, and thus to provide evidence that self-lofting occurred. Self-lofting leads to a prolongation of the lifetime of all these light-absorbing particles in the atmosphere and this aspect is probably not considered in any of the numerous climate models used to predict climate change.

## 7 Data availability

CALIPSO observations were downloaded from the CALIPSO data base (CALIPSO, 2022). Polly lidar observations (level 0 data, measured signals) are in the PollyNet database (Polly, 2022). The radiosonde data are available at **?**. CAMS data are available on the Copernicus website CAMS (2022). MODIS data are available at the NASA data base MODIS (2022). The

same holds for the AIRS data (AIRS, 2022). Forward trajectory analysis has been performed by air mass transport computation with the NOAA (National Oceanic and Atmospheric Administration) HYSPLIT (HYbrid Single-Particle Lagrangian Integrated Trajectory) model (HYSPLIT, 2022).

## 8   Author contributions

The paper was written and designed by KO and AA. The model simulations and data analysis were performed by KO, AA, JW, and GW, supported by HD, FS, and AC (MODIS and AIRS data analysis). All co-authors contributed to the discussion of the results.

## 9   Competing interests

The authors declare that they have no conflict of interest.

## 10   Financial support

The authors acknowledge support through the European Research Infrastructure for the observation of Aerosol, Clouds and Trace Gases ACTRIS under grant agreement no. 654109 and 739530 from the European Union's Horizon 2020 research and innovation programme.

*Acknowledgements.*   We thank Reinout Boers (KMNI, The Netherlands) to motivate us to perform this extended study on smoke-self-lofting. We also thank Blake T. Sorenson (University of North Dakota) for providing and explaining the timeseries of AI and AOT measured over the Arctic Ocean in 2019. We are grateful to the CALIPSO team for their well-organized easy-to-use internet platforms. Analyses of AIRS data used in this study were produced with the Giovanni online data system, developed and maintained by the NASA GES DISC.

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

**Table 1.** Important input parameters in the self-lofting simulations, their influence on the results (in a linear or nonlinear way), and typical uncertainties in the self-lofting results caused by uncertainties in these input pararmeters. More details are given in the text.

| Parameter | Impact | Uncertainty |
|---|---|---|
| AOT | ≈linear | 50% |
| Layer thickness | ≈linear | 20% |
| Layer height | nonlinear | 20% |
| Injection height | nonlinear | 20% |
| BC/OC ratio | ≈linear | 30% |
| BC type | nonlinear | 20% |
| Cloudyness | ≈linear | 50% |
| Relative humidity | nonlinear | 5% |
| Pot. temp. gradient | linear | 10% |

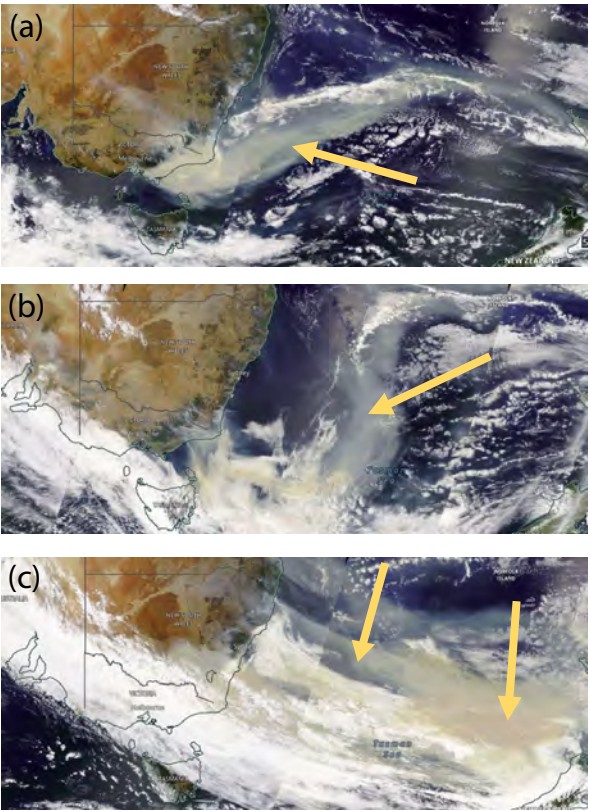

**Figure 1.** MODIS visible satellite images over the Pacific between Australia and New Zealand on (a) 3 January, (b) 4 January, and (c) 5 January 2020. The smoke fields indicated by yellow arrows traveled eastward towards South America.

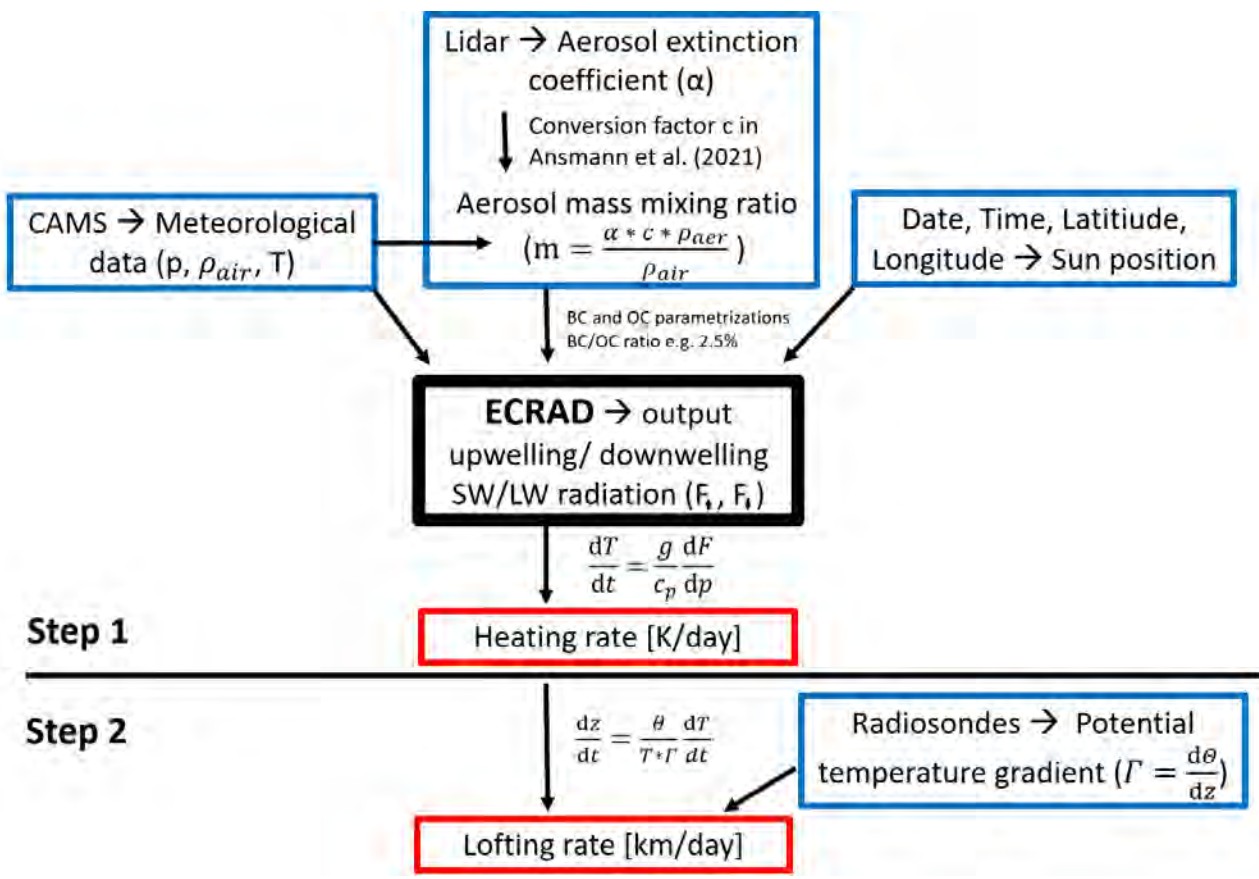

**Figure 2.** Simulation flowchart with the ECRAD simulation model in the center. Input are the height profile of the particle extinction coefficient (or AOT, e.g., from lidar), CAMS meteorological parameters (CAMS, 2022), and the diurnal cycle of Sun position. ECRAD output allows us to calculate heating rates. In a second, independent step (indicated in red), these heating rates, in combination with radiosonde profiles of temperature (**?**), are used to compute the lofting rates. Further information is given in the text.

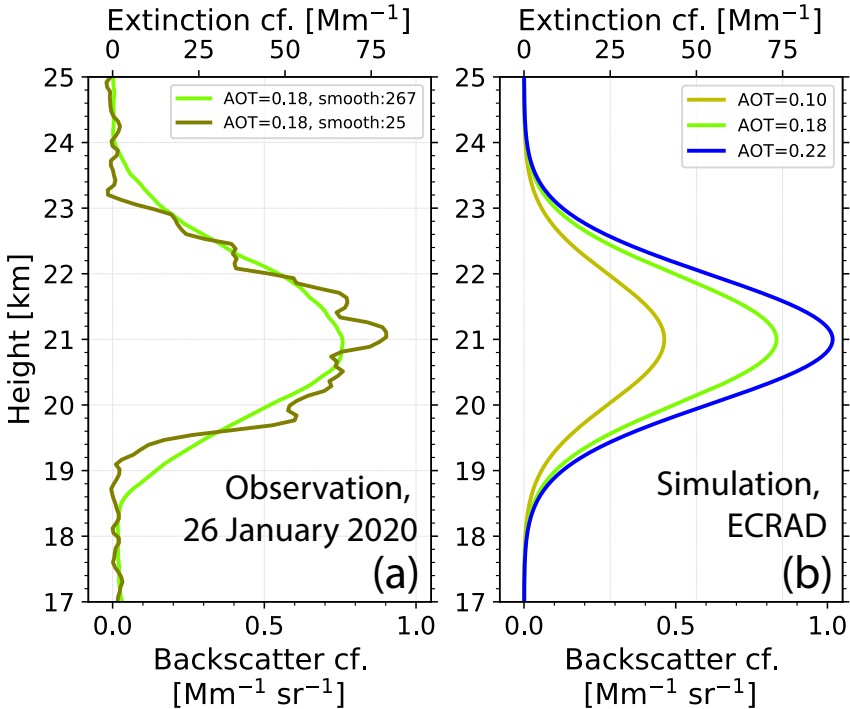

**Figure 3.** a) 532 nm particle backscatter coefficient of a wildfire smoke layer between 19 and 23 km height. The smoke plume was measured with ground-based lidar at Punta Arenas on 26 Jan 2020, 04:27-06:18 UTC (Ohneiser et al., 2022). The vertical signal smoothing length is 187.5 m (25 bins, olive profile) and 2002.5 m (267 bins, green profile). The extinction coefficient is obtained by multiplying the smoke backscatter coefficients with a lidar ratio of 91 sr (Ohneiser et al., 2022). The smoke layer optical thickness is 0.18. b) Parameterized backscatter and extinction profiles (Gaussian shape) with adjustable layer center and layer thickness, here for AOT=0.10, 0.18, and 0.22. The parameterized profiles are used as input in the heating rate simulations.

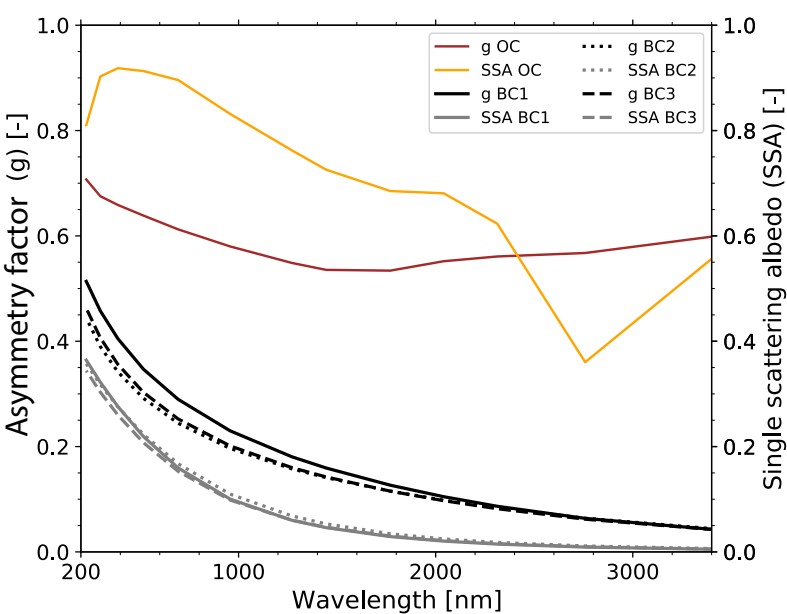

**Figure 4.** Optical properties of wildfire smoke. Asymmetry factor g and single scattering albedo SSA of OC (orange, brown) and for three different BC parameterizations (BC1, BC2, BC3) used in the ECRAD simulations.

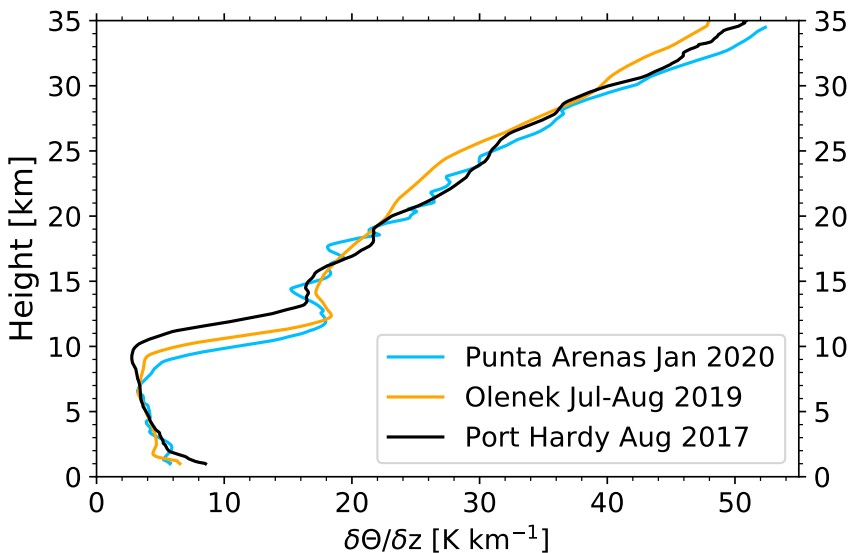

**Figure 5.** Gradient of the potential temperature at Punta Arenas (Chile, 53.17°S, 70.93°W, January 2020), Olenek (Russia, 68.50°N, 112.43°E, July-August 2019), and Port Hardy (Canada, 50.68°N, 127.36°W, August 2017) obtained from radiosonde data (**?**) and used in the self-lofting simulations (see Fig. 2).

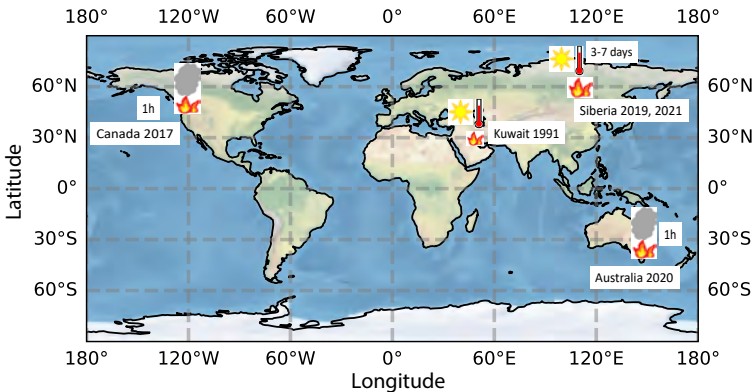

**Figure 6.** Map of the fire events discussed in this study. PyroCb-related smoke lofting (symbolized by a grey cloud above the fires) occurred over British Columbia, Canada, in August 2017 and over Southeast Australia in December 2019 and January 2020. It is assumed that self-lofting of wildfire smoke (symbolized by a Sun and a thermometer over the fires) occurred over Siberia and the outflow regime (i.e., mainly over the Arctic region north of 60-70°N) in July and August 2019. Similarly, we assumed self-lofting over the oil-burning smoke areas in Kuwait in March 1991. Smoke reaches the tropopause within a short time period (of the order of 1 h) in the case of pyroCb convection and may reach the tropopause after several days (3-7 days) in the case of self-lofting.

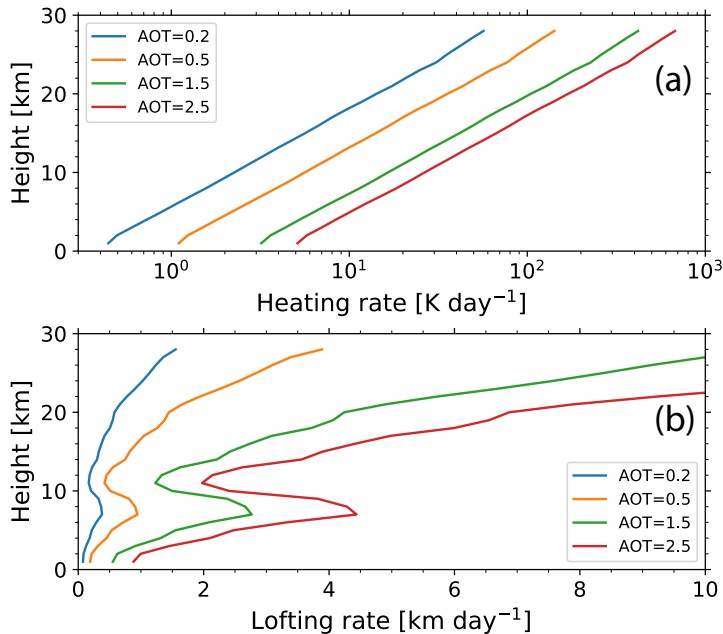

**Figure 7.** ECRAD simulations of (a) heating rate and (b) corresponding lofting rate as a function of height for 4 different AOTs of a 2 km thick smoke layer. The center height is stepwise increased by 1 km in the simulation from 1 km to 28 km height. The Punta Arenas temperature gradient profile in Fig. 5 is used. Daily-average heating and lofting rates are simulated.

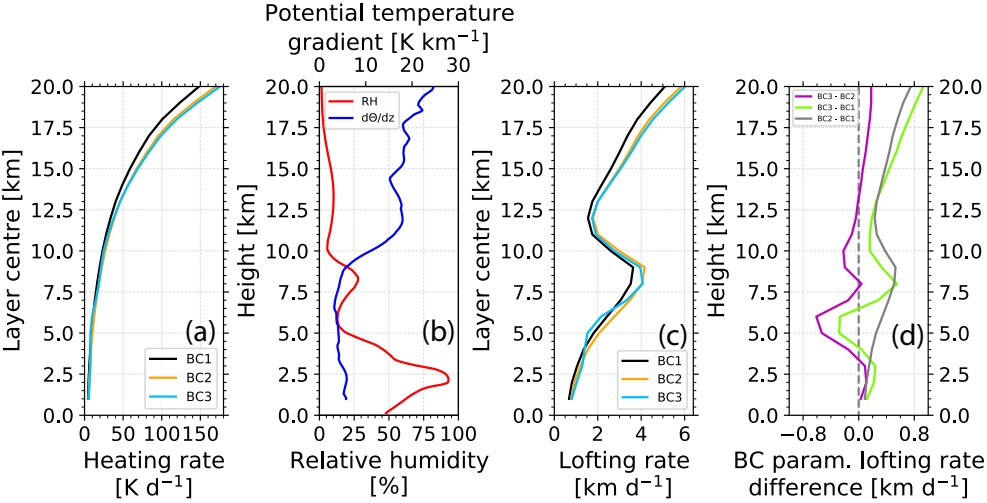

**Figure 8.** ECRAD self-lofting simulation (a: heating rate, c: self-lofting rate) of an ascending 2 km thick smoke layer (AOT=1.5 at 532 nm, Punta Arenas, 26 January 2020) for three different BC parameterizations shown in Fig. 4. The relative humidity in (b) influences the BC parameterization. In (d), differences between lofting rate solutions by considering two of the three BC parameterizations (BC1, BC2, BC3) are shown. The simulated daily-average heating and lofting rates are given for the layer center height.

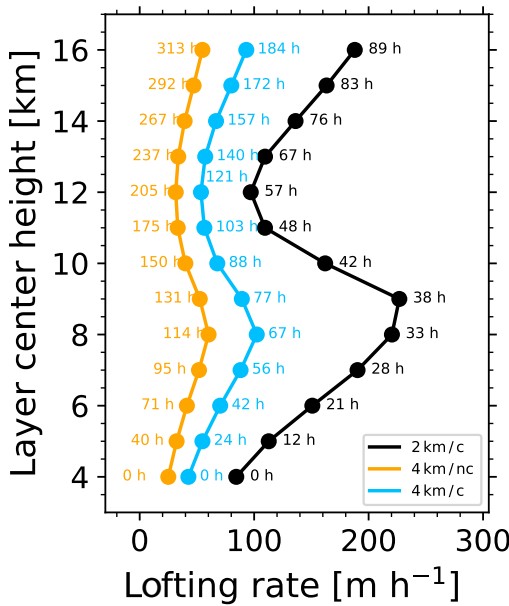

**Figure 9.** Simulation of self-lofting of a wildfire smoke layer. The 532 nm AOT of the 2-km (black profile) and 4 km (orange and blue profiles) thick smoke layers, initially centered at 4 km height, was assumed to be 2.0. Two overcast ('c' - cloudy, black, blue) scenarios and one clear sky ('nc' - no clouds, orange) scenario are simulated. Numbers indicate the time in hours after start of lofting.

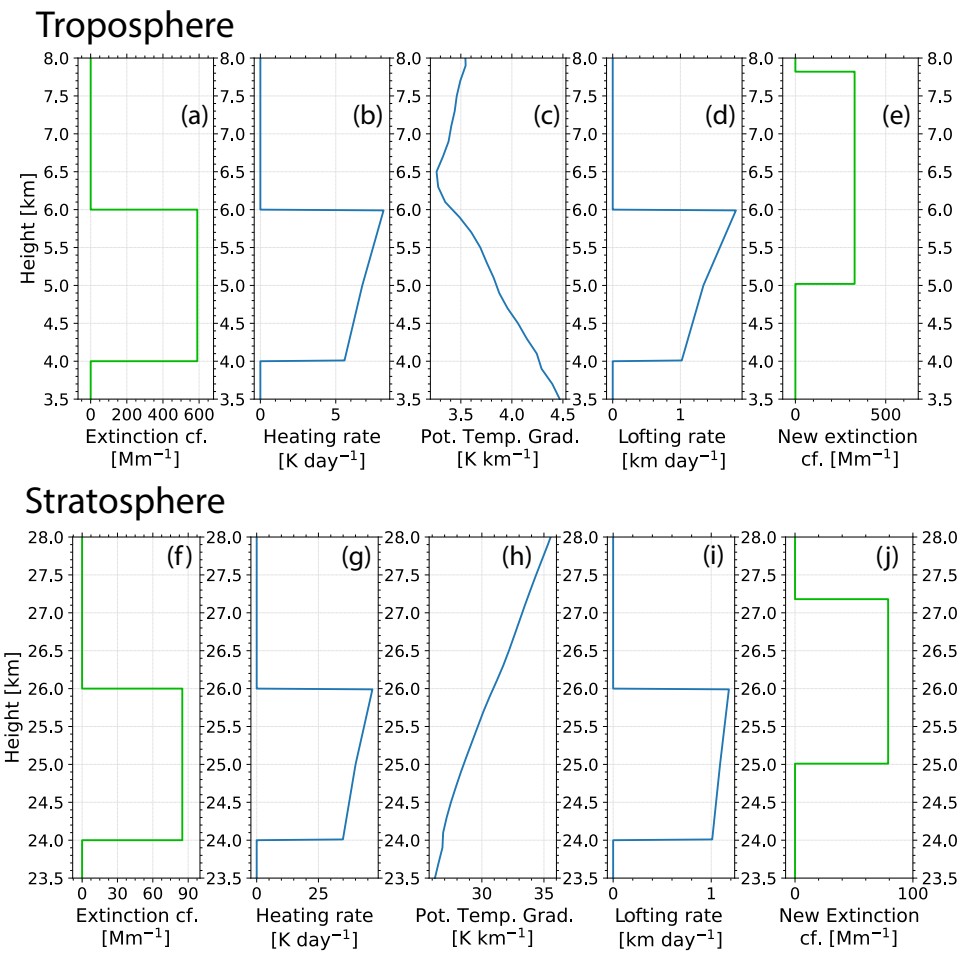

**Figure 10.** Self-lofting effects on layer depth in the troposphere (a-e, layer at 4-6 km height before lofting) and in the stratosphere (f-j, layer at 24-26 km height before lofting). Step by step calculation of the new layer profile after 1 day of heating and lofting. The extinction profiles in (a) and (f) were used in the simulations, (b) and (g) show the resulting heating rate profiles, (c) and (h) the potential temperature profiles at Punta Arenas assumed in the simulations, (d) and (i) the resulting lofting rate profiles, (e) and (j) finally the resulting new extinction coefficient profiles in the troposphere and stratosphere after 1 day of self-lofting. Extinction profiles (old and new) are highlighted in green.

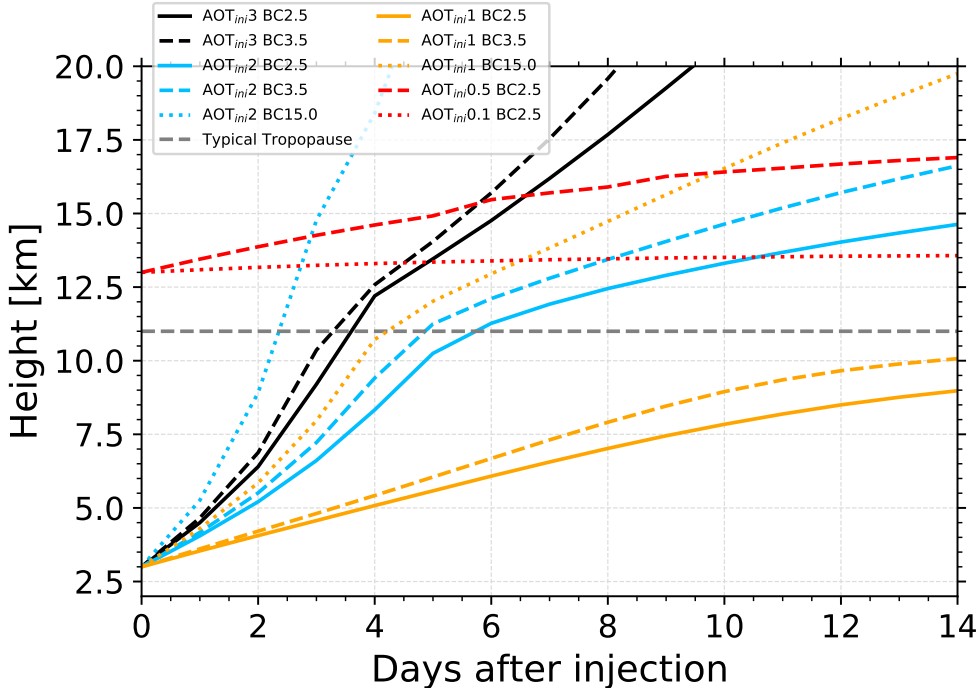

**Figure 11.** Change of the center height of a 2.5 km thick aerosol layer, initially at 3 km height (day 0), during 14 days of continuous lofting (black, blue, orange). In the simulation, the AOT continuously decreases by 15% from day to day. Different scenarios with different initial AOT of 1, 2, and 3 (indicated by index ini), and BC fraction of 2.5%, 3.5% and 15% are simulated. In addition, the lofting behavior of a stratospheric layer initially at 13 km height (red lines, see legend regarding AOT and BC fraction, 15% AOT decrease from day to day) is shown. The grey dashed line represents a typical tropopause height in the mid-latitudes.

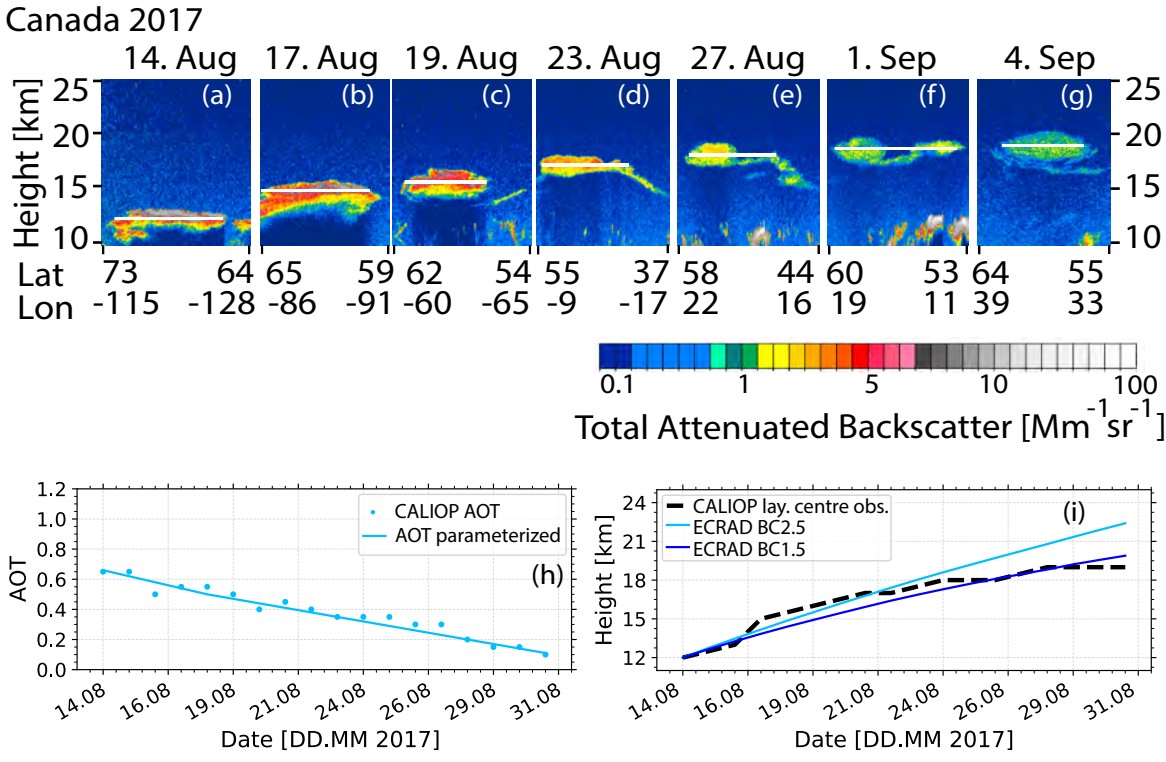

**Figure 12.** (a-g) Height-time display of the total 532 nm (Rayleigh + particle) attenuated backscatter coefficient of a Canadian wildfire smoke layer observed with CALIOP on 6 different days within the three-week period from 14 August to 4 September 2017. The white horizontal lines indicate the layer center height of the ascending smoke layer. In panel h, the AOT observations are given. The CALIOP layer mean total attenuated backscatter coefficient was multiplied with a lidar ratio of 65 sr (Baars et al., 2019) and with the layer thickness (retrieved from panels a-g) in order to obtain the daily AOT (blue dots). The time series of AOT is parameterized (blue line). By using the parameterization in panel h, the heating rates and subsequent lofting rates are simulated, shown in panel i for 1.5% and 2.5% BC fraction and compared with the observed lofting rates (black dashed line).

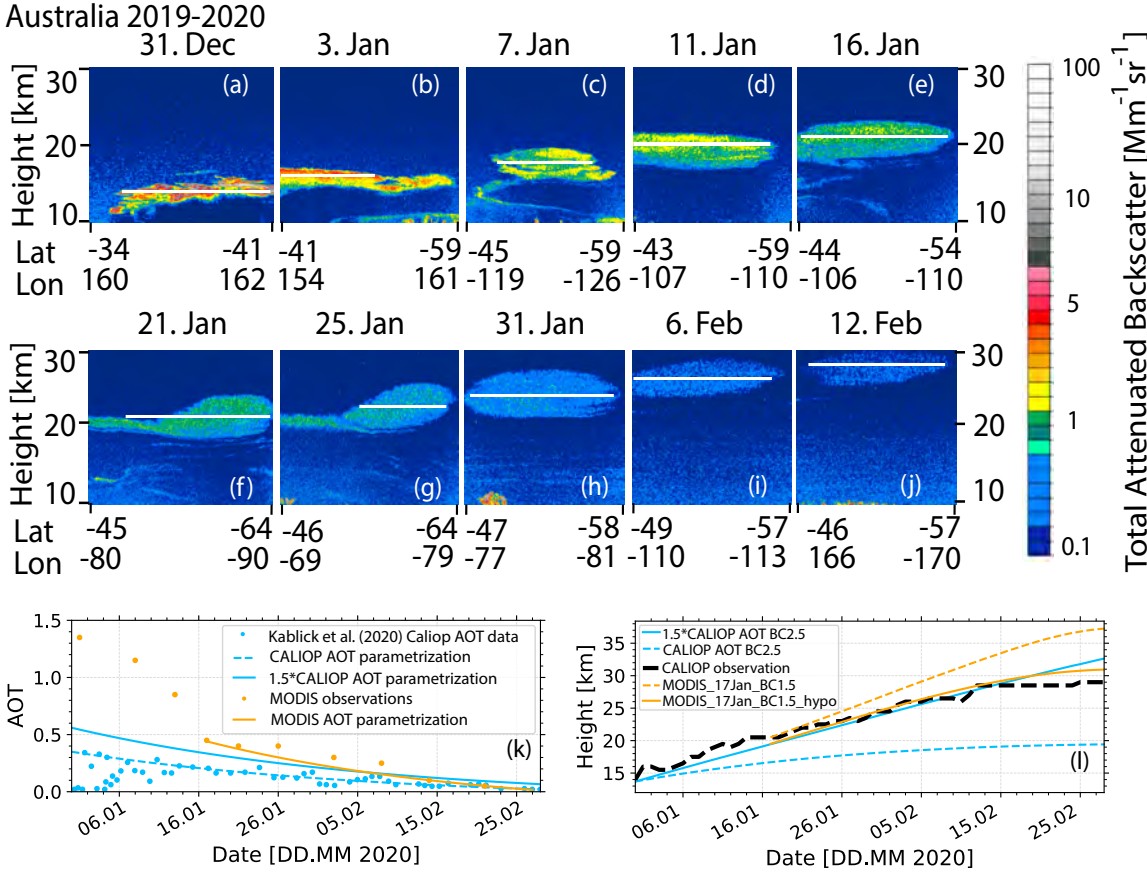

**Figure 13.** (a-j) Height-time display of the total 532 nm attenuated backscatter coefficient of an Australian wildfire smoke layer observed with CALIOP on 10 different days within the 6.5-week period. The white horizontal lines indicate the layer center height of the ascending smoke layer. In panel k, the AOT observations are given, used in the simulations in panel l and retrieved from MODIS observations (orange dots, parameterization as orange line) and derived from the CALIOP observations (blue dots, (Kablick et al., 2020), parameterization in dashed blue, 1.5*AOT parameterization as solid blue line). By using the MODIS and CALIOP AOT parameterizations (MODIS parametrization and simulations start on 17 January), the heating rates and subsequent lofting rates are simulated, shown in panel l, and compared with the observed lofting rates (black dashed line). The different CALIOP AOT parameterizations (blue) were used together with a BC fraction of 2.5%. In the case of the simulations with the MODIS AOT parameterization (orange, BC fraction of 1.5%), the simulations start at different (initial) layer heights. In the simulation, shown in orange (MODIS_17Jan_BC1.5_hypo), the starting height was decreased by 1 km.

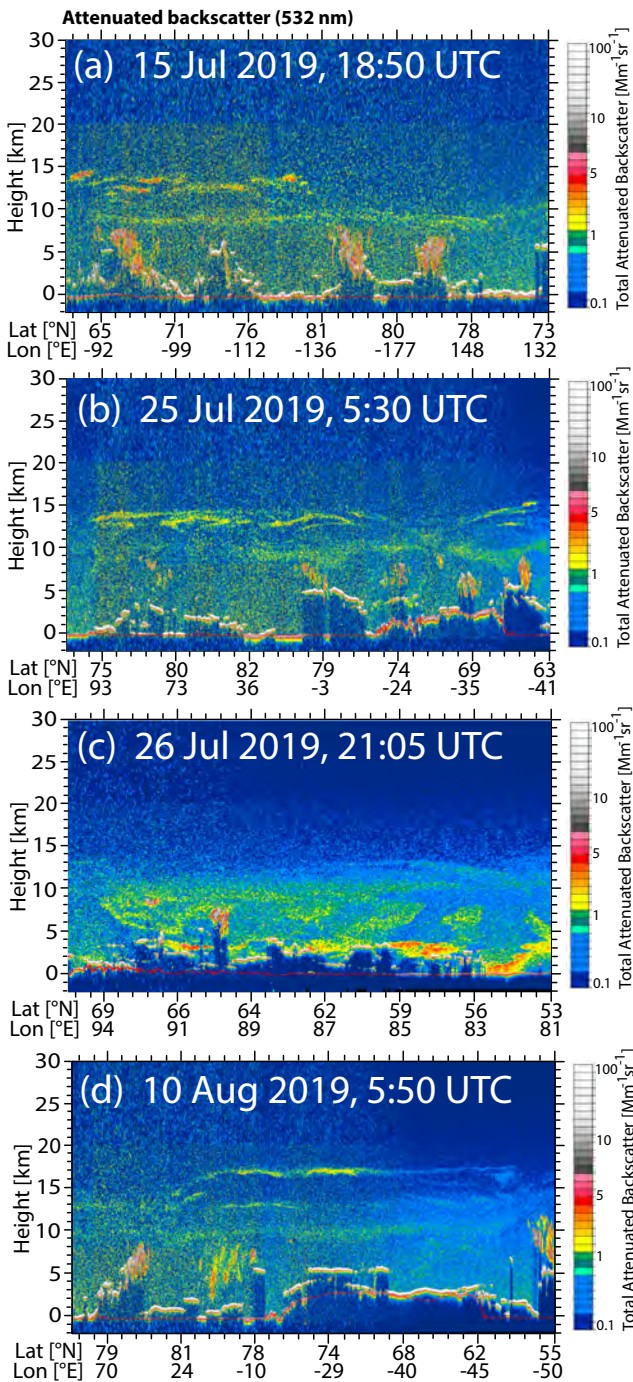

**Figure 14.** CALIOP measurement (height–latitude/longitude display of 532 nm attenuated aerosol backscatter) of wildfire smoke in the UTLS region over central Siberia and the Arctic on (a) 15 July 2019, (b) 25 July 2019, (c) 26 July 2019 (already discussed in Ohneiser et al. (2021), and (d) 10 August 2019. Thick smoke plumes (red, green and yellow) reaching the tropopause at 8-9 km height and forming a coherent layer in the lower stratosphere from 8-10 km height were observed. Another backscatter maximum around 14 km height is caused by pyroCb-related smoke and Raikoke sulfate plumes.

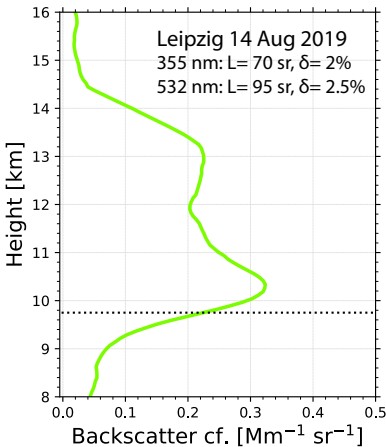

**Figure 15.** 80-min mean height profiles of the 532 nm particle backscatter coefficient showing an extended smoke layer in the UTLS region from 9-14.5 km height over Leipzig (51.3°N), Germany, on 14 August 2019, 00:00-01:20 UTC. The measured layer-mean particle lidar ratios (L) and particle depolarization ratios (δ) at 355 and 532 nm are given as numbers and are typical for aged, spherical smoke particles. The dotted line shows the tropopause as measured by nearby radiosonde.

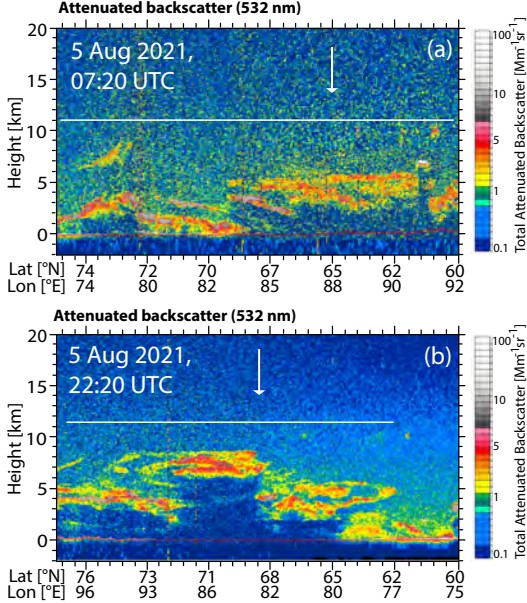

**Figure 16.** CALIOP measurement of 532 nm attenuated aerosol backscatter caused by thick wildfire smoke in the troposphere over central northern Siberia on (a) 5 August 2021, 7:20 UTC (early afternoon over central Siberia) and (b) 5 August 2021, 22:20 UTC (early morning). Horizontal lines indicate the tropopause. The vertical white arrow in (a) shows the starting point of the HYSPLIT forward trajectory and the white arrow in (b) the end point of the forward trajectory (downwind) after 15 hours.