# Peer review of "Self-lofting of wildfire smoke in the troposphere and stratosphere: simulations and space lidar observations"

_Atmospheric Chemistry and Physics, 2022_

## Referee Comment (RC1)

Review of Ohneiser et al. (2022), "Self-lofting of wildfire smoke in the troposphere and stratosphere caused by radiative heating: simulations vs space lidar observations" Reviewer: Mike Fromm

Ohneiser et al. explore the phenomenon of diabatic ascent of solar-radiation-absorbing smoke particles with a combination of satellite lidar- and visible-wavelength imager-based aerosol observations and modeling. The diabatic lofting in their focus is broken down into two categories: free-troposphere-to-stratosphere (hereafter "FTTS") and exclusively stratospheric. The latter of these two has been unequivocally observed in multiple case studies. The former (FTTS) has been hypothesized but not confirmed observationally.

It is the FTTS aspect that is the driver of this work, as revealed in the Abstract: "One of the main goals is to demonstrate that self-lofting processes can explain observed smoke lofting in the free middle and upper troposphere up to the tropopause and into the lower stratosphere without the need for pyrocumulonimbus convection." This is the logical extension of this author-group's prior publications that argued for non-pyroCb stratospheric smoke being the dominant particulate constituent in the Arctic in 2019-2020 (as opposed to Raikoke volcano sulfates, put forward in a host of additional recent publications). Hence, this manuscript attempts to test their hypothesis with observations and modelling support thereof.

The authors' observational test includes not only the Raikoke summer of 2019 but an apparently similar scenario in August 2021 when Siberian fires also produced dense, synoptic-scale, multi-day smoke plumes. They focus on CALIOP backscatter curtains for a few selected days each year over the smoky domain. From those curtains the authors mark vertical locations representing a smoke-layer centroid that presumably represents a traceable ascent of the smoke they attribute to self-lofting. In addition to the CALIOP observations, the authors show HYSPLIT forward trajectories in association with the CALIOP plume heights over the multi-day plume evolution. Back trajectories, briefly described and unshown are also invoked to argue for diabatic ascent of the smoke layers.

I found the entirety of the Siberia CALIOP and trajectory analysis (Section 4.4) to be confusing, arbitrary, speculative, error-prone, hence unconvincing. Below I will point out the various issues I found in greater detail. But in summary, my assessment is that the authors did not demonstrate observational evidence of diabatic self-lofting of Siberian tropospheric smoke in either 2019 or 2021. Absent that evidence, the reader has no basis on which to accept the modeling results presented herein and thus be convinced of the novel hypothesis presented in foundational work of Ohneiser et al. (2021; "021").

Moreover, the authors clearly did not show incontrovertible observational proof of the FTTS diabatic self-lofting pathway required for the presumed stratospheric source term for the Arctic MOSAiC lidar-based conclusions of O21, on which Engelmann et al. (2021), Ansmann et al. (2021a), and Ohneiser et al. (2022) depended. From O21's Figure 13, one can project that the

nascent, tropopause-level Siberian smoke resulting from FTTS transport would have an AOT of ~0.2 to 0.3 in August 2019. Layers embodying such an AOT would present as an unmistakable signal in CALIOP data. By their own analysis in Section 4.4 (Fig. 17), the Siberian smoke in 2019 had ascended only to 5 km (~5 km below the tropopause). Presumably, had there been detectable, efficient self-lofting to the tropopause, the authors certainly would have shown it. In the apparent absence of smoke FTTS proof, the authors were left to state: "Fractions of this smoke plume must have reached the tropopause and later on the lowest stratosphere." Thus, even if their observational interpretations are robust, the authors did not achieve one of their "main goals." The reader is left to rely on hypothetical modeling such as already presented in O21 and updated herein. From the Summary and Outlook section: "the large number of open parameters generates large uncertainty / sensitivity to the modeled lofting rates." Hence, we are no closer to a fire-emission/stratospheric-plume-pathway connection in boreal 2019 as before.

The authors' main goal was to support the contentions of O21, Engelmann et al. (2021), Ansmann et al. (2021a), and Ohneiser et al. (2022) that a boreal stratospheric smoke plume was present in 2019/2020 with AOT and persistence on par with certain volcanic injections and major pyroCb outbreaks. Their approach was to establish proof by observation in support of O21's hypothesis. My assessment is that the observational proof was not given in spite of expectations. If my assessment is accurate, the authors have strategically falsified the O21 hypothesis by looking for and not finding a suitable source term for the stratospheric smoke that O21 and associated papers proffered. This is a valid and valuable scientific conclusion. The current manuscript is thus positioned to challenge those previous conclusions. The authors are asked to consider using this current work to acknowledge the lack of observational evidence for Siberian smoke FTTS transport and the implications for O21, Engelmann et al. (2021), Ansmann et al. (2021a), Ohneiser et al. (2022), etc.

If my assessment is incorrect, it is incumbent on the authors to establish observational proof of the 2019 Siberian-smoke pathway to the stratosphere. The modeling work had essentially been done by O21; the modelling in this paper on its own does not serve as proof.

Regarding the observations and modeling the authors present on the second diabatic pathway, exclusively stratospheric, the simulations make a valuable contribution to the literature. As acknowledged by the authors, the stratospheric scenario is much cleaner and simpler to follow. Moreover, prior publications have already provided a foundation for tracking the smoke-plume features studied herein. Considering my concerns with the FTTS aspect of this manuscript and the implications for publication merit, I will focus my comments on that rather than the stratospheric part.

**Major Concerns**

L49, "Recently, smoke self-lofting was observed in the middle and upper troposphere (Ohneiser et al., 2021; Engelmann et al., 2021).": It is more accurate to say that self-lofting was inferred. Only a single CALIOP curtain was displayed by O21, giving a static view of vertical aerosol placement. Inferences were made based on Lagrangian trajectories and modeling.

L55-57, "...Siberia in the absence of pyroCb convection as spaceborne lidar observations indicated (CALIPSO, 2022).": How can CALIOP data be used to show there was no pyroCb action? The cited reference gives no help. In this Introduction section, it is presumed that any cited material will support the specific statement made. The reference here is a generic identifier of the CALIPSO data repository. In point of fact, there were 5 pyroCbs in the Sakha region of Siberia in July and August, 2021. PyroCb alerts and follow-up discussion in 2021 were shared in near real time on the Worldwide pyroCb Information Exchange (https://groups.io/g/pyrocb), for the specific purpose of engaging researchers across the globe and documenting individual pyroCb eruptions. The authors are asked to review the communication on this open platform and revise their description of the 2021 boreal and Siberia pyroCb occurrence.

L57, "Part of the smoke again entered the stratosphere in August 2021.": No doubt. One or more of the reported pyroCbs (see above) naturally injected smoke to or across the tropopause. Moreover, many other boreal pyroCbs were catalogued in 2021 (see the above link to the reporting medium). It is inappropriate for the Introduction to this paper to combine the claim of no pyroCbs and the supposed self-lofting from Siberia to the stratosphere.

L59-65, Discussion of smoke heating, buoyancy, self-lofting: It is important to treat the arguments of Boers et al., de Laat et al., Torres et al., and Yu et al. separately. Torres et al. and Yu et al. dealt explicitly with a puroCb-generated source term at the tropopause and primarily stratospheric transport whereas Boers et al. and de Laat et al. considered lofting of non-pyroCb-generated smoke in the lower to middle troposphere (analogous to this manuscript's Siberia hypothesis). The distinction I call for is because stratospheric self lofting has been well characterized observationally (to which the Kablick, Khaykin, and other citations in this paragraph attest) whereas the FTTS pathway has not. It also must be noted that the Boers/de Laat pathway on Australia's Black Saturday has been reinterpreted as a classic pyroCb event, with rapid injection to the tropopause (Fromm et al., 2021, https://doi. org/10.1029/2021JD034928). Please revise the discussion here to reflect the disparate levels of certainty regarding pyroCb vs. non-pyroCb smoke pathways.

L199, "Chemical aging is assumed to lead to a spherical shape...": On what basis is this assumed? Please defend this statement. There is abundant literature showing that aged tropospheric smoke particles embody depol. ratios that are significantly greater than the limiting value for uniform spheres, e.g. Dahlkötter et al (2014; doi:10.5194/acp-14-6111-2014), Burton et al. (2015; doi:10.5194/acp-15-13453-2015), Hu et al. (2022; https://doi.org/10.5194/acp-22-5399-2022),

Liu et al. (2022, https://doi.org/10.1016/j.jqsrt.2022.108080). Hence, Table 2's "2%" "spherical" characterization is apparently an inaccurate simplification of non-pyroCb, tropospheric smoke.

Please revise to reflect the literature. Since the weight of the above-mentioned papers rests on the idea that aged smoke is better represented by an aspherical model than monolithic spheres, the implications of changing Table 2 are momentous. The authors are asked to reconsider their assumptions for aged tropospheric smoke and the implications for the optical properties they observed in the stratosphere during MOSAiC.

Regarding the concern above regarding the authors' assumption of spherical, aged smoke in Siberia (and eventually in the stratosphere), it is important to note that the only other published work describing non-pyroCb, FTTS transport is de Laat et al., (2012). Even if it is maintained that their "solar escalator" was an accurate depiction, the resultant stratospheric smoke from Black Saturday had depolarization ratios weeks after onset so great that the CALIOP observations thereof triggered a feature classification of "ice" ( https://acp.copernicus.org/preprints/acp-2021-117/acp-2021-117-RR1.pdf). Hence these aged smokes were wholly inconsistent with the assumptions shown in Table 2. The authors are asked to comment on this inconsistency and make the appropriate revisions to their current assumptions on smoke aging.

Section 4.4. This section is problematic. My concerns are many and deep. This part of the paper is seriously flawed. Below I list the issues individually.

L482, "The smoke at different heights was transported in the same air column.": Meteorologically, I cannot make sense of this. What does "same air column" mean? The trajectories take different paths and have very different endpoints. This implies considerable speed and directional shear, which by itself seems to suggest a variable air column. Please explain.

Discussion of Figures 17 and 18: How "old" are the smoke layers on 12 August 2019 and 4 August 2021? Without knowing that, the reader has no idea how all those smokes got to that start point. Almost assuredly they didn't all arrive in a single "air column," but no information is given. I.e. these seem like arbitrary start points and hence it is unclear how to assess transport after the start points in relation to transport to them.

**L482, 483, "The trajectories analysis in Fig. 18a**

indicated similar wind speed and directions at different heights in the middle troposphere.": I cannot reconcile Fig. 18a with this description. There are significant differences in both aspects. If the authors want to convince the reader that these trajectories are similar, they need to quantify that and show what dissimilar trajectories would look like.

**L483, 484, "The smoke slowly ascended from**

around 3-4 km height to 5-6 km height.": The trajectories themselves account for almost a 1-km altitude gain. I.e. resolvable meteorology is responsible, not diabatic lofting. The authors need to acknowledge this.

L484, 485, "The backward trajectories (not shown) indicate no direct link of the air mass at 5-6 km height with lower heights (and thus a direct fire smoke uptake from the sources)." For this to make sense, the trajectories must be shown. Even so, it is difficult to understand what the authors mean here. What is "direct...uptake"? What are "the sources"? Either remove this sentence or bolster it with displayed trajectories and precise terminology.

L489-494, Nearly identical trajectory discussion regarding the 2021 smoke: I have the same concerns as stated above. Please take the same corrective actions.

L494, "Initial smoke plumes were at around 3 km height on 4 August 2021.": What is meant by "initial" when the previous sentence refers to smoke 10 days earlier. Please expand and clarify with more precise information.

L495, 496, "Fractions of this smoke plume must have reached the tropopause and later on the lowest stratosphere.": What does this mean? Why "must" the smoke have behaved like that? Was there a stratospheric, Arctic aerosol plume in Fall 2021 on par with the Raikoke/MOSAiC episode of 2019? Given the suggestiveness of the CALIOP observations in 2021 as compared to 2019 (the plume altitude and AOD are greater), one might hypothesize that a similarly strong and long-lasting plume was in evidence in 2021/22. Can the authors confirm such a plume? If so, that would be a powerful set of circumstances. If not, it would tend to shed doubt on the smoke composition of the 2019 Arctic stratospheric plume. The authors are asked to quantify the Fall/winter 2021 stratospheric aerosol load in comparison to 2019 and draw appropriate conclusions.

Figure 17. Unless the authors explain the rationale or algorithm for marking the "plume center height" in each panel, it is reasonable to conclude that the choices are arbitrary. If they have just been eyeballed, one could make an independent set of marks justifying no altitude gain; there are aerosols from the boundary layer to nearly 10 km in most panels. Moreover, the authors show one selected CALIOP curtain each day and thus don't account for the full synoptic view of the smoke on each date. A fuller synoptic view of the plumes on these dates, involving multiple CALIOP curtains per day and horizontal map views of the overall plume (using something like visible imagery, UV aerosol index, carbon monoxide) are essential for the reader to have confidence in the arguments brought here. And, as mentioned above, some traceability from 12 Aug 2019/4 Aug 2021 back to a definable source is also called for.

Figure 17a: The longitude "170" seems to be incorrect for this CALIPSO orbit. It should be ~150E.

L486, "The observation were taken over the East Siberian Sea north of Siberia (78°N, 160°E).": What observation? Do the authors mean "observations"? If so, what observations? 78N, 160E is not near the CALIPSO track, as far as I can tell. The orbit shown in Fig 17a is much farther west. On that track, the Ion at 78N is ~123E. The center of the thick smoke layer in 17a is north of 80N, but still far west of the trajectory start point. Please clear up the confusion. Figure 18: Why are the trajectories initialized at 07 UTC? This is neither close to the night or day orbits of CALIPSO as far as I can tell. Why not initialize them at the time of 12 Aug 2019, 4 Aug 2021 CALIPSO observations? If they were initialized at the CALIPSO observation time, that should be clarified in the caption. Assuming I have found the correct CALIPSO orbits to go with the two start points, the forward trajectories launched from them are significantly different than those shown. Please clarify or explain.

Figure 17a caption: Why are these heights (3, 5, 7 km) chosen? The initial layer is said to be at 3 km. Secondly, the displayed map legend says 3, 4, 5 km. The trajectories conform to these values. Are the heights in the caption the ones intended for display, or is the caption incorrect?

L489-497, Discussion of the 2021 smoke: There were 5 pyroCbs in the Sakha region of Siberia between 26 July and 7 August. No accounting of pyroCbs in 2021 is given by the authors. In addition to the Siberia pyroCbs, there were numerous pyroCbs in the USA and Canada in July and August. There was even a possible pyroCb in Greece in early August. Hence, there were multiple direct inputs of smoke into the UTLS that could have been transported to the Siberia zone in the time frame under study. It is therefore possible that some of the elevated smoke in Figure 17b is from pyroCb or pyroCu injections (when a pyroCb event occurs, naturally and frequently they are accompanied by numbers of pyroCu, which efficiently pollute the free troposphere. See Figure 7 of Fromm et al. (2021; https://doi.

org/10.1029/2021JD034928 ) and attendant discussion. The authors are asked to account for the added complexity and uncertainty regarding Figure 17 and associated analysis.

L487, "The black dots indicate the position of the smoke layer...": The smoke in each panel is seemingly much wider than the dots represent. A clearer explanation of the dot location choice is needed.

Figure 17, star symbol: What is the significance of the star symbol versus the dot? The star symbol is not defined in the text or figure caption.

L511-513 (End of Section 4.4), "We can conclude that even tropospheric smoke can ascend significantly from lower tropospheric injection heights up to the tropopause level within a few days and even enter the lower stratosphere as demonstrated in Ohneiser et al. (2021).": Once again, O21 did not "demonstrate" with observations that smoke self-lofted across the tropopause in amounts to justify their conclusions. They were only able to hypothesize this. In the current work, the observational aspect did not demonstrate the FTTS diabatic pathway, even if what they showed in Figure 17 was totally robust. No stepwise continuation of the smoke lofting to and through the tropopause was demonstrated. This should have been achievable given the enormous AODs and multi-season aerosol persistence O21 reported. Moreover, in all comparative respects, 2021 Siberian smoke equaled or exceeded 2019 in terms of AOD and height. Considering these two primary factors as inputs to the self-lofting model,

Fall and winter 2021's boreal stratosphere should embody a smoke plume at least on par with 2019. If that condition did not occur, there was effectively no observational demonstration of the FTTS diabatic lofting in either season. The authors are encouraged to assess that logic, dispute it, or consider a reinterpretation of their hypothetical model.

**Minor Issues**

Figure 1 (and lines 38-41): The images are offset by one day from those stated in the caption. The actual dates are 3-5 January, not 2-4 January. The thickest smoke on 3 and 4 Jan over the Tasman Sea is lower to mid-tropospheric. The UTLS smoke from the December phase of ANYSO pyroCb event had already blown east of New Zealand. The only substantial UTLS smoke in Fig. 1 is on 5 Jan. The arrows point to the plume from the 4 Jan pyroCb phase.

L42-46, discussion of smoke over cloud: The radiative implications of smoke over cloud are very complex. Referring to them in the context of Fig. 1 snapshots seems to draw relations that are likely to be irrelevant. Smoke plumes blown by fast UTLS winds will be almost totally decoupled from most cloud systems. There won't be much time for a cloud, small or big, to have an impact on a plume blowing faster than and independently of that cloud. Making the case for cloud effect on self-lofting calls for a much more sophisticated discussion.

L43, "compact layers": Please define "compact."

L67-68, discussion of dust self-lofting: The Daerden et al. paper is on Martian dust. Applicability to the Earth is unclear and not mentioned. Please elaborate or remove this citation. It was not evident to me that Gasteiger et al. discussed anything but vertical mixing, which is not synonymous with self-lofting. I could not find observational evidence therein of dust diabatic lofting. Please point it out if I missed it.

L187, "Siberian fire smoke reached the stratosphere via the self-lofting process.": As in other places in this manuscript, this is stated as fact. The authors hypothesized this in O21, and make statements elsewhere herein to the effect of "it must have happened," but definitive observational proof does not predate this paper. The authors are asked to modify the wording here and throughout to reflect the uncertainty of the FTTS self-lofting pathway.

L187-192: Citations are needed for the Kuwait- and Canadian-plume BC fraction. Same goes for the fuel types in Canada, Siberia, and Australia.

L193-195: "more homogeneous": Please define "homogeneous" since it is used as a comparator to pyroCb plumes.

L196, 197, discussion of pyroCb-plume depolarization ratio: A citation is needed for the "20%" value. Also, these double-digit depolarization ratios last far longer than the "fresh" stage. Please reword accordingly.

L271, "coherent structures": Please define "coherent."

L312-319, discussion of Kuwaiti oil-rig fire plumes: One of the prime science topics centered on these plumes was a validation of the Nuclear Winter hypothesis. Because of the size and darkness of these smoke plumes, it was thought that this might have been a natural (or at least inadvertent) experiment in self-lofting. Yet there are no publications to my knowledge that revealed significant self lofting. Rather, the consensus seemed to be that in spite of the absorption and optical depth of the plumes, little or no diabatic lofting was observed (Hobbs and Radke, 1992; Larry Radke, personal communication, 2008). Hence the strategic value of this case study is brought into question. The authors are asked to reflect on the published findings of observed Kuwaiti smoke heights as a function of time and make suitable revisions to the manuscript.

L446-454, Discussion of Figure 16 and MODIS AOT: This is a very interesting result. But the method needs to be explained better. Just how are the MODIS AOT data evaluated? What is done when there are no MODIS AOT retrievals due to clouds or smoke flagged as cloud? Given that MODIS data are daytime values, were CALIOP daytime curtains used for MODIS associations? And what are the implications of the results? Are the CALIOP-based AOT's low biased? Are the MODIS data a better approximation for the stratospheric plume AOT?

L471-476: The importance of this paragraph is not clear. The authors seem to be saying that if several sources of uncertainty all align in one direction, the overall error in final plume height can be very large. If that is a correct interpretation, isn't that self-evident? Please clarify.

L481, "Almost coherent smoke plume structures...": Assuming "coherent" was already defined, what is meant by "almost coherent?"

---

## Author Comment (AC1)

Dear Reviewer (Dear Mike)!

We thank You for careful reading and for the critical, constructive, fruitful comments and suggestions. After considering it all, we think that our revised manuscript now is in a good shape. We fully agree with your criticism concerning our tropospheric self-lofting part (observations vs simulations). The discussion was not convincing and straight forward. We also apologize that we obviously were not clear enough in our statements that smoke self-lofting in the troposphere remains a hypothesis. You are right! We did not observe self-lofting directly.

The topic we study in our manuscript is still a very new aspect in atmospheric science. Even if results of our measurements aimed to demonstrate that self-lofting played a major role in vertical smoke transport processes in the troposphere are not comprehensive enough, our simulations, the impact and uncertainty study (Sections 3.1-3.6), and the comparison of stratospheric CALIOP Canadian and Australian smoke observations with ECRAD simulations (Section 4 ) events are worth to be published and shared with scientific community. No doubt. On the other hand, we extensively improved the manuscript along your comments.

One of the main motivating points for the entire study was, that Raikoke sulfate aerosol alone cannot explain the observed AOTs of 0.1-0.2 (at 550 nm, in August 2019, as discussed in Ohneiser et al., 2021). The Raikoke-related AOT (according to the emitted SO2 mass) was of the order of <0.025 at northern latitudes. Impossible to have a Raikoke AOT of 0.05. Furthermore, pyroCb activity was low over Siberia in the summer 2019 (that was even mentioned in an e-mail by D. Peterson to us, when we asked him for his opinion about the pyroCb situation over Siberia in the summer of 2019). So, how to explain the found of massive pollution in the stratosphere, if Raikoke aerosol cannot explain it? That was the basic and main driving question for us to perform this detailed self-lofting study.

Back to the revision, as a consequence of all the shortcomings in the discussions in the submitted version, we completely removed those contents in the article that were dealing with the comparison of tropospheric observations with respective simulations. Now we present a very different discussion, based on new data and a new analysis strategy, that we believe is more convincing that tropospheric smoke self-lofting obviously occurred. This discussion is given in Section 5. We analyzed the entire CALIOP data set (day by day, scene by scene) from the Raikoke eruption on 21-22 June 2019 until the end of October 2019 to have an overlap with the MOSAiC Polarstern lidar observations that started in the end of September 2019. The focus is on smoke layers (pyroCb-related, self-lofting related) but also on features produced by Raikoke sulfate aerosol. We concentrated on the high northern latitudes >65°N. We found a number of fingerprints and arguments for self-lofting in the upper troposphere as discussed in Section 5, however, it remains a hypothesis that self-lofting really occurred. This is emphasized in the revised version several times.

Another point: New articles were published on wildfire smoke (Xian et al. 2022) and on stratospheric aerosol typing (Knepp et al., 2022, Boone et al., 2022) that forced us to consider them in our discussions. Especially the Boone et al. article contains a number of, to our opinion, misleading, incomplete, and thus incorrect conclusions on the aerosol mixture in the Arctic stratosphere so that we were forced to provide a short reply in Section 5, but also to write a comment article to Boone et al. article that we submitted to JGR on 25 October 2022.

The first part of the article (Sections 1-3), and also the discussion of comparisons of observations vs simulations for stratospheric ascending Canadian and Australian wildfire smoke layers (in Section 4) remained widely unchanged. Section 5 is, however, completely new.

Now to the item by item response:

Our responses are in BLUE.

All changes in our revised version of the manuscript is highlighted in BOLD.

Before we start, please note that the title of the article has been shortened and we added a co-author (Fabian Senf), an expert for atmospheric dynamics and modeling.

**Review of Ohneiser et al. (2022), "Self-lofting of wildfire smoke in the troposphere and stratosphere caused by radiative heating: simulations vs space lidar observations" Reviewer: Mike Fromm**

Ohneiser et al. explore the phenomenon of diabatic ascent of solar-radiation-absorbing smoke particles with a combination of satellite lidar- and visible-wavelength imager-based aerosol observations and modeling. The diabatic lofting in their focus is broken down into two categories: free-troposphere-to-stratosphere (hereafter "FTTS") and exclusively stratospheric. The latter of these two has been unequivocally observed in multiple case studies. The former (FTTS) has been hypothesized but not confirmed observationally.

To be more specific, in our revised version, the main goal is the ascent of smoke plumes and layers from typical injections heights of 2-6 km to the tropopause. Further stratospheric ascent is similar to pyroCb-related lofting events. PyroCb convection is usually restricted to heights below or around the tropopause, and only in the minority of cases, the smoke directly reached the lower stratosphere. Self-lofting from the tropopause towards greater heights is already well documented in many other papers (Kablick et al, Torres et al, and others, all cited in this article) and thus needs not to be investigated and highlighted in our manuscript.

It is the FTTS aspect that is the driver of this work, as revealed in the Abstract: "One of the main goals is to demonstrate that self-lofting processes can explain observed smoke lofting in the free middle and upper troposphere up to the tropopause and into the lower stratosphere without the need for pyrocumulonimbus convection." This is the logical extension of this author-group's prior publications that argued for non-pyroCb stratospheric smoke being the dominant particulate constituent in the Arctic in 2019-2020 (as opposed to Raikoke volcano sulfates, put forward in a host of additional recent publications). Hence, this manuscript attempts to test their hypothesis with observations and modelling support thereof.

Yes, after presenting all the pure simulation results (in Sections 2-3), the goal is to find convincing arguments, signatures, and fingerprints that indicate that self-lofting may have contributed to the smoke transport from the injection heights (2-6 km) up to the tropopause. This is the main goal of the article (and therefore main topic of the discussions in Section 5).

The authors' observational test includes not only the Raikoke summer of 2019 but an apparently similar scenario in August 2021 when Siberian fires also produced dense, synoptic- scale, multi-day smoke plumes. They focus on CALIOP backscatter curtains for a few selected days each year over the smoky domain. From those curtains the authors mark vertical locations representing a smoke-layer centroid that presumably represents a traceable ascent of the smoke they attribute to self-lofting. In addition to the CALIOP observations, the authors show HYSPLIT forward trajectories in association with the CALIOP plume heights over the multi-day plume evolution. Back trajectories, briefly described and unshown are also invoked to argue for diabatic ascent of the smoke layers.

This part was removed from our revised version. .

I found the entirety of the Siberia CALIOP and trajectory analysis (Section 4.4) to be confusing, arbitrary, speculative, error-prone, hence unconvincing. Below I will point out the various issues I found in greater detail. But in summary, my assessment is that the authors did not demonstrate observational evidence of diabatic self-lofting of Siberian tropospheric smoke in either 2019 or 2021. Absent that evidence, the reader has no basis on which to accept the modeling results presented herein and thus be convinced of the novel hypothesis presented in foundational work of Ohneiser et al. (2021; "O21").

We agree. And this comment motivated us to make a new effort to look at the CALIOP data again in this Raikoke summer 2019, but without trying to compare observed scenes directly with simulations. In Section 5 of our revised version, no simulations are shown.

Moreover, the authors clearly did not show incontrovertible observational proof of the FTTS diabatic self-lofting pathway required for the presumed stratospheric source term for the Arctic MOSAiC lidar-based conclusions of O21, on which Engelmann et al. (2021), Ansmann et al. (2021a), and Ohneiser et al. (2022) depended. From O21's Figure 13, one can project that the nascent, tropopause-level Siberian smoke resulting from FTTS transport would have an AOT of ~0.2 to 0.3 in August 2019. Layers embodying such an AOT would present as an unmistakable signal in CALIOP data. By their own analysis in Section 4.4 (Fig. 17), the Siberian smoke in 2019 had ascended only to 5 km (~5 km below the tropopause). Presumably, had there been detectable, efficient self-lofting to the tropopause, the authors certainly would have shown it. In the apparent absence of smoke FTTS proof, the authors were left to state: "Fractions of this smoke plume must have reached the tropopause and later on the lowest stratosphere." Thus, even if their observational interpretations are robust, the authors did not achieve one of their "main goals." The reader is left to rely on hypothetical modeling such as already presented in O21 and updated herein. From the Summary and Outlook section: "the large number of open parameters generates large uncertainty / sensitivity to the modeled lofting rates." Hence, we are no closer to a fire-emission/stratospheric-plume-pathway connection in boreal 2019 as before.

We agree. To our opinion, it is generally impossible to show 'incontrovertible observational proofs' in the case of slow tropospheric self-lofting processes, i.e., slow compared to these cases of explosion-like pyroCb events. As a consequence, we spent quite a long time to carefully check all CALIOP data, back and forth (mainly over the Arctic, >65°N), to find more solid, clearer, and more reasonable indications for self-lofting processes. These indications are described in Sections 5. One of the most interesting features (for us at least!) was the occurrence of a near-tropopause aerosol layer. This layer is in line with our simulations. There may be a number of other reasons that may also be used to explain such a layer, but the near-tropopause layer is predicted by the model in cases of long-lasting fire events, and that counts here for us. Nevertheless we clearly state that our argumentation is still a hypothesis, as all other explanations would also be hypotheses.

The authors' main goal was to support the contentions of O21, Engelmann et al. (2021), Ansmann et al. (2021a), and Ohneiser et al. (2022) that a boreal stratospheric smoke plume was present in 2019/2020 with AOT and persistence on par with certain volcanic injections and major pyroCb outbreaks. Their approach was to establish proof by observation in support of O21's hypothesis. My assessment is that the observational proof was not given in spite of expectations. If my assessment is accurate, the authors have strategically falsified the O21 hypothesis by looking for and not finding a suitable source term for the stratospheric smoke that O21 and associated papers proffered. This is a valid and valuable scientific conclusion. The current manuscript is thus positioned to challenge those previous conclusions. The authors are asked to consider using this current work to acknowledge the lack of observational evidence for Siberian smoke FTTS transport and the implications for O21, Engelmann et al. (2021), Ansmann et al. (2021a), Ohneiser et al. (2022), etc.

We agree. We believe that our new approach presented in Section 5 of our revised version of the manuscript is now more convincing. As mentioned already, the most important finding is the development of a diffuse tropopause aerosol layer that is fully in consistency with the self-lofting simulation study (Figure 7b), i.e., the decreasing ascent rate with height in the upper troposphere (and the minimum in the ascent rate profile at the tropopause). This leads to accumulation of smoke aerosol at the tropopause.

If my assessment is incorrect, it is incumbent on the authors to establish observational proof of the 2019 Siberian-smoke pathway to the stratosphere. The modeling work had essentially been done by O21; the modelling in this paper on its own does not serve as proof.

Here we partly disagree. The modeling work was NOT essentially done in Ohneiser et al. 2021. In that paper (as well as in Ansmann et al., 2021), we presented (some kind of) preliminary simulation results. We never presented the simulation tool itself. We were always planning to present the simulation scheme together with an extended uncertainty analysis in an independent article, i.e., in this article here. And even without a clear conclusion regarding the impact of smoke self-lofting in the troposphere, our manuscript presents so many new aspects and results from the simulation study. Even if we cannot present clear proofs, then the next generation of scientists can test our hypothesis, and work on the open questions. Otherwise, there is nothing to motivate future work in this field. Sure, others can use our work (published as ACPD version), but is that a correct way to push science forward? In this regard, we partially present the evidence and highlight what are gaps that still need to be studied in deep. All this can be found in the sections 4 and 5 and the summary section 6.

Regarding the observations and modeling the authors present on the second diabatic pathway, exclusively stratospheric, the simulations make a valuable contribution to the literature. As acknowledged by the authors, the stratospheric scenario is much cleaner and simpler to follow. Moreover, prior publications have already provided a foundation for tracking the smoke-plume features studied herein. Considering my concerns with the FTTS aspect of this manuscript and the implications for publication merit, I will focus my comments on that rather than the stratospheric part.

Thank YOU for this statement!

Major Concerns

L49, "Recently, smoke self-lofting was observed in the middle and upper troposphere (Ohneiser et al., 2021; Engelmann et al., 2021).": It is more accurate to say that self-lofting was inferred. Only a single CALIOP curtain was displayed by O21, giving a static view of vertical aerosol placement. Inferences were made based on Lagrangian trajectories and modeling.

We agree. We re-arranged the text to avoid such an impression.

L55-57, "…Siberia in the absence of pyroCb convection as spaceborne lidar observations indicated (CALIPSO, 2022).": How can CALIOP data be used to show there was no pyroCb action? The cited reference gives no help. In this Introduction section, it is presumed that any cited material will support the specific statement made. The reference here is a generic identifier of the CALIPSO data repository. In point of fact, there were 5 pyroCbs in the Sakha region of Siberia in July and August, 2021. PyroCb alerts and follow-up discussion in 2021 were shared in near real time on the Worldwide pyroCb Information Exchange (https://groups.io/g/pyrocb), for the specific purpose of engaging researchers across the globe and documenting individual pyroCb eruptions. The authors are asked to review the communication on this

open platform and revise their description of the 2021 boreal and Siberia pyroCb occurrence.

Thank You for these hints! We checked the communication and the different data sources. It remains to say, that such big pyroCb events as observed on 12 August 2017 (Canada, British Columbia) or in the beginning of 2020 (Australian fires) did not occur over Siberia in July and August 2019. And that was also mentioned in the paper of Knepp et al. (2022). Furthermore, the observed very low depolarization ratios support the absence of pyroCbs. In all cases with pyroCb-related smoke, the depolarization ratio was clearly enhanced (Canadian smoke, Australian smoke). Even in June and the beginning of July 2019, several pyroCb-related smoke layers were detected by CALIOP (probably formed over the North American continent), enhanced depolarization ratios were observed, indicating smoke lofted by pyroCbs.

L57, "Part of the smoke again entered the stratosphere in August 2021.": No doubt. One or more of the reported pyroCbs (see above) naturally injected smoke to or across the tropopause. Moreover, many other boreal pyroCbs were catalogued in 2021 (see the above link to the reporting medium). It is inappropriate for the Introduction to this paper to combine the claim of no pyroCbs and the supposed self-lofting from Siberia to the stratosphere.

We agree, and kept this in mind, when re-arranging the text. In 2021 (but also for 2019), there are contributions from both pyroCb-related lofting and smoke self-lofting.

L59-65, Discussion of smoke heating, buoyancy, self-lofting: It is important to treat the arguments of Boers et al., de Laat et al., Torres et al., and Yu et al. separately. Torres et al. and Yu et al. dealt explicitly with a puroCb-generated source term at the tropopause and primarily stratospheric transport whereas Boers et al. and de Laat et al. considered lofting of non- pyroCb-generated smoke in the lower to middle troposphere (analogous to this manuscript's Siberia hypothesis).  The distinction I call for is because stratospheric self lofting has been well characterized observationally (to which the Kablick, Khaykin, and other citations in this paragraph attest) whereas the FTTS pathway has not. It also must be noted that the Boers/de Laat pathway on Australia's Black Saturday has been reinterpreted as a classic pyroCb event, with rapid injection to the tropopause (Fromm et al., 2021, https://doi.org/10.1029/2021JD034928). Please revise the discussion here to reflect the disparate levels of certainty regarding pyroCb vs. non-pyroCb smoke pathways.

We agree (and thank you for the details!). We re-arranged the text accordingly, and separated the approach of Boers et al. and deLaat et al. from the one presented in the articles of Torres et al.  and Yu et al. (see Section 1).

L199, "Chemical aging is assumed to lead to a spherical shape…": On what basis is this assumed? Please defend this statement. There is abundant literature showing that aged tropospheric smoke particles embody depol. ratios that are significantly greater than the limiting value for uniform spheres, e.g. Dahlkötter et al (2014; doi:10.5194/acp-14-6111-2014), Burton et al. (2015; doi:10.5194/acp-15-13453-2015), Hu et al. (2022; https://doi.org/10.5194/acp-22-5399-2022), Liu et al. (2022, https://doi.org/10.1016/j.jqsrt.2022.108080). Hence, Table 2's "2%" "spherical" characterization is apparently an inaccurate simplification of non-pyroCb, tropospheric smoke.

First of all, we removed Table 2.

As requested we extend the explanations on this (in Section 3), i.e., regarding particle aging, aging periods, and the consequences for particle shape and observed depolarization ratios. However, as experimentally working lidar scientists, we must clearly say: If we measure depolarization ratios of 2-3% then the particles MUST be spherical. Even small deviations from the spherical form will already introduce a significant jump

in the depolarization ratio. Furthermore, real world lidar observations can never deliver 0% depolarization ratio which should theoretically be measured in the case of ideal spheres and in total absence of any multiple scattering effect. There is always signal noise and the remaining uncertainties in all the channel calibration efforts make it impossible to have at the end 0% for the particle depolarization ratio. Rayleigh scattering already introduces 1-2% depolarization (that must be corrected).

Please revise to reflect the literature. Since the weight of the above-mentioned papers rests on the idea that aged smoke is better represented by an aspherical model than monolithic spheres, the implications of changing Table 2 are momentous. The authors are asked to reconsider their assumptions for aged tropospheric smoke and the implications for the optical properties they observed in the stratosphere during MOSAiC.

We disagree. If aspherical particles are present, then they cannot be aged, they are fresh particles. These fresh particles, producing enhanced depolarization ratios, had not enough time to age and to develop a compact spherical shape. We discussed all this already in many papers (Haarig et al. 2018, Ohneiser et al., 2020, Ansmann et al. 2021, review on smoke optical and microphysical properties in ACP). With time, all smoke particles will end up as spheres when the aging process is completed (after days in the humid troposphere, or weeks to months in the dry upper troposphere and especially in the stratosphere, as Baars et al. (2019) showed). We think the improved discussion in Section 3 will help to better understand these differences of aged and fresh smoke particles and the consequences for measured depolarization ratios.

Regarding the concern above regarding the authors' assumption of spherical, aged smoke in Siberia (and eventually in the stratosphere), it is important to note that the only other published work describing non-pyroCb, FTTS transport is de Laat et al., (2012). Even if it is maintained that their "solar escalator" was an accurate depiction, the resultant stratospheric smoke from Black Saturday had depolarization ratios weeks after onset so great that the CALIOP observations thereof triggered a feature classification of "ice" ( https://acp.copernicus.org/preprints/acp-2021-117/acp-2021-117-RR1.pdf). Hence these aged smokes were wholly inconsistent with the assumptions shown in Table 2. The authors are asked to comment on this inconsistency and make the appropriate revisions to their current assumptions on smoke aging.

We avoided to discuss the deLaat results in this lofting paper, to keep the discussions as short as possible. And we removed Table 2. Regarding the Black Saturday smoke: The depolarization ratio for the aerosol above 20 km is clearly enhanced. That means the particles were non-spherical. If self-lofting in the free troposphere was involved then the upper troposphere must have been super dry and condensable gases were not available so that particles could not age quickly and could not develop a compact and spherical. To our opinion, the detected Black Saturday smoke above 20 km, showing enhanced depolarization, was lofted by pyroCbs.

Section 4.4. This section is problematic. My concerns are many and deep. This part of the paper is seriously flawed. Below I list the issues individually.

We totally agree with you, it was our fault to present it. We apologize for that. We removed the content of Section 4.4. Our revised version now contains a new section 5 with a discussion on potentially observed fingerprints for tropospheric self-lofting of smoke over central eastern Siberia and the adjacent Arctic in the summer of 2019 based on CALIOP data (June to October 2019).

L482, "The smoke at different heights was transported in the same air column.": Meteorologically, I cannot make sense of this. What does "same air column" mean? The trajectories take different paths and have

very different endpoints. This implies considerable speed and directional shear, which by itself seems to suggest a variable air column. Please explain.

As mentioned, all this is removed now.

Discussion of Figures 17 and 18: How "old" are the smoke layers on 12 August 2019 and 4 August 2021? Without knowing that, the reader has no idea how all those smokes got to that start point. Almost assuredly they didn't all arrive in a single "air column," but no information is given. I.e. these seem like arbitrary start points and hence it is unclear how to assess transport after the start points in relation to transport to them.

As mentioned, all this is removed now.

L482, 483, "The trajectories analysis in Fig. 18a indicated similar wind speed and directions at different heights in the middle troposphere.": I cannot reconcile Fig. 18a with this description. There are significant differences in both aspects. If the authors want to convince the reader that these trajectories are similar, they need to quantify that and show what dissimilar trajectories would look like.

It was excluded from our revised version.

L483, 484, "The smoke slowly ascended from around 3-4 km height to 5-6 km height.": The trajectories themselves account for almost a 1-km altitude gain. I.e. resolvable meteorology is responsible, not diabatic lofting. The authors need to acknowledge this.

As mentioned, we removed this section.

L484, 485, "The backward trajectories (not shown) indicate no direct link of the air mass at 5-6 km height with lower heights (and thus a direct fire smoke uptake from the sources)." For this to make sense, the trajectories must be shown. Even so, it is difficult to understand what the authors mean here. What is "direct…uptake"? What are "the sources"? Either remove this sentence or bolster it with displayed trajectories and precise terminology.

As mentioned, it was removed.

L489-494, Nearly identical trajectory discussion regarding the 2021 smoke: I have the same concerns as stated above. Please take the same corrective actions.

As mentioned, it was removed.
However, we present new analyses in Section 5.1, based on a CALIOP observation in August 2021. We combined subsequent CALIOP observations and backward and forward trajectories to investigate self-lofting effects.

L494, "Initial smoke plumes were at around 3 km height on 4 August 2021.": What is meant by "initial" when the previous sentence refers to smoke 10 days earlier. Please expand and clarify with more precise information.

As mentioned, all this is removed now.

L495, 496, "Fractions of this smoke plume must have reached the tropopause and later on the lowest stratosphere.": What does this mean? Why "must" the smoke have behaved like that? Was there a

stratospheric, Arctic aerosol plume in Fall 2021 on par with the Raikoke/MOSAiC episode of 2019? Given the suggestiveness of the CALIOP observations in 2021 as compared to 2019 (the plume altitude and AOD are greater), one might hypothesize that a similarly strong and long-lasting plume was in evidence in 2021/22. Can the authors confirm such a plume? If so, that would be a powerful set of circumstances. If not, it would tend to shed doubt on the smoke composition of the 2019 Arctic stratospheric plume. The authors are asked to quantify the Fall/winter 2021 stratospheric aerosol load in comparison to 2019 and draw appropriate conclusions.

As mentioned, we exclude this discussion. Concerning the CALIOP observations in 2021. The troposphere was filled with smoke up to the tropopause over the Arctic, especially in the second half of August. The layering structures were different compared to 2019. Such an (isolated) diffuse near-tropopause layer was not found in 2021. Instead, often the entire troposphere between 5km and the tropopause was filled with smoke over a large latitudinal range (65-80°N) so that an extra layer around the tropopause could not be resolved. Sometimes the depolarization ratio was enhanced and may indicate pyroCb lofting, sometimes the depolarization ratio was close to zero.

It is not only a question of the aerosol conditions. Self-lofting needs probably also favorable stagnant conditions… and the meteorological conditions always vary and are never exactly the same from year to year during the burning season.

Figure 17. Unless the authors explain the rationale or algorithm for marking the "plume center height" in each panel, it is reasonable to conclude that the choices are arbitrary. If they have just been eyeballed, one could make an independent set of marks justifying no altitude gain; there are aerosols from the boundary layer to nearly 10 km in most panels. Moreover, the authors show one selected CALIOP curtain each day and thus don't account for the full synoptic view of the smoke on each date. A fuller synoptic view of the plumes on these dates, involving multiple CALIOP curtains per day and horizontal map views of the overall plume (using something like visible imagery, UV aerosol index, carbon monoxide) are essential for the reader to have confidence in the arguments brought here. And, as mentioned above, some traceability from 12 Aug 2019/4 Aug 2021 back to a definable source is also called for.

Section 4.4, including Figure 17, is removed.

Figure 17a:  The longitude "170" seems to be incorrect for this CALIPSO orbit. It should be ~150E.

Section 4.4, including Figure 17, is removed.

L486, "The observation were taken over the East Siberian Sea north of Siberia (78°N, 160°E).": What observation? Do the authors mean "observations"? If so, what observations? 78N, 160E is not near the CALIPSO track, as far as I can tell.  The orbit shown in Fig 17a is much farther west. On that track, the lon at 78N is ~123E. The center of the thick smoke layer in 17a is north of 80N, but still far west of the trajectory start point. Please clear up the confusion.

Section 4.4, including Figure 17, is removed.

Figure 18: Why are the trajectories initialized at 07 UTC? This is neither close to the night or day orbits of CALIPSO as far as I can tell. Why not initialize them at the time of 12 Aug 2019, 4 Aug 2021 CALIPSO observations? If they were initialized at the CALIPSO observation time, that should be clarified in the caption. Assuming I have found the correct CALIPSO orbits to go with the two start points, the forward trajectories launched from them are significantly different than those shown. Please clarify or explain.

Section 4.4, including Figure 18. is removed.

Figure 17a caption: Why are these heights (3, 5, 7 km) chosen? The initial layer is said to be at 3 km. Secondly, the displayed map legend says 3, 4, 5 km. The trajectories conform to these values. Are the heights in the caption the ones intended for display, or is the caption incorrect?

Section 4.4, including Figure 17, is removed.

L489-497, Discussion of the 2021 smoke: There were 5 pyroCbs in the Sakha region of Siberia between 26 July and 7 August. No accounting of pyroCbs in 2021 is given by the authors. In addition to the Siberia pyroCbs, there were numerous pyroCbs in the USA and Canada in July and August. There was even a possible pyroCb in Greece in early August. Hence, there were multiple direct inputs of smoke into the UTLS that could have been transported to the Siberia zone in the time frame under study. It is therefore possible that some of the elevated smoke in Figure 17b is from pyroCb or pyroCu injections (when a pyroCb event occurs, naturally and frequently they are accompanied by numbers of pyroCu, which efficiently pollute the free troposphere. See Figure 7 of Fromm et al. (2021; https://doi.org/10.1029/2021JD034928) and attendant discussion. The authors are asked to account for the added complexity and uncertainty regarding Figure 17 and associated analysis.

Section 4.4, including Figure 17, is removed.

Nevertheless, in the new Section 5.1, we show one CALIOP smoke case and then discuss briefly the CALIOP smoke profile observations in August 2021. It is mentioned that pyroCb-related lofting was involved in the smoke transport.

L487, "The black dots indicate the position of the smoke layer…": The smoke in each panel is seemingly much wider than the dots represent. A clearer explanation of the dot location choice is needed.

Section 4.4, including Figure 17, is removed.

Figure 17, star symbol: What is the significance of the star symbol versus the dot? The star symbol is not defined in the text or figure caption.

Section 4.4, including Figure 17, is removed.

L511-513 (End of Section 4.4), "We can conclude that even tropospheric smoke can ascend significantly from lower tropospheric injection heights up to the tropopause level within a few days and even enter the lower stratosphere as demonstrated in Ohneiser et al. (2021).": Once again, O21 did not "demonstrate" with observations that smoke self-lofted across the tropopause in amounts to justify their conclusions. They were only able to hypothesize this. In the current work, the observational aspect did not demonstrate the FTTS diabatic pathway, even if what they showed in Figure 17 was totally robust. No stepwise continuation of the smoke lofting to and through the tropopause was demonstrated. This should have been achievable given the enormous AODs and multi-season aerosol persistence O21 reported. Moreover, in all comparative respects, 2021 Siberian smoke equaled or exceeded 2019 in terms of AOD and height.

Yes, we agree. We did not observe self-lofting. All the CALIOP data analysis (even in the new Section 5) does not allow us to state: We found a proof for smoke self-lofting. NO! In our revised version of the manuscript, we carefully and critically present our results.

Considering these two primary factors as inputs to the self-lofting model, Fall and winter 2021's boreal stratosphere should embody a smoke plume at least on par with 2019. If that condition did not occur, there was effectively no observational demonstration of the FTTS diabatic lofting in either season. The

authors are encouraged to assess that logic, dispute it, or consider a reinterpretation of their hypothetical model.

As mentioned, one cannot expect to see the same evolution of smoke layers in the upper troposphere and lower stratosphere in 2019 and 2021…. Meteorological conditions are never totally equal. We see smoke in the CALIOP data up to the tropopause or better around the tropopause in both years. CALIOP observation could not be used to see diffuse thin aerosol layers higher up in both years. Only in cases with pyroCb-lofted smoke one may have a good chance to see compact, isolated ascending plumes producing considerably high backscatter together with enhanced depolarization, as it was the case for the record-breaking Canadian and Australian smoke events.

The goal of our manuscript is not to illuminate in detail the wildfire and smoke situation in 2021. The goal is to discuss the 2019 smoke over the Arctic.

Minor Issues

Figure 1 (and lines 38-41): The images are offset by one day from those stated in the caption. The actual dates are 3-5 January, not 2-4 January. The thickest smoke on 3 and 4 Jan over the Tasman Sea is lower to mid-tropospheric. The UTLS smoke from the December phase of ANYSO pyroCb event had already blown east of New Zealand. The only substantial UTLS smoke in Fig. 1 is on 5 Jan. The arrows point to the plume from the 4 Jan pyroCb phase.

We improved Figure 1, and also rephrased the text in the introduction. But the message is that self-lofting can immediately produce very complex layering structures. The yellow arrows shall guide the reader to see the smoke. Many scientists may have no idea how to distinguish smoke and clouds.

L42-46, discussion of smoke over cloud: The radiative implications of smoke over cloud are very complex. Referring to them in the context of Fig. 1 snapshots seems to draw relations that are likely to be irrelevant. Smoke plumes blown by fast UTLS winds will be almost totally decoupled from most cloud systems. There won't be much time for a cloud, small or big, to have an impact on a plume blowing faster than and independently of that cloud. Making the case for cloud effect on self-lofting calls for a much more sophisticated discussion.

We re-arranged the text to better explain the cloud impact. The impact is via changed albedo conditions. As soon as the smoke is above a white cloud layer, there is almost a factor of 2 more solar radiation available for absorption.

L43, "compact layers": Please define "compact."

We changed the text (left out: compact). Words need to be self-explaining. Otherwise, should be better not used.

L67-68, discussion of dust self-lofting: The Daerden et al. paper is on Martian dust. Applicability to the Earth is unclear and not mentioned. Please elaborate or remove this citation. It was not evident to me that Gasteiger et al. discussed anything but vertical mixing, which is not synonymous with self-lofting. I could not find observational evidence therein of dust diabatic lofting. Please point it out if I missed it.

We changed the text and removed this point.

L187, "Siberian fire smoke reached the stratosphere via the self-lofting process.": As in other places in this manuscript, this is stated as fact. The authors hypothesized this in O21, and make statements elsewhere

herein to the effect of "it must have happened," but definitive observational proof does not predate this paper. The authors are asked to modify the wording here and throughout to reflect the uncertainty of the FTTS self-lofting pathway.

We agree! We carefully went through the text to emphasize: All our studies may indicate self-lofting effects, but it remains a hypothesis that self-lofting contributed to the smoke vertical transport.

L187-192: Citations are needed for the Kuwait- and Canadian-plume BC fraction. Same goes for the fuel types in Canada, Siberia, and Australia.

We improved Kuwait-fire-related information and references in the Introduction, in Section 3, and 3.5. Concerning fuel types that is, to our opinion, google-like or text-book-like knowledge, however, we cite Ohneiser et al. 2022 here. Regarding BC fractions we also give references now. All this is given in Section 3.

L193-195: "more homogeneous": Please define "homogeneous" since it is used as a comparator to pyroCb plumes.

Done, it was rephrased! Avoid to use 'more homogeneous'.

L196, 197, discussion of pyroCb-plume depolarization ratio: A citation is needed for the "20%" value. Also, these double-digit depolarization ratios last far longer than the "fresh" stage. Please reword accordingly.

Reference is now given!

L271, "coherent structures": Please define "coherent."

We do not mention 'coherent structures' anymore.

L312-319, discussion of Kuwaiti oil-rig fire plumes: One of the prime science topics centered on these plumes was a validation of the Nuclear Winter hypothesis. Because of the size and darkness of these smoke plumes, it was thought that this might have been a natural (or at least inadvertent) experiment in self-lofting. Yet there are no publications to my knowledge that revealed significant self lofting. Rather, the consensus seemed to be that in spite of the absorption and optical depth of the plumes, little or no diabatic lofting was observed (Hobbs and Radke, 1992; Larry Radke, personal communication, 2008). Hence the strategic value of this case study is brought into question. The authors are asked to reflect on the published findings of observed Kuwaiti smoke heights as a function of time and make suitable revisions to the manuscript.

We checked the literature comprehensively. Limaye et al. (1991) observed lofting. They found plumes at 6-7 km height in a distance 2000 km away from the sources. Close to the sources the plumes were below 3 km. We mention that in the introduction and in Section 3.5.

L446-454, Discussion of Figure 16 and MODIS AOT: This is a very interesting result. But the method needs to be explained better. Just how are the MODIS AOT data evaluated? What is done when there are no MODIS AOT retrievals due to clouds or smoke flagged as cloud? Given that MODIS data are daytime values, were CALIOP daytime curtains used for MODIS associations? And what are the implications of the results? Are the CALIOP-based AOT's low biased? Are the MODIS data a better approximation for the stratospheric plume AOT?

We describe how we used the CALIOP and MODIS data to obtain our parameterization (Section 4.1). When

there are no MODIS pixels left after cloud screening then there are no MODIS AOTs. It is not necessary to have rather exact AOT values and day by day AOTs to obtain the AOT parameterizations used.

L471-476: The importance of this paragraph is not clear. The authors seem to be saying that if several sources of uncertainty all align in one direction, the overall error in final plume height can be very large. If that is a correct interpretation, isn't that self-evident? Please clarify.

We removed this paragraph to avoid confusion and to keep the discussion short here.

By the way we changed the figures (now we show more compact Figures 12 and 13 in Section 4). We reduced in this way the number of figures in Section 4.

L481, "Almost coherent smoke plume structures…": Assuming "coherent" was already defined, what is meant by "almost coherent?"

We removed such statements.

To sum up:

All in all, we think that the manuscript is in a good shape now. We also believe that we could convincingly present a number of arguments that the 2019-2020 aerosol conditions as observed with CALIOP and the MOSAIC Polarstern lidar in the UTLS height range point to smoke self-lofting as a significant process to transport smoke upward into the tropopause region over high northern latitudes in the summer of 2019.

The key findings concerning tropospheric self-lofting over Siberia in 2019 were summarized in Section 5 as follows:

We found to our opinion clear evidence supporting our hypothesis that self-lofting processes played a key role in the formation of a smoke-dominated UTLS smoke layer. First of all, the observed AOTs were a factor of at least 5 higher AOT than expected from the Raikoke $SO_2$ emission, so there were additional aerosol particles in the UTLS height range besides the Raikoke sulfate particles. Typical optical fingerprints for aged smoke particles were found from the polarization Raman lidar observations. The inverse spectral slope of the lidar ratio and the high 532~nm lidar ratios are a clear and unique sign for the dominance of smoke in the UTLS aerosol layer. No other aerosol type was ever observed with lidars showing such a spectral behavior of the lidar ratio. The observed low particle depolarization ratios clearly show that lofting by pyroCb convection played no or only a minor role. The occurrence of pyroCb-related lofted smoke was always found in published lidar observations to be associated with enhanced depolarization ratios caused by irregularly-shaped, fastly ascending smoke particles. Then, only self-lofting is left to explain the built-up of the optically relatively dense, long-lasting smoke layer observed over the Arctic for almost a year until May 2020. The low depolarization values pointing in addition to relatively low ascent rates in the troposphere so that the smoke particles had sufficient time to age and to develop a compact and spherical core-shell structure. This feature is expected according to our simulations when self-lofting comes into play. Finally, the formation of a near tropopause smoke layer is also in line with the simulations suggesting or predicting an accumulation of upward moving particles around the tropopause because of the height-dependent ascent rate (with minimum at the tropopause). However, it remains to be emphasized that all these arguments can not be used as a proof that self-lofting processes really occurred and triggered significant upward motion of smoke-filled tropospheric air parcels. It remains a hypothesis.

Our final remark in this reply letter:

There is no reason anymore to reject such a paper that will trigger future work concerning the role of self-lofting of light-absorbing aerosols in the atmosphere. There is so much smoke around the world (central and southern Africa, in South America, western North America, western to eastern Mediterranean, Middle East and central Asia, Southeast Asia, Australia, Alaska, Canada, Siberia), that we believe, self-lofting plays a role on a global scale. However, it remains rather difficult to provide clear observation of slowly ascending of likewise thin aerosol layers. Self-lofting leads to a prolongation of the lifetime of all these light-absorbing particles in the atmosphere and this aspect is not considered in any of the numerous climate models used to predict climate change.

Finally, we should add that we presently have to review a JGR manuscript on global aspects of ascending BC-containing aerosol layers and these authors find similar results concerning self lofting in the troposphere and stratosphere as we discuss in our manuscript. One highlight is that the used Earth System Model indicates a strong increase in UTLS smoke mass concentrations (by 50%, annual mean increase) in the 8-22 km height range of the northern part of the Northern Hemisphere when BC absorption of solar radiation is considered in the simulations. As these authors pointed out in the introduction, their JGR article was motivated by our self-lofting studies presented in Ohneiser et al. (2021) on MOSAiC observations and by this ACPD manuscript (Ohneiser et al., 2022), and by other recent papers dealing with self lofting processes.

---

## Author Comment (AC2)

Dear Reviewer!

We thank You for reading the manuscript and for your comments and remarks! We agree with your criticism concerning our tropospheric self-lofting part (observations vs simulations). The discussion was not convincing and straight forward.

As a consequence, we re-wrote the entire section on tropospheric self-lofting of smoke (now section 5, section 4.4 in the submitted version). We skipped the simulation part in Sect. 5. CALIOP observations (and one Leipzig lidar observation) are presented only in this revised section. We re-analyzed the entire CALIOP data set over central eastern Siberia and the Arctic from 22 June 2019 (at that day the Raikoke volcano erupted) until October 2019 (to have an overlap with the MOSAiC Polarstern lidar observations that started in the end of September 2019). Together with new literature on stratospheric smoke (Knepp et al., 2022, Xian et al. 2022), corroborating our hypothetical approach, we believe that we have now several indications that self-lofting contributed to the upward transport of smoke within the troposphere. However, we agree, it remains a hypothesis! And that is emphasized several times in the revised manuscript.

However, the topic is a very new aspect in atmospheric science that clearly deserves publication. Even if we are in trouble to demonstrate that self-lofting played a major role in vertical smoke transport processes in the troposphere, our simulations, the impact study and uncertainty analysis, and the applications to Canadian and Australian fire events are worth enough to be published. No doubt!

The first part of the article (Sections 1-3), and also the discussion of comparisons of observations vs simulations for stratospheric ascending Canadian and Australian wildfire smoke layers (in Section 4) remained widely unchanged with respect to the submitted version.

Now to the item by item response:

Our answers are in BLUE!

Significantly changed text or added text is given in BOLD in the revised version of the manuscript.

Before we start, please note that the title of the article has been shortened and we added a co-author (Fabian Senf), he is an expert for atmospheric dynamics and modeling.

The manuscript by Kevin Ohneiser and coauthors addresses the solar-driven lofting of wildfire smoke plumes in the troposphere and stratosphere using ECMWF radiation transfer scheme with different parameterizations and satellite observations using CALIOP and MODIS instruments. The ascent rates of smoke plumes produced by Canadian, Australian and Siberian wildfires derived from CALIOP observations are compared with the calculated ascent rates from radiative transfer simulations. The main goal of the study, as stated in the abstract, is to demonstrate that the radiative heating of intense smoke plumes is capable of lofting them from the free troposphere up to the tropopause and into the stratosphere without the need of PyroCb injections.

We changed a bit the main goal. The main focus is now on tropospheric self-lofting only, i.e., from the injection heights of 2-6 km to the tropopause. Lofting of smoke from the tropopause to greater heights within the stratosphere is already well documented and well described in the literature (as described in the introduction).

After a detailed description of the modeling setup, sensitivity tests and uncertainty discussion, the authors demonstrate in Fig. 11 that a 2.5 km- thick smoke plume with a realistic BC fraction of 2.5% and a very large AOT above 2 should rise from 3 km altitude into the lower stratosphere in two weeks. However, the analysis of CALIOP observations of tropospheric smoke from Siberian wildfires in the following section does not provide any support for the cross-tropopause transport of aerosol plumes rendering the main goal of the study unachieved and casting doubt on the usefulness of the simulation results.

Yes, this is true! Therefore, we changed the strategy of our study. As mentioned above, the study is now based on the CALIOP data set (day by day, scene by scene) from the Raikoke eruption on 21-22 June 2019 until the end of October 2019 with focus on smoke layers (pyroCb-related, self-lofting related) but also on features produced by Raikoke sulfate aerosol. We concentrated on the high

northern latitudes >65°N. We found a number of fingerprints and arguments for self-lofting in the upper troposphere in line with our simulations. But we do not show any simulation here. Our 1D simulation model cannot be used to simulate 3D-air motions in the troposphere. We mention that in Section 3.4. However, even after listening a number of convincing arguments in Section 5, it remains a hypothesis that self-lofting really occurred. This is emphasized frequently!

More specifically, there are several major issues as follows.

The description of the satellite instruments, data versioning, measurement uncertainties and the approach to data treatment is totally missing in the manuscript.

We agree! However, we exactly provide similar information on CALIOP and MODIS and other instruments, data use, etc. as in our foregoing papers on Arctic smoke (Ohneiser et al., ACP, 2021) and Australian smoke (Ohneiser et al, 2022). And that was acceptable. CALIOP and MODIS are so well known that such a detailed introduction of these instruments (including, data handling, quality levels, uncertainty analysis, etc) is no longer needed to our opinion. And all necessary information, needed in this self-lofting paper, is given in Section 4.1, i.e., how we got the layer center heights (in the case of the presented Canadian and Australian smoke layers) and how we got the AOT values from CALIOP observations and (now also in more detail, from MODIS data).

As far as I understood, the authors used CALIOP quicklooks to derive the layer thickness and mean attenuated backscatter, from which the AOT is calculated using an arbitrarily chosen factor of 1.5, which should account for the light attenuation. While the derivation of layer thickness from the quicklooks may be deemed sufficiently accurate (although for compact stratospheric plumes only), it is unclear how the authors derived the layer mean backscatter from the images. Was it done by reading the colors of each individual pixels and using the color bar to retrieve the values? If so, the uncertainty of such estimates might be unacceptably high and I wonder how such estimates would compare with those by Kablick et al. provided in Fig. 16.

The factor of 1.5 accounts for the probably too low lidar ratio of 65 sr (the smoke lidar ratio in the CALIOP data base), because we observed lidar ratios of 90-100 sr for the Australian smoke (Ohneiser et al., 2022). This is now correctly mentioned in the revised version. This was not well described in the submitted version.
The use of quicklooks of attenuated backscatter (color plots) or the use of downloaded particle backscatter values doesn't matter much. At the end, the obtained AOT values may differ by about 5-10%. That is our experience. Kablick also used backscatter coefficients, multiplied by the smoke lidar ratio, and then integrated the obtained extinction values from layer base to top.

Constrained by the CALIOP AOT from Kablick et al., the simulated ascent is nowhere near the observed one and the authors opt to constrain the simulation with MODIS total AOT data (ignoring the tropospheric aerosols), which is substantially higher than both the CALIOP-derived AOT and, what is particularly puzzling, much higher than the estimates by Ohneiser et al. (2020), their Fig. 5b, reporting the lidar-derived AOT@532 of 0.1 – 0.3 for the Australian smoke plume in late January 2020 (which would be consistent with Kablick et al. data in Fig. 16). The authors thus seem to deliberately ignore their own observations for the sake of reproducing the observed lofting in the simulation.

The observations in Ohneiser et al. (2020) were done at the edge of the smoke-filled vortex and the AOT values were therefore smaller. The MODIS data overestimate the UTLS AOTs before 17 January (this is shown now in the new, more compact Figure 13) and after 17 January the values became reasonably small (after subtracting a typical marine AOT value). Then, the CALIOP AOTs of Kablick et al. and the MODIS AOTs compared reasonably well.
In the revised version, we mention that a tropospheric marine background AOT of 0.05 was subtracted from the observed total smoke MODIS AOT. The tropospheric AOT contribution was cleatly underestimated until 16 January when using a tropospheric AOT of 0.05.
As a consequence, the simulations based on MODIS AOTs start on 17 January 2020.

Section 4.4 and Fig. 17. The Siberian tropospheric smoke plumes show rather complex vertical

structures, whereas the determination of the aerosol layer vertical boundaries (critically influencing the AOT estimate) appear to be somewhat too arbitrary. Personally, I do not see any significant lofting for the both cases shown in Fig. 17. It rather appears that the smoke was found in the UT from the very beginning, which would point to the PyroCb-driven vertical transport.

Section 4.4 including Figure 17 is removed now, and substituted by the new section 5.

Discussing the Siberian smoke plumes, the authors state that "fractions of this plume must have reached the tropopause and later on the lower stratosphere" without providing any supporting observations, and the only reason I can possibly think of is the absence of such observational evidence. Moreover, the simulations based on an assumption of the persistent Gaussian vertical shape of the layer (which is obviously not the case here) show even weaker lofting than what is inferred from CALIOP quicklooks. I also wonder why the simulation was not extended further in time (using, e.g. 15% daily AOT decrease) to provide at least the modeling support for the potential lofting up to the tropopause level.

Again, Section 4.4 is removed, and substituted by the new section 5. It makes no sense to compare simulations with observations in the convective and turbulent troposphere. We explain why our simple ECRAD model cannot be used to simulate 3D-air motions in the troposphere in Section 3.4.

The assumption of the real shape structures or the assumption of a Gaussian shape profile almost lead to the same results.

All in all, we think that the manuscript is in a good shape now. We also believe that we could convincingly show that the 2019-2020 aerosol conditions as observed with CALIOP and the MOSAIC Polarstern lidar in the UTLS height range can only be reasonably explained by considering smoke self lofting as a significant process to transport smoke into the tropopause region. To summarize again the main 3 arguments that point to a strong role of smoke self-lofting are:

(a) The observed UTLS AOT (caused by smoke and sulfate) was much higher (0.1-0.2 at 532 nm) than the expected sulfate AOT (<0.025) originating from the Raikoke eruption. However, the observed low depolarization ratios pointed to a minor impact of pyroCb-related smoke lofting. Consequently, another process is needed to transport smoke upward, towards the tropopause.

(b) By carefully expecting all CALIOP observations at high northern latitudes from June to October 2019, we detected a near-tropopause aerosol layer, clearly visible from mid July to mid of August (in the period of strongest fires over Siberia). The occurrence of this layer is in line with one of the main simulation results. The lofting rate decreases with height in the upper troposphere, and shows a minimum at the tropopause. In case of a steady upward transport of smoke this leads to smoke accumulation at the tropopause.

(c) The found lidar ratio characteristics (355 nm lidar ratio is much smaller than the 532 nm lidar ratio, lidar ratio at 532 is typically larger than 70 sr indicating strongly absorbing aerosol particles) unambiguously points to smoke as the dominating aerosol type in the UTLS aerosol layer. Such a unique spectral slope of the lidar ratio has never been observed by the lidar community for any other aerosol type.

Finally, we should add that we presently have to review a JGR manuscript on global aspects of ascending BC-containing aerosol layers and these authors find similar results concerning self lofting in the troposphere and stratosphere as we discuss in our manuscript. One highlight is that the used Earth System Model indicates a strong increase in UTLS smoke mass concentrations (by 50%, annual mean increase) in the 8-22 km height range of the northern part of the Northern Hemisphere when BC absorption of solar radiation is considered in the simulations. As these authors pointed out in the introduction, their JGR article was motivated by our self-lofting studies presented in Ohneiser et al. (2021) on MOSAiC observations and by this ACPD manuscript (Ohneiser et al., 2022), and by other recent papers dealing with self lofting processes.

---

## Referee Report (RR1)

**Review of Ohneiser et al. (2022), "Self-lofting of wildfire smoke in the troposphere and stratosphere: simulations and space lidar observations"**

Reviewed by Mike Fromm

Note: "O22" is shorthand for the author group of this manuscript. "O21" is used to refer to Ohneiser et al. (ACP, 2021).

This manuscript is a first revision.

**Assessment Overview**

This review was hampered by the fact that the author tracked change (ATC) document is inaccurate. For example, the paragraph on lines 204-213 is not in the original manuscript. Yet it is un-highlighted in the ATC document. Hence, I abandoned my attempt to use the ATC document.

O22 have responded to both reviewers' comments and substantially modified the manuscript. My assessment is that their responses to my deepest concerns were inadequate and the revised manuscript is as flawed as the original. Their crucial new section, replacing the flawed original section, abounds with demonstrable inaccuracies, misinterpretations of satellite data, conflicting messaging, and unmet expectations from their new, season-long CALIOP analysis.

O22 have now made two attempts to justify the Ohneiser et al. (2021) hypothesis of tropospheric smoke self-lofting without success, in my assessment. Unless the scope of this paper is refined by eliminating the section on tropospheric self-lofting, it does not merit publication.

Since little was changed in sections other than the replaced one, I will limit my review to the major change they made.

**Major Concerns**

First, it must be stated that a core tenet of O22's thesis is that Raikoke AOT cannot explain any more than about 10% of the MOSAiC stratospheric aerosol. They established that argument in the published O21, arguing that Raikoke sulfate AOT did not exceed 0.025, citing Kloss et al. (2021). Therein O21 acknowledged that the 0.025 value is a "mean value for the latitudinal belt from 40–55N in August 2019" as compared with specific MOSAIC lidar measurements. This is an "apples-to-oranges" comparison. Even though this is now established in peer-reviewed form, it must be invoked anew and questioned because O22 maintain this assertion as fundamental to their argument that smoke dominates the MOSAIC stratosphere while inadvertently presenting and interpreting contradictory lidar data (details given below). In short, they show an Arctic CALIOP layer, ascribed to Raikoke sulfates, that has an AOT far exceeding 0.025, even 0.1. Thus, it is abundantly clear that an apples-to-apples comparison of native lidar data diminishes the published and maintained assertion that Raikoke cannot explain the MOSAIC AOT.

**15 July 2019 CALIOP Analysis (Fig. 14a)**

In O21 and again in O22, the Siberian tropospheric smoke buildup deemed to be the source of the self-lofting began about 20 July 2019 and reached a peak around 26 July. O22 now present a CALIOP tropopause-level aerosol observation on 15 July 2019 as their centerpiece "footprint" of the self-lofting pathway, "(downwind) of the main fire areas" (See Line 530-531). They emphasize that they have examined every relevant CALIOP high-latitude curtain between the Raikoke eruption date and early October. The 15 July aerosol layer is presented as the first signal of UTLS smoke that was lofted diabatically (and support that statement by declaring that it is "expected and predicted by the simulations (Fig. 7) as a consequence of the ascent rate profile with the minimum at the tropopause."). They do not identify any new "fire areas" that are upstream of this 15 July CALIOP observation. It is physically impossible to connect a 15 July aerosol observation to a smoke buildup that begins later. So O22 either failed to introduce a new fire area and smoke buildup prior to 15 July or they have made an illogical source-receptor connection. If there were to be a new fire area and smoke source, it would have to have been in place around the start of July in some unspecified burning area. This new source would make that of O21 (Siberia,

late July into August) irrelevant or at most a secondary contributor.

There is strong, independent evidence that the diffuse 15 July CALIOP layer O22 interpret as smoke (Figure 14a) is Raikoke sulfate. There are two coincidences with ACE-FTS and Imager occultations straddling the CALIPSO orbit. As shown below, the tropopause-level diffuse aerosol layer O22 attribute to non-pyroCb smoke is accompanied by $SO_2$ enhancement yet no CO enhancement.  This is of course more supportive of the layer being a Raikoke sulfate layer than smoke. Hence this 15 July CALIOP curtain, showing a widespread high-latitude tropopause-level sulfate plume actually offers a rebuttal to the O22 argument that this, and other similar looking later layers, were smoke. It will be shown later that CALIOP/ACE coincidences between 20-26 July all show support for volcanic material over biomass burning aerosol.

**ACE figure caption:** Two panels. Each shows Imager 1 μm total extinction (green) and temperature (blue). Background extinction is also plotted (green dashed line), calculated as the average of May 2019 data north of 40°N. Left panel shows FTS SO2, right panel shows FTS CO. Each is plotted in red. Background average and avg. + 3-sigma are gray solid and dashed lines, respectively. Note: Extinction abscissa is not shown, to minimize clutter. Extinction is plotted on a log scale

between 5e-5 and 5e-2/km. Note: CO and SO2 are plotted on a linear scale. SO2 background average and 3-sigma hover close to the x-scale origin but both are visible.  Annotation gives occultation ID, date, time, latitude, longitude.

**CALIOP/ACE figure caption:** O22 Fig. 14a extended to cover Siberia, with ACE SO2 profile overlain. Vertical red arrow shows ACE latitude.

[Figure]

**ACE occultation east of the CALIOP orbit.**

[Figure]

**ACE occultation west of the CALIOP orbit.**

[Figure]

[Figure]

**O22, Figure 14a CALIOP granule extended over Siberia**

As mentioned above, the 15 July onset of the tropopause-level smoke "footprint" is the centerpiece of their revised line of argumentation in support of the tropospheric self-lofting scenario. By itself it renders this section as thoroughly unconvincing. But O22 introduce two other new lines of argument that are equally weak. These will be covered in more detail below. Given that this is their second attempt to tease out tropospheric self-lofting observations, the essential importance of it to O22's overarching claim, and that Raikoke sulfates provide an alternative to smoke even for these stratospheric entry-level aerosols, this should be viewed

as a closed case, in my assessment. The authors are encouraged to refute this conclusion or defend the new material in Section 5.

**Figure 7b and discussion thereof:**
Meteorologically I do not understand how the vertical gradient of potential temperature leads to a local minimum of lifting rate at the tropopause. Potential temperature increases monotonically throughout the tropopause and lower stratosphere. The naturally positive gradient is weak in the well mixed troposphere and larger in the stable stratosphere. The tropopause manifests as the transition from small to large positive gradient.  What is it about the potential temperature gradient change that leads to the local minimum in lifting rate? Some more explanation would be beneficial.

**Abstract, Lines 4-6, "The main goal of the study is to demonstrate that radiative heating of intense smoke plumes is capable of lofting them from the lower and middle free troposphere (injection heights) up to the tropopause without the need of pyrocumulonimbus (pyroCb) convection.:** This has already been accomplished by Boers et al. (2010), who prescribed

similar conditions involving super strong smoke AOT and little or no diffusion over several days. For this work to represent new information it would have to show observations in support of simulations like Boers et al. or this one. O22 state in the body of this work that this is essentially "impossible." Hence, a demonstration (beyond modeling) has not been shown. How does this affect O22's main goal?

**Line 209-210, "…can complete the aging process and as a result get compact and spherical in shape.** This manifestation of smoke aging was hypothesized by O21. Here it is taken as a given. In my first review I pointed out that several pubs showed aged tropospheric smoke retaining depolarization ratios outside the realm of pure spheres. O22 did not dispute the papers I cited. However, they did acknowledge that small departures from a perfect sphere will introduce a "significant jump" in depolarization. Consequently, the previously published reports of aspherical aged tropospheric smoke must either be disputed or else the O21 aged, pure spherical smoke hypothesis remains in dispute.

**Line 220-222, "All observed pyroCb-related stratospheric smoke plumes, without any exception, show a high particle linear depolarization up to 0.2 at 532 nm…":** This is incorrect. Siebert et al. (2000) and Fromm et al. (2008) show, for two major separate pyroCb events, smoke depolarization ratio that is in the aspherical regime but much less than the "large" values in the cited papers. Hence, even for undisputed pyroCb plumes, the depolarization ratio spans values from small to large. These publications should be mentioned along with the others and the implications discussed.

**Line 253-254, "Only spherical particles are able to produce these rather low particle depolarization ratios of 0.02-0.03 as measured in the stratosphere in the summer 2019.":** As acknowledged herein, O22 point out Raikoke sulfate observations using CALIOP and associate them with these near-zero depolarization ratios. From Line 521-523, "From end of June to mid July the number of spot-like layers with strong backscattering increased. Besides smoke layers, more and more volcanic sulfate plumes (indicated by a low depolarization ratio) appeared…" So in this regard, the authors have affirmed that sulfate typing can be inferred from depolarization ratio in isolation. Doesn't this complicate the CALIOP analysis performed herein?

**Line 257-259, "A compact overview of the microphysical, chemical, optical and cloud-relevant properties of tropospheric and stratospheric smoke and changes of these properties during the aging process can be found in Ansmann et al. (2021b, 2022).":** Fiebig et al. (2002; https://doi.org/10.1029/2000JD000192) conclude that the 9-day old free tropospheric smoke over Lindenberg in August 1998 was nonspherical based on lidar depolarization ratios between 6-11%. This is yet another published example of aged non-pyroCb smoke that has larger than spherical depolarization ratios. The authors are encouraged to include this reference and discuss the wider implications on their conclusions.

**Line 527-529, "Very low wind speeds and weak horizontal air mass transport (stagnant conditions) favored the accumulation of smoke, the evolution of high AOTs on a regional scale, and thus self-lofting effects.":** Indeed O21 showed AOT ramping up after 20 July. The AOT peak occurred on or about 26 July. Any "self-lofting effects" like accumulation of tropopause-level smoke would not begin until at least a few days after this AOT ramp-up, according to arguments made in this paper. Here O22 clearly stake their following

arguments on the Siberia smoke build-up established by O21. This is problematic when considering the analysis that follows this statement. The authors should address this apparent problem.

**Line 530, "On 15 July (Fig. 14a),…":** This is 5-10 days before the Siberia smoke started increasing. O22 say this smoke is downwind of the main fire areas, but there are no main fire areas until later in July. Moreover, the 9-10 km layer stretches from east longitudes to west longitudes, as far from Siberia as Hudson Bay. To state that this diffuse layer is downstream of Siberia is a stretch is seemingly in defiance of logic. Please explain.

**Line 531-532, "This layer in the 8-10 km height range (not visible in the CALIOP data before 15 July) was…":** Here O22 clearly establish 15 July as the onset of their hypothesized, post-tropospheric-lofting smoke condition. It implies that something important started some days before. If so, there was no evidence presented showing that the Siberia fires were in action before 15 July. Neither did they introduce an earlier tropospheric smoke build-up anywhere in the northern latitudes. Please explain what precursor conditions existed, if any.

**Line 532-533, "…well distinguishable from the plume-like pyroCb-related smoke layers and volcanic sulfate plumes at 13-15 km height. ":** Which ones are smoke? Sulfate? All of them have nil depolarization ratio.
[https://www-calipso.larc.nasa.gov/products/lidar/browse_images/show_detail.php?s=expedited&v=V3-30&browse_date=2019-07-15&orbit_time=18-00-00&page=3&granule_name=CAL_LID_L1_Exp-Prov-V3-40.2019-07-15T18-00-00Z.hdf](https://www-calipso.larc.nasa.gov/products/lidar/browse_images/show_detail.php?s=expedited&v=V3-30&browse_date=2019-07-15&orbit_time=18-00-00&page=3&granule_name=CAL_LID_L1_Exp-Prov-V3-40.2019-07-15T18-00-00Z.hdf)
Note that AOT in the strongest plugs (gray backscatter) exceeds 0.3. Depolarization ratio there is nil. So, if these are Raikoke sulfates, O22 have shown that the volcanic sulfates have native AOT far exceeding 0.025.

**Line 536-537, "The occurrence of such a diffuse layer around the tropopause was expected and predicted by the simulations (Fig. 7) as a consequence of the ascent rate profile with the minimum at the tropopause.":** If this is expected and predicted by the model, then why did this layer just show up on 15 July at the tropopause? If it was the result of slow lofting in stagnant conditions, one would find this layer on earlier days at lower altitudes. Moreover, one would be able to trace it

downward to very intense smoke layers. And that would have to have been earlier in July, when there was no reported "main fires" or smoke buildup. Please explain.

**Line 548, "Shortly after the 26 July, Xian et al. (2022) report a strong increase of aerosol pollution over the High Arctic.":** Xian et al. do not present any such data in 2019. Their case study is for August 2021. Please explain or remove this statement.

**Line 548-550, "The area mean 550 nm AOT for the Arctic region from 70°-90°N increased from long-term mean values of 0.14 before 28-29 July 2019 to the record-breaking value of 0.4 on 10 August 2019. Never before such a High Arctic mean AOT was observed the authors** [Xian et al.] **stated.":** Xian et al. do not present any data for 10 August for any year. I could find no place in that paper where they made a claim about any 10 August AOT being the largest ever recorded. How can the source for record breaking AOT in 2019 come from fires in 2021? This appears to be a misattribution of Xian et al. Please explain.

**Line 551, "The source for the record-breaking Arctic aerosol can only be the Siberian fires in July and August**

**2019.":** This is clearly at odds with their analysis of and importance ascribed to the CALIOP diffuse layer on 15 July. This line of reasoning is therefore problematic and needs to be revised.

**Line 555-557, "Under these conditions with very large AOT values over extended Siberian and Arctic terrain one can assume that there were several subregions with AOTs>1.5 over days if not for more than a week so that the probability for self-lofting events was high in July and August 2019.":** Here O22 unambiguously describe the conditions that are favorable for eventual lofting of smoke to the UT. It involves days of super large AOT in the lower troposphere. O21 claimed that these conditions ensued after ~21 July. So how can any of the CALIOP layers they discuss on 15, 25, and 26 July be the result of this mechanism? At the very least O22 are encouraged to abandon the O21 source term and find another high AOT event before 15 July (their first day of tropopause-level diffuse smoke attributed to this pathway).

**Line 563-564, "For comparison, a Raikoke-related AOT of 0.025 was expected at 532 nm at high northern latitudes considering the emitted SO2 mass of around 1.5 Tg (Ohneiser et al., 2021).":** AOT observations of Raikoke sulfates reported in this paper far exceed 0.025. This is at odds with O21. O21 did not argue the 0.025 limit based on SO2-sulfate conversion calculations, but rather other observations such as Kloss et al. As previously discussed in prior reviews, Kloss et al.'s AOT values were biased low with respect to point measurements such as those from lidar.

**Line 576-577, "Such a high lidar ratio has never been observed for volcanic sulfate aerosol.":** Perhaps until now. Given that O22 have perhaps inadvertently demonstrated the omni-presence of Raikoke sulfates, with small and large AOT, from the tropopause to lower stratosphere, days to weeks before smoke could have entered the UTLS in abundance, it may be reasonable to conclude that indeed some sulfates may have the optical properties that O22 relegate to smoke presence. Please comment on this and/or make suitable revisions.

**Line 577-578, "The particle depolarization ratios were <0.03 at both wavelengths (as given in Fig. 15), a clear signature of perfect spherical particles, ...":** Agreed. And

perfect spherical particles are not the norm for tropospheric smoke of this age. See my prior review and comments above. Please explain how the previously published reports of aged tropospheric and pyroCb smoke with depolarization ratios ~0.03-0.11 fit into O22's interpretation.

**Line 579-580, "In cases of pyroCb-aided lofting, the depolarization ratios were always observed to be >0.1 during the first month after the pyroCb events.":** This was not the case for Norman Wells (Siebert et al., 2000) or Chisholm (Fromm et al. 2008). This disparity must be recognized, acknowledged, and explained.

**Line 585-586, "We therefore have our doubts that one can obtain a clear picture of the aerosol composition from infrared absorption spectra alone.":** These doubts are well founded. It is prudent to doubt the full veracity of any composition determination based on any single remote sensing data item. That doubt applies equally to lidar backscatter, in this case the over reliance on spectral dependence of lidar ratio. Hence, doubts should be spread equally and the authors are advised to consider that.

Perhaps more importantly, Boone et al. (2022) did not rely solely on the IR spectra. A full interpretation of that paper must take into account the associated ACE Imager aerosol extinction and ACE-FTS SO2 and HCN measurements. These orthogonal indicators were all presented together by Boone et al., leading to their robust conclusion of sulfate dominance over smoke.

**Line 587-593, discussion of AIRS CO:** It is essential for O22 to show these results such that they can be evaluated and reproduced. Their claim here is brand new and of fundamental substance. The AIRS averaging kernels are such that the CO signal peaks between 300-600 hPa, so a strong signal at 100 hPa cannot be divorced from the total column amount. Moreover, a check that I performed on Siberia/Arctic AIRS August monthly CO showed no obvious enhancement at 100 hPa. More reason for O22 to fully lay out this analysis.

For additional benefit to the authors I append below an analysis similar to that shown above, combining CALIOP 532 nm attenuated backscatter coefficient and ACE data for daily coincidences over Siberia between 20-26 July 2019. Interpretations are given with each set of ACE plots and a wrap-up discussion follows all. In a nutshell, the

ACE data show SO2 enhancements with each coincidence and an absence of CO enhancement within any aerosol layer. Hence, over Siberia at the critical time of hypothesized self-lofting, all UTLS aerosols are combined with sulfur enhancement. One can compare the 25 and 26 July examples with O22 Fig. 14.

[Figure]

Aerosol and SO2 together. No CO enhancement in layer.

[Figure]

[Figure]

**25 July**

**Aerosol and SO2 together. No CO enhancement in layer.**

[Figure]

[Figure]

**24 July**

Aerosol and SO2 together. CO enhancement at bottom of layer.

[Figure]

23 July

[Figure]

Aerosol and SO2 together. CO enhancement below main SO2 bump .

[Figure]

**22 July**

[Figure]

Aerosol and SO2 together. No CO enhancement in layer.

[Figure]

**21 July**

Aerosol and SO2 together. CO enhancement near bottom of and below layer.

[Figure]

**20 July**

[Figure]

Aerosol and SO2 together. 3-sigma CO enhancement near bottom of and below layer.

[Figure]

**Synthesis:**

UTLS diffuse backscatter enhancements each and every day between 20-26 July over Siberia.

In every example, ACE SO2 was enhanced in the diffuse backscatter layer.

Excellent correspondence between ACE Imager aerosol enhancement and CALIOP backscatter.

On a couple occasions there was a CO enhancement in the lowermost portion of the most prominent ACE
  layer. These UT enhancements were at some points associated with no discernable Imager extinction enhancement.

There were no CO enhancements in the prime diffuse backscatter layers.

In the AOD onset period (20-22 July) there was no justification for locally lofted smoke to the UT. Hence,
 the UT CO enhancements were likely aged air from another source.

**Conclusion**: Raikoke sulfates clearly overwhelm any other explanation for UTLS aerosols over Siberia when the diabatic lofting
 of smoke was hypothesized to have started. The relatively strong but diffuse CALIOP backscatter is demonstrably sulfate
in all the examples presented.

---

## Author Response (AR2)

**Dear Editor,**

In the following, please find our responses to the comments of Mike Fromm.

The reply letter is written by Albert Ansmann.

The statements of Mike Fromm is in BLACK, my response is in BLUE.

After reading the review of Mike Fromm we realized that the scientific discussion concerning the major cause of the aerosol in the ULTS height range over Siberia and the adjacent Arctic Ocean in the second half of 2019 shows no signs of dwindling. Opposing viewpoints have been expressed in recently published papers. One side argues that the major cause for the lower stratospheric aerosol is the Raikoke volcanic eruption (see for example Boone et al., JGR, 2022, doi: 10.1029/2022JD036600), while the other side counters with the presence of an aerosol layer in the lower stratosphere that consisted of a mixture of smoke (80-90% fraction) and sulfate aerosol (10-20%) and was observable in the UTLS height range over the High Arctic until May 2020 (Ohneiser et al, ACP, 2021). In the revised version of the manuscript, we now present a complete order of events regarding the beginning of the long-lasting Siberian fire season (in June 2019), about the transport of smoke towards and across the Arctic Ocean (continuously from June to August 2019), and regarding the potential development of smoke layers near the tropopause (since mid July 2019) and within the entire UTLS height range later on.

**Next, the quality of the review:**

We (all co-authors) were expecting good, constructive, and critical comments with the goal to improve the paper, rather than hard words and destructive comments with the clear goal to prevent the paper by systematically destroying all our arguments. The suggestion to reject our manuscript just because of having opposing viewpoints supported by measurements, on the major cause of smoke in the polar region, is also surprising. Regarding scientific significance, scientific quality, and presentation quality, Mike Fromm selected the lowest possible quality level: LOW. The scientific discussion should be, first of all constructive but not personal. We tried to ignore all unconstructive/personal statements and stick to our observations, conduct detail and comprehensive data analyses as well as referring to already published studies.

Section 5 in the revised version, that Mike Fromm found rather flawed, was written by myself (AA). I did the CALIOP data analysis. I designed a careful, well-balanced and to my opinion trustworthy, accurate, and informative discussion. I was excited to find the nice consistency between the CALIOP observations and the prediction of diffuse aerosol layers close to the tropopause by the simulations analyses.

These 'LOWs' regarding scientific significance, scientific quality, and presentation quality are to my opinion not justified and thus wrong assessments. They are totally in contradiction with three objective facts that should count in judging of our manuscript: (a) the manuscript does not contain any methodological error, (b) the topic is clearly scientifically sound and of very high relevance, and (c) our argumentations in the result sections 4 and 5 are simply correct and carefully done.

In our revised (and previously, heavily criticized Section 5), we now present a complete order of events regarding the Siberian fires and the obvious development of diffuse smoke layers near the tropopause. As mentioned, we provide detailed information about the beginning of the fire season (in June 2019), about the transport towards the Arctic (in July and August 2019), the occurrence of first diffuse layers (mid of July 2019), preferably above the Arctic Ocean, and finally about the lifetime of the entire UTLS aerosol layer over the High Arctic (until the beginning of May 2020).

**Please find below our point by point responses**

Mike Fromm comments are in black

Note: "O22" is shorthand for the author group of this manuscript. "O21" is used to refer to Ohneiser et al. (ACP, 2021).

O22 have responded to both reviewers' comments and substantially modified the manuscript. My assessment is that their responses to my deepest concerns were inadequate and the revised manuscript is as flawed as the original. Their crucial new section, replacing the flawed original section, abounds with demonstrable inaccuracies, misinterpretations of satellite data, conflicting messaging, and unmet expectations from their new, season-long CALIOP analysis.

We clearly disagree with this devastating verdict. And as we will show in this letter, most points of Mike Fromm are wrong. Very disappointing that we get no clarification on what the reviewer means by stating on "misinterpretation of satellite data, demonstrable inaccuracies and conflicting messaging"

O22 have now made two attempts to justify the Ohneiser et al. (2021) hypothesis of tropospheric smoke self-lofting without success, in my assessment. Unless the scope of this paper is refined by eliminating the section on tropospheric self-lofting, it does not merit publication.

The Section 5 on tropospheric self-lofting is one of the highlights of our study and cannot be removed from our manuscript. In this section, exciting CALIOP observations (diffuse smoke layer below the tropopause) in full consistency with the prediction by the simulation model are presented . So again, Section 5 cannot be removed.

Since little was changed in sections other than the replaced one, I will limit my review to the major change they made.

**Major Concerns:**

First, it must be stated that a core tenet of O22's thesis is that Raikoke AOT cannot explain any more than about 10% of the MOSAiC stratospheric aerosol. They established that argument in the published O21, arguing that Raikoke sulfate AOT did not exceed 0.025, citing Kloss et al. (2021). Therein O21 acknowledged that the 0.025 value is a "mean value for the latitudinal belt from 40–55N in August 2019" as compared with specific MOSAIC lidar measurements. This is an "apples- to-oranges" comparison.

The Raikoke volcano emitted around 1.5 Tg SO2. This amount of SO2 was then converted to sulfate aerosol. According to many publications regarding the context of emitted SO2 -> converted sulfate aerosol -> resulting hemispheric mean stratospheric 500nm AOT, there is a clear link between the emitted SO2 amount and the resulting hemispheric mean AOT (or maximum AOT in case of a uniform sulfate distribution six weeks after the eruption) (Haywood et al., 2010, Begue et al., 2017, Ansmann ezt al., 1997). And for the emitted Raikoke SO2 amount, the maximum Raikoke Northern Hemispheric mean 500 nm AOT is 0.025. This maximum AOT occurred in mid-August 2019, about six weeks after the eruption (Kloss et al., 2021). This 'universal law' (SO2-> sulfate conversion -> sulfate AOT) is well described in the literature for, e.g., in the case of the Sarychev eruption (Haywood et al., JGR, 2010), Calbuco eruption (Begue et al., ACP, 2017) and Pinatubo eruption (in the case of mid latitude volcanoes)

or even until the next winter half year (in the case of tropical volcanoes as Pinatubo) before an almost homogeneous or uniform distribution of sulfate aerosol can be expected. Then, the hemispheric mean AOTs can be measured over large parts of an hemisphere (i.e., when the sulfate is uniformly distributed). By assuming typical e-folding decay times of 3-5 months, the 500 nm Raikoke sulfate AOTs were then about 0.01-0.015 in the stratosphere in September and October 2019, but AOTs around 0.1-0.15 were observed in the UTLS height range over the Arctic in October 2019, during the MOSAiC campaign (Ohneiser et al., ACP, 2021). So, the Raikoke sulfate fraction was of the order of 10%.

Even though this is now established in peer-reviewed form, it must be invoked anew and questioned because O22 maintain this assertion as fundamental to their argument that smoke dominates the MOSAIC stratosphere while inadvertently presenting and interpreting contradictory lidar data (details given below). In short, they show an Arctic CALIOP layer, ascribed to Raikoke sulfates, that has an AOT far exceeding 0.025, even 0.1. Thus, it is abundantly clear that an apples-to-apples comparison of native lidar data diminishes the published and maintained assertion that Raikoke cannot explain the MOSAIC AOT.

So, the facts were already described above. The Raikoke layers that produced AOTs>0.1 according to the CALIOP backscatter coefficients in pronounced plumes occurred in July 2019. In this month, just 10-40 days after the Raikoke eruption and almost 2-6 weeks after the start of the Siberian fire season, the aerosol situation was rather complex, and the SO2 gas and the sulfate aerosol was highly non-uniformly distributed . Many aerosol plumes and small layers occurred in the stratosphere (appearing as patches features in the CALIOP data) as a result of inhomogeneously distributed SO2 plumes. Even several pyroCb-related smoke plumes originating from fires in Alaska and other regions in North America showed up in the CALIOP observations. You cannot expect in these early times after the eruption (in July 2019) to observe homogeneous sulfate structures and layering features. Sure, at these spatially inhomogeneous conditions some of the patchy Raikoke sulfate layers showed AOTs of 0.2-0.3, if not more. To repeat, the only constraint is that the Northern Hemispheric mean 500 nm AOT was certainly still below 0.02 in July 2019. The maximum Northern Hemispheric mean AOT was expected around 10 August 2019 with values of 0.025.

To find an almost homogeneous sulfate distributions (north of e.g. 50°N), one had to wait until the end of August 2019, or even until mid of September and October 2019, and then Raikoke AOTs were certainly around 0.015 and less when considering e-folding decay times of 3-5 months, disregarding where your observational site was located (at low or high mid latitudes or Arctic latitudes).

All this, what I explained here about stratospheric volcanic aerosols, are simple and not new facts and aspects! Everyone who is dealing with stratospheric aerosol should know that!

In O21 and again in O22, the Siberian tropospheric smoke buildup deemed to be the source of the selflofting began about 20 July 2019 and reached a peak around 26 July.

We agree, there was not enough or partly confusing information about the start of the 2019 Siberian fire season. This section in our revised manuscript is significantly improved. Meanwhile, we are able to present a complete order of Siberian smoke-related events. A new paper came out in December 2022 (Sorenson et al., ACPD, 2022). In this paper, the authors clearly show (in their Figure 9) that the smoke in the Arctic in August 2019 originated from the strong fires in Siberia. The Aerosol Index (AI) time series shown by Sorenson et al. (2022) and also shown in our reply letter (see figure below) suggest, that the Siberian smoke season started around 10-15 June 2019. This figure in this reply letter was exclusively prepared by Blake Sorenson. Xian et al 2022 (ACP, part 2, published in August 2022, Blake Sorenson is coauthor on that paper) then reported a steady and almost monotonic increase of the AOT over the Arctic region (daily means, 70-90°N mean) from 22 June 2019 to 11 August 2019 (see Figure 9 in Xian et al., 2022) from background values close to 0.05 on 20-22 June to >0.4 on 11 August and then back to <0.075 in September 2019 with a contribution of about 0.01 by Raikoke sulfate (to our opinion). Such a coherent and rather strong increase of the AOT over six weeks and to reach a record-breaking AOT value of >0.4 can only be caused by a huge and long lasting fire event such as the Siberian fires. We can conclude from these findings of Xian et al. (2022) that the Siberian fire season started around 15 June 2019, one week before the steady increase of the AOT over the Arctic began.

Note, at the time of the submission of the original version of the self-lofting manuscript (in May 2022), we had only the article of Johnson et al. (2021, Atmos. Env.) that focused on Siberian fires (besides Ohneiser et al., 2021). Johnson et al. (2021) emphasized that the most intense fire period over central-eastern Siberia was from 19 July 2019 to 14 August 2019. So, 19 July was not the beginning of the Siberian fire season. Johnson et al. (2021) concentrated on July and August 2019 only, and according to their study, the first intense fires were on 4-5 July 2019 in central-eastern Siberia (north of Lake Baikal). All this is now given in our new (revised) version of the manuscript. In the beginning of Section 5, we introduced several new paragraphs with all the information from the Johnson et al. (2021), Sorenson et al. (2022) and Xian et al. (2022) papers.

Figure 1: Aerosol Index (AI, daily mean, 70-80°N mean), from Figure 9a in Sorenson et al (ACPD 2022, https://doi.org/10.5194/acp-2022-743), here only for 2019. The extreme AI peak was measured on 11 August 2019. This was the highest peak they ever observed.

Figure 14d in Section 5 in our manuscript shows a CALIOP observation over the Arctic from 10 August 2019 (the CALIOP data base has no data for 11 August). Johnson et al (2021) already showed that Siberian smoke plumes went to the Arctic. Xian et al. (2022, part 2) and Sorenson et al. (2022) definitely show that huge amounts of Siberian smoke producing record-breaking daily mean AOTs of > 0.4 (for the polar region at latitudes >70°N in August 2019) traveled to the Arctic in the summer of 2019.

Now, the following fundamental questions arise: What happens with all the sun-light-absorbing smoke plumes (during almost unlimited sunshine periods over high northern latitudes)? What will be the consequence of heating of all the aerosol layers in the troposphere over days to weeks? What process is able to prevent them from (well-organized large scale) self-lofting? And finally, what features produced by the expected regional-scale lofting events may be detectable in space borne lidar data sets?

The first sign for an organized diffuse layer close to the tropopause as a consequence of self-lofting was observed in the CALIOP data base on 10 July 2019. And the first nice coherent structure (over more than 3000 km along a polar CALIOP track) was then visible in the CALIOP data over the Arctic on 15 July 2019, and this case is selected to be shown in Fig. 14a. That was the simple motivation for showing the 15 July CALIOP case. It was the first time that we detected such coherent structures (over 3000 km!).

O22 now present a CALIOP tropopause- level aerosol observation on 15 July 2019 as their centerpiece "footprint" of the self-lofting pathway, "(downwind) of the main fire areas" (See Line 530-531). They emphasize that they have examined every relevant CALIOP high-latitude curtain between the Raikoke eruption date and early October.

Yes, I like to emphasis that here again. It was my personal pleasure to do all this, digging in the CALIOP color plots, back and forth. For more than two weeks I was busy with this and looked at all these lidar data with my (almost) 40-year experienced aerosol lidar eyes. At the end, I was very pleased about the results of my study! In the beginning, I was surprised that one could see these diffuse smoke layers in the CALIOP data, but then, when these layers showed up again and again, mostly at 70-80°N, over several months, I was convinced that this must have to do with the self-lofting of Siberian smoke.

We have never observed such persistent layers below and around the troposphere with lidar at Leipzig after volcanic eruptions, and we do continuous stratospheric observations since the Pinatubo eruption in 1991. These near-tropopause aerosol features cannot be explained by Raikoke aerosol.

And what happens with the smoke that reached the tropopause and started to penetrate into the stratosphere by further self-lofting? It is very likely that both, the sulfate aerosol (that formed in the stratosphere) as a well as the smoke aerosol (that was lofted from the mid troposphere), contributed to enhance aerosol extinction coefficients above the tropopause up to 15 km height over Siberia and the downwind regime over the Arctic Ocean in July 2019. All this is discussed in Section 5. And our hypothesis is in excellent agreement with lidar observations at Leipzig (14 August 2019, Fig. 15 in Section 5, already discussed in Ansmann et al., Frontiers, 2021)) and over the High Arctic during the MOSAiC expedition (end of September 2019 to the beginning of May 2020) as reported by Ohneiser et al. (2021).

The 15 July aerosol layer is presented as the first signal of UTLS smoke that was lofted diabatically (and support that statement by declaring that it is "expected and predicted by the simulations (Fig. 7) as a consequence of the ascent rate profile with the minimum at the tropopause."). They do not identify any new "fire areas" that are upstream of this 15 July CALIOP observation. It is physically impossible to connect a 15 July aerosol observation to a smoke buildup that begins later. So O22 either failed to introduce a new fire area and smoke buildup prior to 15 July or they have made an illogical source-receptor connection. If there were to be a new fire area and smoke source, it would have to have been in place around the start of July in some unspecified burning area. This new source would make that of O21 (Siberia, late July into August) irrelevant or at most a secondary contributor.

Because of this confusing situation that relates to the beginning of the Siberian fire season, we introduced several new paragraphs. The season obviously started in the beginning of June (according to the papers of Sorenson et al., 2022, and Xian et al., 2022), first intensive fires occurred on 4-5 July 2019, and the most intensive period was then observed from 19 July to 14 August 2019.

And to repeat again: The 15 July 2019 case in Fig. 14a is just the first day where we found a nice coherent aerosol structure close to the tropopause in the CALIOP data base. There is no other motivation behind.

There is strong, independent evidence that the diffuse 15 July CALIOP layer O22 interpret as smoke (Figure 14a) is Raikoke sulfate. There are two coincidences with ACE-FTS and Imager occultations straddling the CALIPSO orbit. As shown below, the tropopause-level diffuse aerosol layer O22 attribute to non-pyroCb smoke is accompanied by SO2 enhancement yet no CO enhancement. This is of course more supportive of the layer being a Raikoke sulfate layer than smoke. Hence this 15 July CALIOP curtain, showing a widespread high-latitude tropopause- level sulfate plume actually offers a rebuttal to the O22 argument that this, and other similar looking later layers, were smoke. It will be shown later that CALIOP/ACE coincidences between 20-26 July all show support for volcanic material over biomass burning aerosol.

We disagree. The layer below the tropopause cannot be explained by volcanic sulfate aerosol. That would be the first time in my life as lidar scientist that a sulfate layer originating from moderate volcanic eruptions, could establish a persistent layer below the tropopause. It is not possible.

Now to the ACE-FTS observations introduced by Mike Fromm in his review. The satellite observations of ACE-FTS (plus IR extinction channel) are trustworthy only for heights above the tropopause, or better from 1 km or 2 km above the tropopause upward. One has to be very careful in the interpretation of ACE-FTS products when assigned to height levels in the upper troposphere. For example, the unknown amount of sub visible cirrus does not allow a proper aerosol extinction retrieval around and below the tropopause.

Nevertheless, ACE-FTS shows high CO values up to the tropopause on all days presented by Mike Fromm (15 July 2019 and from 20-26 July 2019). Especially on 23 July 2019, the CO concentration was enhanced up to heights of 14 km, several kilometers above the tropopause. This is a clear indication that smoke-containing air was present in the lower stratosphere, and not only up the tropopause. The found stratospheric SO2 levels are a hint that Raikoke air was present as well. But this knowledge does not automatically indicate that the measured extinction coefficients were exclusively caused by sulfate aerosol. *It is more likely that there was a mixture of wildfire smoke (ascended from the middle troposphere upward and penetrating into the lower stratosphere) and volcanic sulfate aerosol (formed in the lower stratosphere from the Raikoke SO2).* The situation obviously changed in August, September, and October 2019. Over Leipzig (14 August 2019, Ansmann et al., Frontiers, 2021) and during MOSAiC (Ohneiser et al., ACP, 2021), we clearly found that smoke dominated in the UTLS height range up to 14 km and up to 12-13 km (over the High Arctic), and not sulfate aerosol.

All the other examples (20-26 July) Mike Fromm presents and discusses at the end of his review, can be interpreted in the same way as the one for 15 July 2019. We agree, we should better emphasize the complex aerosol situation in July 2019. This is now done in Section 5 of the revised version of the manuscript.

As a new point, we checked many Arctic radiosonde temperature profiles (Wyoming radiosonde data base) regarding the tropopause heights on 15 and 25 July and 10 August (these smoke cases are shown in Figure 14a, b, and d) to be sure that the diffuse smoke layers were up to the tropopause only, and not above the tropopause, as we partly stated in the last version of the manuscript. The radiosonde profiles indicate that all the diffuse layers were within the uppermost troposphere. This information is added in Section 5 as well. And the text in Section 5 is now adjusted to these findings.

As mentioned above, the 15 July onset of the tropopause-level smoke "footprint" is the centerpiece of their revised line of argumentation in support of the tropospheric self-lofting scenario. By itself it renders

this section as thoroughly unconvincing. But O22 introduce two other new lines of argument that are equally weak. These will be covered in more detail below. Given that this is their second attempt to tease out tropospheric self-lofting observations, the essential importance of it to O22's overarching claim, and that Raikoke sulfates provide an alternative to smoke even for these stratospheric entry-level aerosols, this should be viewed as a closed case, in my assessment. The authors are encouraged to refute this conclusion or defend the new material in Section 5.

I think we explained everything exhaustingly enough before, and have not to answer these questions again and again. The Raikoke eruption (and corresponding sulfate aerosol amount) was definitely too weak and the Siberian fires (and smoke load) too strong, so that the Raikoke eruption cannot be used to explain the strong stratospheric perturbation in the second half of 2019.

Maybe the following aspect helps to assess the relevance of Raikoke aerosol: Xian et al. (2022) found this daily mean AOTs of >0.4 at 550 nm on 11 August 2019 as an average value for the 70-90°N region. This huge value was much larger than the Pinatubo maximum AOT over the High Arctic of about 0.25-0.3 (as we conclude from our long-term lidar observations in Germany at 53°N, Ansmann et al., 1997). Pinatubo emitted 20 Tg SO2. Raikoke emitted more than an order of magnitude less SO2 than Pinatubo (1.5 Tg), consequently the maximum AOT was of the order of 0.025 (in case of uniformly distributed sulfate). And this Raikoke aerosol should now be able to dominate the aerosol conditions over the Arctic in 2019 up to May 2020?

We explained the relative contribution of the Raikoke sulfate aerosol to the total stratospheric AOT in the second half of 2019 for the fourth or fifth time in several replies to Mike Fromm's reviews.

Figure 7b and discussion thereof:

Meteorologically I do not understand how the vertical gradient of potential temperature leads to a local minimum of lifting rate at the tropopause. Potential temperature increases monotonically throughout the tropopause and lower stratosphere. The naturally positive gradient is weak in the well mixed troposphere and larger in the stable stratosphere. The tropopause manifests as the transition from small to large positive gradient. What is it about the potential temperature gradient change that leads to the local minimum in lifting rate? Some more explanation would be beneficial.

**We improved this part and significantly extended the explanations (see Section 3.1, page 9).**

Abstract, Lines 4-6, "The main goal of the study is to demonstrate that radiative heating of intense smoke plumes is capable of lofting them from the lower and middle free troposphere (injection heights) up to the tropopause without the need of pyrocumulonimbus (pyroCb) convection.: This has already been accomplished by Boers et al. (2010), who prescribed similar conditions involving super strong smoke AOT and little or no diffusion over several days. For this work to represent new information it would have to show observations in support of simulations like Boers et al. or this one. O22 state in the body of this work that this is essentially "impossible." Hence, a demonstration (beyond modeling) has not been shown. How does this affect O22's main goal?

We totally disagree with most points of the reviewer. (1) First of all, the methodology (as given in our manuscript) has never been presented in that extent before, in any paper dealing with smoke self-lofting. (2) The uncertainty and impact analysis (in our manuscript) has also never been presented before. These two facts alone already justify publication.

It is noteworthy to mention here that our work was guided by Boers et al. (2010). We contact Reinout Boers, who is now retired. I personally know him well since the 1980s when he was with Harvey Melfi's famous Raman lidar team at NASA Goddard. Reinout gave us the advice to develop our own software and to present the methodology together with an extended sensitivity and uncertainty analysis because, for his opinion, that has never been done/published in the literature. All this is shown now, and it is for the first time, based on our knowledge. No doubt at all.

(3) Third point: Yes, we were believing that in the turbulent troposphere, observations of the lofting of coherent smoke layers over days is possible and presented examples in the originally submitted version from May 2022. However, we had to accept (forced by Mike Fromm's review) that this was not a good idea and not a convincing approach. That was the motivation and starting point for the new approach. In the new attempt, I personally systematically checked all CALIOP observations from June to October 2019, and in this way, I detected these diffuse layers in the uppermost troposphere in line with the predictions by the simulations. So, now we think we reached our goal. Nevertheless, it remains a hypothesis that these layers were caused by self-lofting of smoke. We emphasize that in Section 5 and in the conclusions. It is clear, we need future work in this research field direction. For example, aircraft observations of microphysical and optical properties of smoke and of the chemical composition in the upper troposphere and lower stratosphere. However, if this manuscript would be rejected, nothing would happen regarding these necessary future efforts. Proposals need motivating papers as justification for the planned goals.

Line 209-210, "...can complete the aging process and as a result get compact and spherical in shape. This manifestation of smoke aging was hypothesized by O21. Here it is taken as a given. In my first review I pointed out that several pubs showed aged tropospheric smoke retaining depolarization ratios outside the realm of pure spheres. O22 did not dispute the papers I cited. However, they did acknowledge that small departures from a perfect sphere will introduce a "significant jump" in depolarization. Consequently, the previously published reports of aspherical aged tropospheric smoke must either be disputed or else the O21 aged, pure spherical smoke hypothesis remains in dispute.

We extended the discussion in Sect. 3 (page 8). However, to our opinion, we do not need a more extended discussion on this aspect. It is quite simple: enhanced particle depolarization ratios indicate non-spherical particles, low particle depolarization ratios spherical particles. Spherical shape points to particles that could complete the aging process, and non-spherical shape to particles that could not finalize aging. There is no need to include more references here.

Line 220-222, "All observed pyroCb-related stratospheric smoke plumes, without any exception, show a high particle linear depolarization up to 0.2 at 532 nm...": This is incorrect. Siebert et al. (2000) and Fromm et al. (2008) show, for two major separate pyroCb events, smoke depolarization ratio that is in the aspherical regime but much less than the "large" values in the cited papers. Hence, even for undisputed pyroCb plumes, the depolarization ratio spans values from small to large. These publications should be mentioned along with the others and the implications discussed.

**This statement of Mike Fromm is wrong.**

I checked the two papers. In Fromm et al (2008) the lidar observations were of low quality, especially the ones of Ny Alesund. The contributing lidar groups were unable to calculate particle linear depolarization ratios from the measured volume depolarization ratios. The volume depolarization ratios are a strong function of Rayleigh depolarization. When the air is only weakly polluted (as it was the case), the volume depolarization ratio is close to zero because Rayleigh depolarization ratios are close to zero! So, only

after correcting for Rayleigh depolarization effects, the depolarization ratio provides light-depolarization information for pure particles. And only that information is useful! In the case of the Ny Alesund lidar color plots in the Fromm et al. paper the observations indicate slightly enhanced volume depolarization ratio so that I concluded, if they would have corrected that depolarization profile for the Rayleigh impact they would get particle depolarization ratios clearly >10 %. So in this paper, NO particle depolarization ratios close to zero are shown. The quality of all lidar observations in Fromm et al. (2008) was unfortunately low.

In the Siebert et al. (2000) paper they also do not show particle depolarization ratios, only the observed volume depolarization ratios of 4%. Here the lidar experts around Fricke (Uni Bonn) clearly stated that the enhanced volume depolarization ratios indicate that the backscattering particles were clearly non spherical in shape, knowing that the correction for Rayleigh backscattering would increase the depolarization ratio by a factors of 3 or even more.

It should be added here that the CALIOP classification scheme is exactly based on stratospheric particle depolarization ratios >10% to identify stratospheric smoke, assuming that only pyroCbs are able to loft smoke into the stratosphere. So obviously all stratospheric smoke layers (the CALIOP science team found and analyzed) showed enhanced depolarization ratios.

Line 253-254, "Only spherical particles are able to produce these rather low particle depolarization ratios of 0.02-0.03 as measured in the stratosphere in the summer 2019.": As acknowledged herein, O22 point out Raikoke sulfate observations using CALIOP and associate them with these near-zero depolarization ratios. From Line 521-523, "From end of June to mid July the number of spot-like layers with strong backscattering increased. Besides smoke layers, more and more volcanic sulfate plumes (indicated by a low depolarization ratio) appeared..." So in this regard, the authors have affirmed that sulfate typing can be inferred from depolarization ratio in isolation. Doesn't this complicate the CALIOP analysis performed herein?

Yes, Raikoke sulfate particles can in principle be identified by the depolarization ratio. They are liquid and thus spherical and thus they produce no depolarization of linearly polarized emitted laser photons. Correct! However, if you have a mixture of spherical smoke and spherical sulfate particles you are lost. Clear aerosol typing is no longer possible (as shown in Ansmann, et al., Frontiers, 2021).

In the period from end of June to mid of July 2019, more and more layers with sharp boundaries (the opposite to the diffuse boundaries of these diffuse smoke layers near and below the tropopause) appeared. And in these cases with pronounced plumes or layers (either produced by pyroCb lofting of smoke or conversion of SO2 into sulfate), we can identify the aerosol type. If particle depolarization ratio <5% then it is sulfate and when >10%, it is smoke.

However, the summer 2019 was quite unique regarding the aerosol pollutions and mixtures, so that a simple aerosol typing was (almost) impossible. This is exactly discussed in our paper (Ansmann et al. Frontiers, 2021, Misclassification ... by CALIOP). In this summer 2019, with the Raikoke aerosol and the self-lofted smoke aerosol, there was no way to use the particle depolarization ratio alone to perform a proper aerosol typing. Therefore, we had to use the dual-wavelength Raman lidar approach to determine the extinction-to-backscatter ratios (lidar ratios) at 355 and 532 nm. Only then an unambiguous fingerprint of smoke became visible, namely the inverse lidar-ratio spectrum (larger lidar ratios in the green, 70-100sr, than in the UV, 45-70 sr). We successfully apply this method to identify aged smoke since 1998. The method works perfectly as shown in numerous papers (e.g., Wandinger et al., JGR, 2002,

Mueller et al., JGR, 2005, Murayama et al., GRL, 2004, Haarig et al., ACP, 2018, Ohneiser et al., ACP, 2020). There is no (zero) case at all where the method failed. In contrast, we know from numerous lidar observations of volcanic aerosol layers and also many simulation papers that the 532 nm lidar ratio for typical stratospheric volcanic sulfate layers is < 50 sr. All this helps and is at the end needed to identify the dominating aerosol type. The particle depolarization ratio is sensitive to the shape of the backscattering particles, whereas the lidar ratio depends on size distribution and, more important, on the chemical composition. Smoke particles consist of OC and BC, both materials are absorbing in the visible spectrum and produce lidar ratios of 70-100 sr, sulfate particles are non-absorbing droplets consisting of sulfuric acid and water and produce lidar ratios typically clearly below 40-50 sr.

**Line 257-259, "A compact overview of the microphysical, chemical, optical and cloud-relevant properties of tropospheric and stratospheric smoke and changes of these properties during the aging process can be found in Ansmann et al. (2021b, 2022).": Fiebig et al. (2002;**

https://doi.org/10.1029/2000JD000192) conclude that the 9-day old free tropospheric smoke over Lindenberg in August 1998 was nonspherical based on lidar depolarization ratios between 6-11%. This is yet another published example of aged non-pyroCb smoke that has larger than spherical depolarization ratios. The authors are encouraged to include this reference and discuss the wider implications on their conclusions.

This is again a point we cannot implement in our revised version of the manuscript and the mentioned reference cannot be included as relevant. Even after 9 days of long-distance travel, the aging process could obviously not be entirely completed. The smoke was obviously injected into a dry air mass over Canada on 1-3 August 1998 so that the aging process was rather slow. As a consequence of incomplete aging, layers with enhanced particle depolarization ratios can occur everywhere, i.e., in both the troposphere as well as in the stratosphere.

By the way, the Canadian smoke lofted by pyroCbs on 12-13 August 2017 needed more than 3-5 months before the aging process was completed (Baars et al., 2019). The particle depolarization ratio decreased rather slowly from 20% in the beginning to below 5% after several months. Again, the rather low humidity in the stratospheric was the reason for this rather slow aging. In contrast, a lot of water vapor was injected into the stratosphere during the strong Australian fires (according to Khaykin et al., 2020), and in this case the particle depolarization ratio decreased in less than 2 months down to values indicating spherical particle forms.

Line 527-529, "Very low wind speeds and weak horizontal air mass transport (stagnant conditions) favored the accumulation of smoke, the evolution of high AOTs on a regional scale, and thus self-lofting effects.": Indeed O21 showed AOT ramping up after 20 July. The AOT peak occurred on or about 26 July. Any "self-lofting effects" like accumulation of tropopause- level smoke would not begin until at least a few days after this AOT ramp-up, according to arguments made in this paper. Here O22 clearly stake their following arguments on the Siberia smoke build-up established by O21. This is problematic when considering the analysis that follows this statement. The authors should address this apparent problem.

As mentioned, we improved this part in Section 5 to avoid confusion. To repeat: The central eastern Siberian smoke season started around 15 June 2019 (according to the paper of Sorenson et al., 2022 and Xian et al., 2022, Figure 9). Strong aerosol pollution (AOT) over the main fire area was discussed already by Ohneiser et al., (2021). The smoke was then obviously preferably transported into the Arctic (Sorenson et al., 2022, Xian et al., 2022, parts 1 and 2). The AOT over the Arctic region (shown as daily means, 70-90°N mean) almost monotonically increased from 22 June 2019 to 11 August 2019, and reached record-breaking AOT values over the Arctic of more than 0.4 (daily mean, 70-90°N area mean). There was already a period with strong fires over central eastern Siberia on 4-5 July 2019 (Johnson et al., 2021). However, the most intense fire period was from 19 July 2019 to 14 August 2019 and pushed probably self-lofting processes a lot in August. All this is now given in the revised version of the manuscript (Section 5). So there is no confusion left in the case of the discussion of the 15 July self-lofting event.

Line 530, "On 15 July (Fig. 14a),…": This is 5-10 days before the Siberia smoke started increasing. O22 say this smoke is downwind of the main fire areas, but there are no main fire areas until later in July. Moreover, the 9-10 km layer stretches from east longitudes to west longitudes, as far from Siberia as Hudson Bay. To state that this diffuse layer is downstream of Siberia is a stretch is seemingly in defiance of logic. Please explain.

As we already pointed out, because of these many statements by Mike Fromm concerning the obvious mismatch (beginning of the fire period on 20 July and detection of the first diffuse layer on 15 July) we were forced to add these new paragraphs in the beginning of Section 5. Now we provide clear information about the beginning of the Siberian fire season, the first intensive fires on4-5 July, and the most intensive fire period from 19 July to 14 August 2019, and that the smoke mainly traveled to the Arctic. Now, all the confusions are eliminated.

Line 531-532, "This layer in the 8-10 km height range (not visible in the CALIOP data before 15 July) was...": Here O22 clearly establish 15 July as the onset of their hypothesized, post-tropospheric-lofting smoke condition. It implies that something important started some days before. If so, there was no evidence presented showing that the Siberia fires were in action before 15 July. Neither did they introduce an earlier tropospheric smoke build-up anywhere in the northern latitudes. Please explain what precursor conditions existed, if any.

**See explanations above.**

Line 532-533, "...well distinguishable from the plume-like pyroCb-related smoke layers and volcanic sulfate plumes at 13-15 km height. ": Which ones are smoke? Sulfate? All of them have nil depolarization ratio. Note that AOT in the strongest plugs (gray backscatter) exceeds 0.3. Depolarization ratio there is nil. So, if these are Raikoke sulfates, O22 have shown that the volcanic sulfates have native AOT far exceeding 0.025.

All layers which are clearly visible (with clear and sharp boundaries) and are fully in the stratosphere (above the tropopause) cannot be linked to self-lofting processes because these lofting processes are too slow and cannot produce such sharp boundaries and structures. They must be caused by conversion of SO2 plumes into sulfate plumes (in situ) in the stratosphere in these cases with particle depolarization ratio close to zero in these layers with sharp edges.

Second point: As mentioned, it is not surprising to find sulfate layers with AOT exceeding 0.3 in July 2019, when so many patchy features were observed. However, it would be surprising if the Northern Hemispheric mean AOT (integral over all patchy sulfate features) would exceed 0.02 in July or 0.025 later on in mid August 2019. If the Northern Hemispheric mean 550 nm AOT would have been larger than 0.025, as Mike Fromm wants to convince us all the time, the universal law regarding the link between emitted SO2 and resulting hemispheric mean AOT would have been violated for the first time ....!

Line 536-537, "The occurrence of such a diffuse layer around the tropopause was expected and predicted by the simulations (Fig. 7) as a consequence of the ascent rate profile with the minimum at the tropopause.": If this is expected and predicted by the model, then why did this layer just show up on 15 July at the tropopause? If it was the result of slow lofting in stagnant conditions, one would find this layer on earlier days at lower altitudes. Moreover, one would be able to trace it downward to very intense smoke layers. And that would have to have been earlier in July, when there was no reported "main fires" or smoke buildup. Please explain.

In the previous review by Mike Fromm, he criticized that we compared lofted layers, observed over several days within the turbulent troposphere, and we thus concluded that such an approach is useless. So, we selected another way this time and found these diffuse layers close to the tropopause. Now we were surprised to find that the reviewer wants that we implement the 'useless' approaches? We disagree. There is no way to trace back diffuse layers, that reached the tropopause, 3, 5,10 or even 20 days after smoke injection into the 3-6 km height range.

However, we show a case from August 2021 with lofting plumes in Section 5. But here we were lucky to find a quite stable layer and that we had two CALIOP observations within 15 hours.

Line 548, "Shortly after the 26 July, Xian et al. (2022) report a strong increase of aerosol pollution over the High Arctic.": Xian et al. do not present any such data in 2019. Their case study is for August 2021. Please explain or remove this statement.

Xian et al. do present such data in 2019 .... Please have look on Figure 9a and 9b.... It was recently even approved during my personal communication with Blake Sorenson (he is co-author of the Xian paper) that the peaks in the Xian paper are from 11 August 2019. In Figure 9, one can also clearly see the steadily increasing AOT since 22 June 2019 towards 11 August 2019.

Line 548-550, "The area mean 550 nm AOT for the Arctic region from 70°-90°N increased from longterm mean values of 0.14 before 28-29 July 2019 to the record- breaking value of 0.4 on 10 August 2019. Never before such a High Arctic mean AOT was observed the authors [Xian et al.] stated.": Xian et al. do not present any data for 10 August for any year. I could find no place in that paper where they made a claim about any 10 August AOT being the largest ever recorded. How can the source for record breaking AOT in 2019 come from fires in 2021? This appears to be a misattribution of Xian et al. Please explain.

I analyzed Figure 9 with a linear (triangle ruler). So I measured the full August scale in Figure 9 and then the part up to the peak. I found out it must be 11 August 2019. And it is 11 August 2019 as Blake Sorenson confirmed by e-mail. However, by repeating my 'measure', I realized now that I made a mistake in defining the beginning of the fire season. The beginning of the fire season is definitely a few days before 22 June 2019. That is now clearly stated in Section 5.

Line 551, "The source for the record-breaking Arctic aerosol can only be the Siberian fires in July and August 2019.": This is clearly at odds with their analysis of and importance ascribed to the CALIOP diffuse layer on 15 July. This line of reasoning is therefore problematic and needs to be revised.

As we already mentioned above several times (and here again), the central eastern Siberian smoke season started around 15 June 2019 (according to the papers of Sorenson et al., 2022, and especially, of Xian et al., 2022). Strong aerosol pollution (AOT) over the main fire area was discussed already by Ohneiser et al. (2021). The smoke was then obviously preferably transported into the Arctic (Sorenson et al.

al., 2022, Xian et al., 2022, parts 1 and 2). The AOT over the Arctic region (shown as daily means, 70-90°N mean) almost monotonically increased from 22 June 2019 to 11 August 2019. There was already a period with strong fires over central eastern Siberia on 4-5 July 2019 (Johnson et al., 2021). However, the most intense fire period was from 19 July 2019 to 14 August 2019 and pushed probably self-lofting processes a lot in August. All this is now given in the revised version of the manuscript (Section 5).

Line 555-557, "Under these conditions with very large AOT values over extended Siberian and Arctic terrain one can assume that there were several subregions with AOTs>1.5 over days if not for more than a week so that the probability for self-lofting events was high in July and August 2019.": Here O22 unambiguously describe the conditions that are favorable for eventual lofting of smoke to the UT. It involves days of super large AOT in the lower troposphere. O21 claimed that these conditions ensued after ~21 July. So how can any of the CALIOP layers they discuss on 15, 25, and 26 July be the result of this mechanism? At the very least O22 are encouraged to abandon the O21 source term and find another high AOT event before 15 July (their first day of tropopause-level diffuse smoke attributed to this pathway).

We changed the text regarding this in Section 5 as explained several times above. Furthermore, we leave the options more open. Besides AOT, BC fractions play an important role, and even at very low AOTs lofting takes place but may take weeks before the smoke traces show up at the tropopause.

Line 563-564, "For comparison, a Raikoke-related AOT of 0.025 was expected at 532 nm at high northern latitudes considering the emitted SO2 mass of around 1.5 Tg (Ohneiser et al., 2021).": AOT observations of Raikoke sulfates reported in this paper far exceed 0.025. This is at odds with O21. O21 did not argue the 0.025 limit based on SO2-sulfate conversion calculations, but rather other observations such as Kloss et al. As previously discussed in prior reviews, Kloss et al.'s AOT values were biased low with respect to point measurements such as those from lidar.

There is nothing left to explain. As mentioned Pinatubo SO2 of 20 Tg produced maximum AOTs of 0.3 (when the sulfate was uniformly distributed over the Northern Hemisphere in the winters of 1991-1992 and 1992-1993). So, the Raikoke maximum AOTs can only be as high as 0.025 (for uniformly distributed sulfate aerosol) because the emission was below 2 Tg. Uniformly distributed aerosol is observable after about 2-3 month after the volcanic event in the case of mid latitude volcanoes. All this is quite trivial.

Final remark: I asked Corinna Kloss why she did not take the Siberian smoke into account in her paper (Kloss et al., 2021), and she told me: We did not know about it. That's why we have this paper exclusively discussing all findings in terms of Raikoke aerosol.

**We need to stop now this endless discussion.**

Line 576-577, "Such a high lidar ratio has never been observed for volcanic sulfate aerosol.": Perhaps until now. Given that O22 have perhaps inadvertently demonstrated the omni-presence of Raikoke sulfates, with small and large AOT, from the tropopause to lower stratosphere, days to weeks before smoke could have entered the UTLS in abundance, it may be reasonable to conclude that indeed some sulfates may have the optical properties that O22 relegate to smoke presence. Please comment on this and/or make suitable revisions.

We strongly disagree with the reviewer statement. As long as all observations and all simulations, after moderate to major eruptions, show that the 532 nm SULFATE lidar ratio was clearly below 50 sr (and mostly even below 30 sr during the first year after the eruption), and all SMOKE observations show lidar

ratios at 532 nm of typically 70 sr and more and at the same time mostly lidar ratios of 40-55 sr at 355 nm, there is no (zero) room for discussion on the observed aerosol type.

The aerosol over the High Arctic during MOSAiC showed unambiguously: dominance of smoke. We explained all this exhaustingly in the Ohneiser et al. (2021) paper and in every of the numerous reply letters to these questions and comments of Mike Fromm in the case of previous papers. Mike Fromm comes again and again with same comments.

Unfortunately, there was no time period during which the Raikoke aerosol was the only aerosol type in the stratosphere. Sulfate obviously dominated in most of the patchy aerosol features at heights clearly in the stratosphere in July 2019. Nevertheless, there was no time period with pure Raikoke aerosol aerosol conditions because the fire season began around 15 June 2019 in central eastern Siberia and the Raikoke eruption occurred on 22 June 2019. The maximum of the SO2 to sulfate conversion was in mid-August, and the smoke occurrence in the upper troposphere over the Arctic was at its maximum in mid-August.

Line 577-578, "The particle depolarization ratios were <0.03 at both wavelengths (as given in Fig. 15), a clear signature of perfect spherical particles, ...": Agreed. And perfect spherical particles are not the norm for tropospheric smoke of this age. See my prior review and comments above. Please explain how the previously published reports of aged tropospheric and pyroCb smoke with depolarization ratios ~0.03-0.11 fit into O22's interpretation.

As we already explained above, the particle depolarization ratio is needed to identify the aerosol type, or better the identify the shape of the particles. The volume depolarization ratio is often strongly influenced by the Rayleigh depolarization ratio which is close to zero. The Rayleigh impact on the volume depolarization ratio needs to be removed before you can use the remaining particle depolarization ratio for aerosol typing. And if the volume depolarization ratio is already enhanced (0.03-0.11, and thus clearly above 0.01) the particle depolarization ratio was obviously 0.1 to >0.2. To repeat again: If the particle depolarization ratio is close to zero than the particles are spherical, no doubt. If the particle depolarization ratio of 0.11 are observed, particle depolarization ratios are probably >0.2.

Line 579-580, "In cases of pyroCb-aided lofting, the depolarization ratios were always observed to be >0.1 during the first month after the pyroCb events.": This was not the case for Norman Wells (Siebert et al., 2000) or Chisholm (Fromm et al. 2008). This disparity must be recognized, acknowledged, and explained.

We explained this point above. All observations of pyroCb-lofted smoke clearly shows enhanced particle depolarization ratios. There is no (zero) exception. Many scientists however use volume depolarization ratios, and that is the source for all the confusions. But we agree we should often precisely write: particle depolarization ratio, and not simply: depolarization ratio.

Line 585-586, "We therefore have our doubts that one can obtain a clear picture of the aerosol composition from infrared absorption spectra alone.": These doubts are well founded. It is prudent to doubt the full veracity of any composition determination based on any single remote sensing data item. That doubt applies equally to lidar backscatter, in this case the over reliance on spectral dependence of lidar ratio. Hence, doubts should be spread equally and the authors are advised to consider that. Perhaps more importantly, Boone et al. (2022) did not rely solely on the IR spectra. A full interpretation of that paper must take into account the associated ACE Imager aerosol extinction and ACE-FTS SO2 and HCN measurements. These orthogonal indicators were all presented together by Boone et al., leading to their robust conclusion of sulfate dominance over smoke.

The spectral dependence of the lidar ratio is the best fingerprint of aged smoke you can have: We use this fingerprint since 1998 as mentioned above. On the other hand, the lidar ratio for non-absorbing sulfate aerosol is so different. We are thus very happy to have this 'robust' lidar ratio parameter, in addition to the particle depolarization ratio (Ansmann et al., Frontiers, 2021). This is our clear opinion based on 24 years of smoke lidar observations (since the Lindenberg 1998 campaign and the related publication of Wandinger et al., 2002).

Here my opinion to the Boone et al. (2022) approach:

Mike Fromm believes that Boone et al. (2022) draw a very 'robust conclusion'. The main message of the Boone paper is: There was only sulfate aerosol (100% fraction!) in the lower stratosphere over the Arctic in September and October 2019. Boone et al. (2022) did not find any indication for the presence of smoke (0% fraction) from their ACE-FTS observations!

This is quite a rather surprising and to my opinion an absolutely wrong conclusion if we take the UTLS AOTs observed with the ACE-FTS 1020 nm extinction channel into account. By the way, the authors did not provide any AOT values in their paper. Nevertheless, I evaluated the extinction profiles in Figure 5 of their paper by myself. The September and October mean AOTs (in the 60-80°N latitudinal belt) were about 0.03 at 1020 nm in their Figure 5. The 1020 nm AOT values correspond to 550 nm AOTs around 0.1, in excellent agreement with the MOSAiC lidar observations in the High Arctic in October 2019 (Ohneiser et al., 2021). These AOTs around 0.1 are, again, in strong contradiction with the Raikoke sulfate AOTs that could only be of the order of 0.01-0.015 in October 2019, according to the law concerning the emitted SO2 amount and resulting hemispheric mean AOT. In September and October 2019, the sulfate was sufficiently uniformly distributed so that these values of 0.01-0.015 can be used as constraint in the analysis of aerosol observations. So, the sulfate fraction was about 10-15% and the smoke fraction was 85-90% (Ohneiser et al., 2021). And this is exactly in agreement with the lidar ratio observations assuming 15% sulfate (lidar ratio of 40sr) and 85% smoke (lidar ratio of 85sr). Sadly, the 'robust conclusions' of Boone et al. (2022) are the opposite. To my opinion, Boone et al. (2022) as well as Mike Fromm did not know about this well-established context of SO2 emission and resulting sulfate AOT, and that is the reason that they never checked the AOT in Figure 5 and mentioned that there is a discrepancy between their observed and the expected Raikoke AOT of 0.003-0.005 (at 1020nm).

Furthermore, it is impossible to draw 'robust conclusions' when having only observations in the infrared wavelength spectrum as is the case for ACE-FTS. Simple Mie computations immediately show that this is impossible. The particle sizes of the volcanic sulfate particles and of the smoke particles were both between 100 and 1000 nm (in diameter). Thus, the particles formed a so-called accumulation mode. Proper measurement wavelengths are then, e.g., 340 to 870 nm (exactly the wavelength at which the AERONET photometers measure). With these wavelengths one can retrieve microphysical and refractive index properties of the measured aerosols. To have additional measurement wavelengths of 1020 nm or 1640 nm (in the case of AERONET) is of advantage to identify the contribution of coarse mode particles (like mineral dust particles) properly. So, our Raman lidar wavelength at 355 and 532 nm are excellent wavelengths for aerosol monitoring in terms of backscatter, extinction, lidar ratio, depolarization ratio and derived microphysical properties as shown for the Arctic aerosol in the articles of Engelmann et al., (ACP, 2021) and Ohneiser et al. (2021). All the 'orthogonal indicators' (FTS and 1000 nm ext information) do not help to obtain trustworthy microphysical and refractive index properties of the accumulation-mode particles.

**Line 587-593, discussion of AIRS CO:** It is essential for O22 to show these results such that they can be evaluated and reproduced. Their claim here is brand new and of fundamental substance. The AIRS averaging kernels are such that the CO signal peaks between 300-600 hPa, so a strong signal at 100 hPa cannot be divorced from the total column amount. Moreover, a check that I performed on Siberia/Arctic AIRS August monthly CO showed no obvious enhancement at 100 hPa. More reason for O22 to fully lay out this analysis. For additional benefit to the authors I append below an analysis similar to that shown above, combining CALIOP 532 nm attenuated backscatter coefficient and ACE data for daily coincidences over Siberia between 20-26 July 2019. Interpretations are given with each set of ACE plots and a wrap-up discussion follows all. In a nutshell, the ACE data show SO2 enhancements with each coincidence and an absence of CO enhancement within any aerosol layer. Hence, over Siberia at the critical time of hypothesized self-lofting, all UTLS aerosols are combined with sulfur enhancement. One can compare the 25 and 26 July examples with O22 Fig. 14.

The AIRS satellite data analysis was performed by our satellite data expert Alexandra Chudnovsky from Tel Aviv University. The figure below shows our basic result for the month of August of the last 10 years. I have no doubt that we made a careful job when looking at the AIRS data. We see clearly that there was enhanced CO in 2019 and 2021 in the data from 2016-2022 for the 150 hPa, 100 hPa, and 50 hPa time series. In Section 5 we now mention: .... and found a clearly enhanced monthly mean CO concentration in the upper troposphere and lower stratosphere (100-150 hPa) in August 2019 compared to the respective August mean values of the background years 2013-2018, 2020, and 2022.

We think, an additional figure, just to show a few AIRS results, in this self-lofting paper with focus on selflofting simulations and self-lofting observations is not justified. We will therefore not add another figure. It is sufficient to provide the general result of our AIRS results in the text.

AIRS, August mean values for the box from 67°-143°E and 70°-87°N, for the years from 2011 to 2022. Years with record-breaking fires were 2019 and especially 2021. Green for 150 hPa, red for 100 hPa, and blue for 50 hPa

**The final attempt: Mike Fromm's synthesis**

At the end of the review, Mike Fromm added further figures concerning ACE-FTS observations (SO2, CO, particle extinction profiles) in the lower stratosphere at 68°N on 20-26 July 2019.

We discussed details for the 15 July 2019 already above. The conclusions drawn for the 15 July case are similar to the ones here, for the observations on 20-26 July 2019. So we leave out to discuss all this again in detail (day by day).

But Mike Fromm presented the following conclusion based on the analysis of the ACE-FTS observations at 68°N (over central eastern Siberia) on 20-26 July 2019.

Mike Fromm's synthesis, points 1 to 4:

1) UTLS diffuse backscatter enhancements each and every day between 20-26 July over Siberia. In every example, ACE S02 was enhanced in the diffuse backscatter layer. Excellent correspondence between ACE Imager aerosol enhancement and CALIOP backscatter.

2) On a couple occasions there was a CO enhancement in the lowermost portion of the most prominent ACE layer. These UT enhancements were at some points associated with no discernable Imager extinction enhancement. There were no CO enhancements in the prime diffuse backscatter layers.

3) In the AOD onset period (20-22 July) there was no justification for locally lofted smoke to the UT. Hence, the UT CO enhancements were likely aged air from another source.

Concerning 1) and 2): First of all, I was surprised about these simple conclusions. I see very complex aerosol conditions in the lower stratosphere and quite significant differences between the CALIOP backscatter and the ACE-FTS extinction values. Furthermore, the observations presented by Mike Fromm (ACE-FTS vs CALIOP) could not be taken at the same time (3-6 hours difference). Both instruments observed different aerosols (in this July 2019 with so inhomogeneous aerosol structures). When keeping all these aspects in consideration, yes then the comparison is not too bad.

And concerning SO2 enhancement in the lowermost stratosphere. This indicates the presence of Raikoke influenced air masses above the tropopause. Not more! This does not tell us anything about the aerosol mixing state (smoke-sulfate mixture). The CO content was enhanced as well up to the tropopause and even higher up (e.g., on 23 July 2019). That tells me there was smoke-filled air up to the tropopause and sometimes above the tropopause as well.

Just to emphasize that again: 'Our' diffuse layers were at all below the tropopause, and the ACE-FTS extinction observations are trustworthy only above the tropopause. All the examples (15 July and 20-26 July 2019) presented by Mike Fromm do not help! They provide no hint on the aerosol properties and aerosol type of the diffuse layers below the tropopause.

Now, what is the most probable solution for the aerosol characteristics in the lowermost stratosphere covered by ACE observations. The most logical explanation is simply: The enhanced extinction coefficients are caused by sulfate aerosol, formed within the stratosphere from the Raikoke SO2, as well as by smoke particles, ascending into the stratosphere from the troposphere. Mike Fromm seems to believe that the presence of SO2 is an unambiguous sign for the dominance of sulfate aerosol in the lower stratosphere. This conclusion is simply too simple. Mike Fromm argues that the CO content was partly zero in the SO2 layer which means that there was no smoke in the SO2 layer and thus only sulfate (100% fraction). As mentioned there was always CO up the tropopause (that already demonstrates that smoke-containing air was able to make it up to the tropopause by self-lofting processes) and in some cases, especially on 23 July, there are clear signs that smoke penetrated several kilometers into the lower stratosphere.

**So, the conclusion can only be: *The aerosol layer in the lower stratosphere in July 2019 consisted of a mixture of smoke and sulfate aerosol.**

Concerning point 3) Mike Fromm's original comment: CO enhancements were likely aged air from another source.

We do not accept this trivial statement. There were huge, record-breaking fires over central-eastern Siberia and then.... the found enhanced CO levels had likely nothing to do with this record-breaking fire storm??? This hypothesis makes no sense at all.

Point 4 of Mike Fromm's synthesis: Conclusion: Raikoke sulfates clearly overwhelm any other explanation for UTLS aerosols over Siberia when the diabatic lofting of smoke was hypothesized to have started. The relatively strong but diffuse CALIOP backscatter is demonstrably sulfate in all the examples presented

Again and again: These trivial remarks and conclusions and the exaggerated wording are not constructive, are not justified, are totally wrong. In our reply letter we provided detailed explanations on potential sources of the complex pollution states. We should stop now with all this.

At the end: The manuscript is revised and Section 5 now even contains comprehensive information on fire season begin, duration of the main fire season, smoke transport towards the High Arctic using the most recent publications (Johnson et al., Sorenson et al., Xian et al., Ohneiser et al.).

The manuscript is therefore ready for the next round of reviews with new reviewers.